# On the Fragility of Latent Knowledge: Layer-wise Influence under Unlearning in Large Language Model

## Abstract

Large language model (LLM) unlearning has emerged as an essential post-training mechanism for erasing specific knowledge or undesirable behaviors. However, forgetting target data often causes an unintended degradation in overall model utility. Although various advanced methods have explored different learning objectives to mitigate the trade-off, it remains unclear how the highly entangled internal representations in LLMs contribute to unlearning. In this work, we introduce the notion of *latent knowledge fragility* to explore the vulnerability of retained knowledge to unlearning. We develop a unified analytical approach via component-wise parameter patching that isolates and quantifies fragility in terms of different transformer blocks. We observe that the LLM encodes different levels of abstraction, from surface syntax in shallow layers to complex semantics in deeper layers, which align with different degrees of representation disruption and utility degradation. Based on the insights, we propose a lightweight framework called *Component-wise Replacement Unlearning* (CRU) that restores fragile layers (also extendable to other components) from the original model based on post-hoc validation, which allows us to obtain a hybrid model without additional training. Extensive experiments on various aspects verify that our method generally improves the trade-off between removal and retention. Our analysis highlights the non-uniform influence of different LLM layers and provides a new possibility of surgical unlearning.

## 1 Introduction

The unprecedented scale and generalization capabilities of large language models (LLMs) [2, 86, 18, 62, 22] have led to significant successes in understanding and generating natural languages for complex tasks [21]. While being widely deployed, LLMs also bring a primary concern given their high tendency to memorize training data [9, 10]. As trained in a broad range corpora [2], some sensitive or even harmful information poses various risks for LLM usage [29], regarding data privacy (e.g., GDPR compliance [51, 84]), ethics [30], safety [88, 28], and intellectual property [79]. In contrast to costly retraining from scratch, LLM unlearning [80, 85, 26, 75, 74] has emerged as an alternative to mitigate the problem, which often involves fine-tuning the model with gradient-based objec-

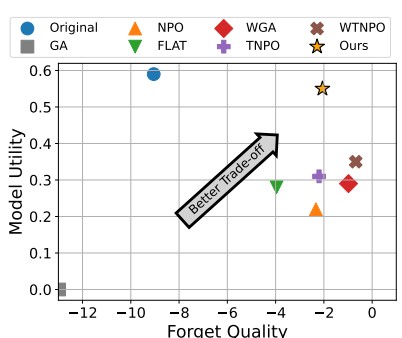

Figure 1: **Our method can achieve a better trade-off** than previous methods.

tives [63, 64, 80, 85, 75] to suppress unwanted behaviors [76] or remove specific knowledge [32, 26].

Despite promising progress in forgetting target content, it remains quite challenging to maintain the overall model utility of LLMs, measured via the discrepancy between outputs of the original and the unlearned model. Taking the gradient ascent (GA) method [80] as a representative example, it directly minimizes the log-likelihood for targeted data to reduce their generation probability, but it can also easily destroy the ability to generate natural language. Subsequent methods [85, 75, 74] developed various advanced objectives to address the excessive unlearning, which still induce collateral degradation in the model's general language capabilities as evident in Figure 1 (evaluated

Figure 2: **Our layer-wise patching approach illustrates the intrinsic functionality differences of different transformer layers.** Highlighted are distinction from the original answer. The shallow layers near token input model surface-level syntax, such as word order and lexical details. The middle layers model entangled knowledge with abstract concepts that encode a range of complex semantics. The deep layers near the output model token-level dependencies, such as contextual correlations.

on TOFU benchmark [32]). However, beyond the design of unlearning objectives [29], the internal representation in LLM received limited attention, which naturally motivates one research question:

*Can we optimize the tradeoff through the lens of LLM latent knowledge?*

In this work, we discuss this trade-off from the intuition that different internal parts of LLM encode different knowledge (refer to Figure 2). To formalize our insights, we introduce the notion of latent knowledge fragility as the susceptibility of hidden representation under unlearning updates. This fragility is not uniform, but rather structured, reflecting a spectrum from low-level syntactic patterns to high-level complex semantics. Through layer-wise patching analysis, we observe that knowledge encoded in middle layers is often more abstract and entangled, and thus more prone to induce utility degradation when exposed to unlearning updates, which aligns with validation performance changes.

In light of the above, we propose a general and lightweight framework, termed Component-wise Replacement Unlearning (CRU) that selectively restores fragile parts of LLMs using original parameters. We mainly focus on layer-wise unlearning but our approach is easily extendable to other components. Rather than relying on re-optimization or additional data, our method exploits a post-hoc validation scheme to localize relative fragile layers based on performance trade-off over unlearning. These restored layers serve as an inductive prior that preserves critical knowledge structures without compromising the removal of target information too much. Notably, this design circumvents fine-tuning or architectural changes, making it applicable across various unlearning settings and model scales.

We evaluate our method with multiple LLMs in different unlearning scenarios like personal content or factual knowledge removal. Experimental results consistently show that our approach improves the removal-retention trade-off, achieving satisfactory target forgetting without sacrificing much model utility. Furthermore, qualitative analyses reveal that our method retains coherent and grounded generations, especially in answers requiring semantic understanding. Interestingly, we also find that our final hybrid model of CRU achieves a better trade-off in a very distinct way from other models in terms of parameter changes. To summarize, our main contribution can be listed as follows.

- We introduce a notion of latent knowledge fragility to qualitatively define how unlearning updates affect different levels of latent knowledge encoded in LLMs. (in Section 3.1)
- We develop a unified analytical approach to quantify layer-wise fragility by analyzing the modular influence of LLMs in relation to the performance trade-off. (in Section 3.2)
- We propose a lightweight and general framework termed CRU that selectively restores fragile layers from the original model based on post-unlearning validation, effectively improving the trade-off between forget quality and model utility. (in Section 3.3 and Section 4)

## 2 BACKGROUND

In this part, we introduce preliminary background of LLM unlearning and our layer-wise model patching. In Appendix B, we provide a more comprehensive discussion of related work.

**Problem Setup for LLM Unlearning.** We consider a pre-trained auto-regressive LLM $f_\theta$ with the model parameters $\theta$, which recursively estimates the probability distribution of the next token

$p(\cdot|s,\theta)$ given the input sequence $s = [s_1, s_2, \cdots, s_{|s|}]$. The model is assumed to be trained on a web-sourced corpora $\mathcal{D}_{\mathrm{w}} = \{s^1, s^2, \cdots, s^n\}$ with the negative log-likelihood (NLL) loss function of $-\log p(s;\theta)$, where $p(s;\theta) = \prod_{i=1}^{|s|} p(s_i|s_{1:i-1};\theta)$ indicates the product of conditional probability for each token given the prefix $s_{1:i-1}$. LLM unlearning [80, 32, 29] refers to a post-training paradigm that removes undesirable knowledge from the original models. Specifically, we are given a *forget set* $\mathcal{D}_{\mathrm{t}} = \{s_{\mathrm{t}}^1, s_{\mathrm{t}}^2, \cdots, s_{\mathrm{t}}^m\}$ that includes the data targeted to be erased, where usually $m \ll n$.

**Primary Goal and Tradeoff.** The goal of LLM unlearning is to construct a modified model $f_{\theta^{\mathrm{u}}}$ that suppresses the undesired knowledge associated with forget set $\mathcal{D}_{\mathrm{t}}$ (referred to *removal*), while preserving the model performance on the remaining data $\mathcal{D}_{\mathrm{r}} = \mathcal{D}_{\mathrm{w}} \backslash \mathcal{D}_{\mathrm{t}}$ (referred to *retention*). Due to the complexity and versatility of LLM [18, 2], the specific evaluation of unlearning also covers a wide range of aspects such as memorization [10], exploration [32], and coherency [29]. To ease our discussion, we mainly follow TOFU [32] focusing on two comprehensive metrics:

- **Forget Quality (FQ)** measures how effectively an LLM forgets specific information. It assesses the similarity between the outputs of an unlearned model and a retain model (trained without $\mathcal{D}_{\mathrm{t}}$) on the target data, which is quantified using statistical tests like the Kolmogorov-Smirnov test [34].

- **Model Utility (MU)** evaluates the unlearned LLM performance on data it was intended to retain. It ensures that the unlearning does not degrade the model's overall capabilities, and is calculated as the harmonic mean of various metrics on the retain set, such as accuracy, factuality and truthfulness.

We leave more metric details in Appendix E.1. The inherent tradeoff between removal and retention is evident in Figure 1 and also revealed in previous works [32, 74, 75], e.g., unlearning methods increase FQ by effectively forgetting targeted information, but often inadvertently reduce MU, impairing the model's performance on retained knowledge, which is a primary challenge in the area of research.

**Representative Unlearning Methods: GA [80] and NPO [85].** There are various advanced methods [75, 85, 15] on objective design for unlearning, which are mainly based on two representative approaches for erasing knowledge. The first is Gradient Ascent (GA), a fundamental method in LLM unlearning that directly minimizes the log-likelihood of target data via $\mathcal{L}_{\mathrm{GA}}(\mathcal{D}_{\mathrm{t}};\theta) = \frac{1}{n}\sum_{s \in \mathcal{D}_{\mathrm{t}}} \log p(s;\theta)$. To refine the objective of GA for mitigating the excessive unlearning [32] that can easily disrupt the whole LLM, Negative Preference Optimization (NPO) derives an variant from DPO [47] to perform an instance-reweighted unlearning, following the objective as $\mathcal{L}_{\mathrm{NPO}}(\mathcal{D}_{\mathrm{t}};\theta) = \frac{1}{n}\sum_{s \in \mathcal{D}_{\mathrm{t}}} \frac{2}{\beta} \log \left[ 1 + \left( \frac{p(s;\theta)}{p(s;\theta^{\mathrm{orig}})} \right)^{\beta} \right]$. A series of later methods focus on objective-level developments by adding regularization on non-target data [32], token-wise reweighting [74], and gradient rectification [24], while the impact on latent knowledge is underexplored for the trade-off.

**Layer-wise Model Patching.** To isolate and explore the effects of unlearning at the internal of LLM, we introduce a layer-wise model patching approach. Previous studies in other domains like representation geometry [45, 57, 58, 39] and mechanism interpretability [6, 49, 83] (further discussed in Appendix B) have shown that the transformer-based models encodes distinct types of linguistic and conceptual information across the model. Given the original model $f_\theta$ and an unlearned model $f_{\theta^{\mathrm{u}}}$, we define a hybrid reference model (see Definition 3 for a formal version) $f_{\theta^r}^{\phi}$ that selectively inherits layers from $f_{\theta^{\mathrm{u}}}$: $f_{\theta^r}^{\phi}(x) = f_\theta^{(L)} \circ \cdots \circ f_{\theta^{\mathrm{u}}}^{(l \in \phi)} \circ \cdots \circ f_\theta^{(1)}(x)$, where $\phi \subset [1, \cdots, L]$ indicates the model parameter of which layer comes from the unlearned model. This formulation allows us to empirically assess the influence of each layer under unlearning by evaluating the retention and removal performance under controlled layer substitutions. It is also straightforward to extend to other components (e.g., attention head, MLP or others) of the LLM as well. In this work, we mainly explore layer-wise patching in light of **two major considerations:** 1) the layer serves as a proper model deconstruction unit with a small search space compared to more fine-grained choice, e.g., there are 32x more attention heads than layers in LLama3.2-1B-instruct [71]; 2) it is more architecture-agnostic and naturally aligns with the modularity for knowledge abstraction [19]. We will discuss more later.

## 3 DELVING INTO THE INTERNALS OF LLM FOR UNLEARNING

In this section, we explore the impact of unlearning from the viewpoint of LLM internal representations. First, we introduce our motivation and knowledge fragility (in Section 3.1). Second, we provide a systematic analysis to understand the knowledge influence in different layers (in Section 3.2). Lastly, we propose a general framework, i.e., component-wise replacement unlearning (in Section 3.3).

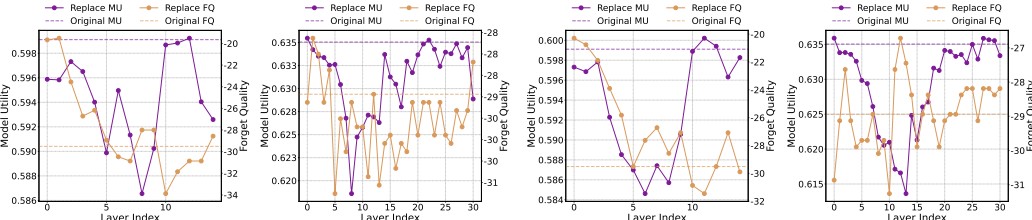

(a) with GA unlearned on 1B and 7B LLM                 (b) with NPO unlearned on 1B and 7B LLM

Figure 3: **Patching layers reveals the middle ones generally causes the significant utility degradation in our tests.** Layer-wise patching from the unlearned LLM on the original LLM to investigate the forget quality and model utility of updated layer on LLMs (left: Llama3.2-1B, right: Llama2-7B).

### 3.1 Motivation: Hypothesis on Structurally Stored Latent Knowledge

Regarding LLMs trained on massive web-sourced corpora [2, 18, 86], the latent knowledge encoded in the representation space is highly complex and entangled. However, in LLM unlearning, the target data for removal usually account for only a small part, and we can hard to pre-define the retention goals which accurately mitigate the excessive forgetting effects. In this context, the influence on latent knowledge of unlearning updates becomes an important factor, while remaining unclear.

**Latent knowledge is structurally encoded in the internal representation.** As the transformer [71] consists of multiple modular blocks, previous works [82, 81, 25, 3] analyzed and revealed various mechanisms in transformer circuits [14]. One typical observation is the linear representation hypothesis [87, 65, 70, 69, 44, 45], which indicates that some high-level concepts (e.g., sentiment [42], truthfulness [25], and refusal [3]) are linearly encoded at some point in the residual stream of the model, and can be manipulated by intervening on attention heads [25] or directly on the residual stream [35, 16]. It also implies that the residual stream contains multi-level conceptual knowledge that entangled and is worth exploring from the latent intrinsic structure. The intuition is that if a concept is added to the residual stream by some component, we can achieve high forget quality by replacing that component with its unlearned counterpart; to maintain high model utility, we must also ensure that the component is not responsible for adding any other concepts to the residual stream.

**Unlearning as a reverse process on exploring knowledge composition.** How knowledge is composed in the original LLM internals matters the difficulty of unlearning to achieve a satisfactory trade-off, especially for those scenarios without including full non-target data for regularization. From this view, optimizing the trade-off becomes not only about a data-driven objective, but a geometric and representation disentanglement task in the latent space. To intuitively present our findings, we first introduce the following qualitative definition of latent knowledge fragility to start our analysis.

**Definition 1** (Latent Knowledge Fragility). *The unintentional influence of unlearning that attempts to remove specific knowledge (e.g., via removing $\mathcal{D}_t$) on distorting latent representation of LLM.*

To explore the effects, we use the aforementioned layer-wise model patching to isolate and examine the unlearning updates of each transformer blocks on outputs, which reveal distinct qualitative results.

**Non-uniform influence from different layers.** Although indiscriminately updating on the whole model can finally achieve the removal target on generating irrelevant answers, the different parts in the LLM internal structure can have non-uniform influence on the final output generation. As demonstrated in Figure 2, the final answers show three types of distinctions from the original output when patching the updated layers from a unlearned model to the original model in shallow, middle, and deep layers. Intuitively, the shallow layers change the entity or syntax content like words orders, the middle layers change high-level semantics with additional information, and the deep layers exhibited disruptions in the natural correlations among words, leading to repetitive use of "vivid".

### 3.2 Quantifying and Interpreting the Influence of Different Layers

To study the trade-off between FQ and MU, we quantitatively estimate the fragility of initial knowledge encoded in different layers, for which we can use the validation-based performance change,

$$\text{Fragility Estimation:} \quad \mathrm{S}_{\mathcal{R}}(l) := \mathcal{R}\left(f_{\theta^{\mathrm{r}}}^{\phi=[l]}; \mathcal{D}_{\mathrm{val}}\right) - \mathcal{R}\left(f_{\theta}; \mathcal{D}_{\mathrm{val}}\right), \tag{1}$$

where $l$ indicates the specific layer, $\mathcal{D}_{\mathrm{val}}$ is subset from $\mathcal{D}_t$ or $\mathcal{D}_r$ corresponding to removal or retention validation set, and $\mathcal{R}$ is the performance measurement of removal or retention part, e.g., FQ and

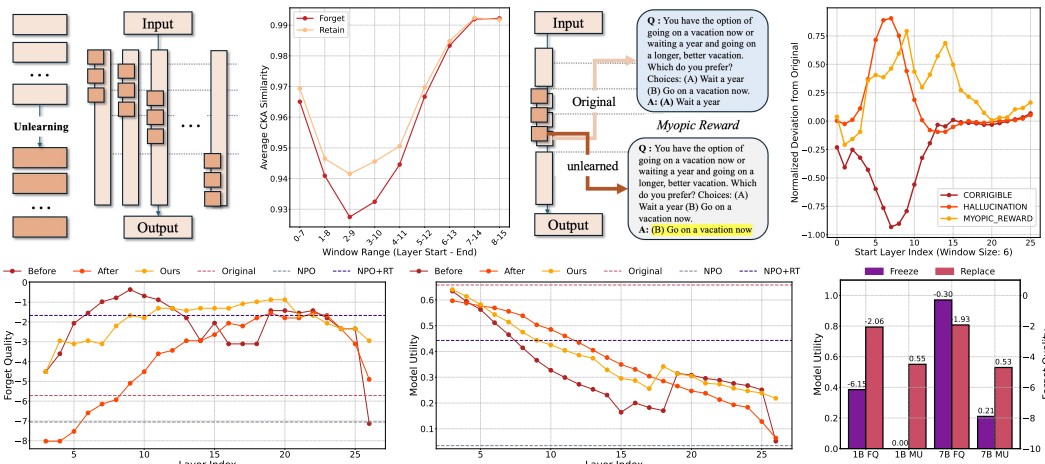

Figure 4: **Left Top:** Patching with sliding windows for representation similarity check using CKA [23]. **Right Top:** investigating the middle layers for encoding high-level concepts [46] before and after unlearning. **Bottom:** the opportunity and rationality on the choice of layer-wise replacement.

MU. In Figure 3, we present an overview of FQ and MU by patching different single layer from the unlearned model (via GA or NPO) to the original ones (pre-unlearned) on the TOFU benchmark [32].

Generally, we find that MU results show an obvious "U Shape" across different setups, which can be divided into three parts to discuss the relative fragility among layers. Note that both FQ and MU are the larger, the better. In shallow layers, both FQ and MU are high for unlearning updates, indicating that low fragility and retention are desirable. In middle layers, a consistently lowest MU value is observed across all results, indicating that these layers encode entangled concepts that are more susceptible to disruption. In later deep layers, although the unlearning update will not affect the MU, we also have a low FQ, indicating that the removal target is less relevant to contextual correlations.

**Interpreting via representation drift.** To explore the underlying mechanism of the lowest MU in the middle layers patching, we further investigate the representation similarity via Centered Kernel Alignment [23] on the latent representation space. In the left top of Figure 4, we reveal that the hidden output similarity of both removal and retention parts drop significantly for the middle layers (specifically localized by our sliding window), which aligns with the previous "U Shape" in Figure 4. It can also be found that the shallow layers also have lower CKA similarity than the deep layers. Assuming the linear representation hypothesis (formal proof is provided in Appendix C), we can obtain the proposition relating the latent knowledge fragility to the representation drifts.

**Proposition 1** (Representation drifts with fragility). *Let $\Phi_\ell^{orig}, \Phi_\ell^{unlearn} \in \mathbb{R}^{n \times d}$ denote the centered hidden representations at layer $\ell$ for a retained dataset $\mathcal{D}_r$ before and after unlearning, respectively. Define the concept-subspace representations as: $Z^{orig} := \Phi_\ell^{orig} P_c, Z^{unlearn} := \Phi_\ell^{unlearn} P_c \in \mathbb{R}^{n \times k}$.*

*Let linear CKA similarity between them be: $CKA_c := \frac{\|Z^{orig\top} Z^{unlearn}\|_F^2}{\|Z^{orig\top} Z^{orig}\|_F \cdot \|Z^{unlearn\top} Z^{unlearn}\|_F}$, and $W_c \in \mathbb{R}^k$ be a linear readout. Then the average output shift due to unlearning at layer $\ell$ satisfies:*

$$\frac{1}{n} \sum_{i=1}^n \left\| f_{\theta^r}^{\phi(l)}(x_i) - f_\theta(x_i) \right\|_2^2 \geq \|W_c\|_2^2 \cdot \left( (\sigma_c^{orig})^2 + (\sigma_c^{unlearn})^2 - 2\sqrt{CKA_c} \cdot \sigma_c^{orig} \cdot \sigma_c^{unlearn} \right) \quad (2)$$

*where $\sigma_c^{orig\,2} := \frac{1}{n}\|Z^{orig}\|_F^2$, $\sigma_c^{unlearn\,2} := \frac{1}{n}\|Z^{unlearn}\|_F^2$ and $S(l) \propto \frac{1}{n} \sum_{i=1}^n \left\| f_{\theta^r}^{\phi=[l]}(x_i) - f_\theta(x_i) \right\|_2^2$.*

**Side-effects for unexpected concept intervention.** As a straightforward corollary of Proposition 1, the representation drifts on middle layers can also induce the unexpected intervention for high-level concepts. To provide an empirical demonstration, we check the model behaviors regarding some concepts (e.g., Corrigible, Hallucination, and so on) from "Advanced AI Risk" [46] before and after layers patching. In the right of Figure 4, we find the middle fragile layers are most affected by unlearning updates and consequently also change the LLM's inclination towards those concepts.

**Potential trade-off space regarding layer stacking.** Since we have one general observation that the middle layer is most fragile under unlearning, we can have two heuristic strategies to restore the

specific layers from original model to maintain the general capability of LLM, which are denoted as "Before" and "After". The former utilizes unlearned layers before specific index and restores the rest layers from original model, while the latter utilizes those after specific index reversely. The results in Figure 4 show there is a potential search space for a better trade-off even for NPO with retention data.

### 3.3 COMPONENT-WISE REPLACEMENT UNLEARNING

Based on our previous insights, we introduce the new Component-wise Replacement Unlearning (CRU) to partition and replace critical parts of LLM for unlearning to restore the general retention knowledge (refer to Algorithm 1). We first present a general definition and then shift our focus to the layer-wise case. For an integer $n > 0$, we let $[n] := \{1, 2, \ldots, n\}$ and we have the following.

**Definition 2** (Component-wise partitioner). *Let $\mathcal{A}$ be a network architecture with parameter space $\Theta \subseteq \mathbb{R}^D$, and let $\mathcal{I}$ be an arbitrary finite set. A* component-wise partitioner *is a function $\rho \colon \mathcal{I} \to [D]$ such that $\rho(I) \cap \rho(I') = \varnothing$ for any $I, I' \in \mathcal{I}$ such that $I \neq I'$. We call $\mathcal{I}$ the* index set *of $\rho$ and $|\mathcal{I}|$ the* size *of $\rho$. For a fixed $\rho$, we let $\theta^{(I)} = (\theta^i)_{i \in I}$ denote all components of $\theta$ associated with index $I$.*

Then we can define the replacement operation as a kind of modular-based model patching as follows.

**Definition 3** (Patched model). *Given two parameters $\theta_{\mathrm{orig}}, \theta_{\mathrm{new}} \in \Theta$ and a patching vector $\boldsymbol{\alpha} \in \{0, 1\}^{\mathcal{I}}$, we define the* patched parameter $\theta_{\boldsymbol{\alpha}}$ *in the following component-wise manner:*

$$(\theta_{\boldsymbol{\alpha}})^I = (\theta_{\mathrm{orig}})^I, \quad \text{If } \alpha_I = 0; \text{ otherwise, } (\theta_{\boldsymbol{\alpha}})^I = (\theta_{\mathrm{new}})^I. \tag{3}$$

*i.e., $\alpha_I = 0$ denotes that $\theta_{\boldsymbol{\alpha}}$ takes the same values as $\theta_{\mathrm{orig}}$ at component $I$, whereas $\alpha_I = 1$ denotes that $\theta_{\boldsymbol{\alpha}}$ takes the same values as $\theta_{\mathrm{new}}$ at component $I$.*

In the layer-wise case, let $\mathcal{A}$ be a transformer-based architecture of LLM with parameter $\theta_{\mathrm{orig}}$ and $L$ layers, the *layer-wise partitioner* $\rho_{\mathrm{layer}}$ has an index set $\mathcal{I}_{\mathrm{layer}} = [L]$, and for any $l \in \mathcal{I}_{\mathrm{layer}}$, $\theta^{(l)}$ denotes the parameters of the $l$-th layer. We have an unlearned LLM with parameter $\theta_{\mathrm{new}}$ with vector $\boldsymbol{\alpha}$ to obtain a hybrid model $\theta_{\boldsymbol{\alpha}}$. For example, $L = 5$ and $\boldsymbol{\alpha} = [1, 0, 0, 0, 1]$ denote restoring the middle three layers of parameter from the original model to the unlearned model. The problem then can be formulated as finding a $\boldsymbol{\alpha}$ to achieve a highest score, e.g., FQ and MU for optimizing unlearning trade-off. In particular, by limiting using $k$ layers from the unlearned model, we show a surprisingly simple solution through the newly defined score and take the top-$k$ layer index as final $\boldsymbol{\alpha}$.

**Definition 4** (Patching Score via Sorted Indices). *Given the index set of candidate layers $\mathcal{I}_{\mathrm{layer}} = [L]$, we define the* patching score $\mathcal{M}(l)$ *for each layer $l \in [1, L]$ as the sum of its ranks in two sorted lists: one based on MU and the other on FQ. Let $\mathcal{T}_{MU}(l)/\mathcal{T}_{FQ}(l)$ denote the rank index of layer $l$ when all layers are sorted in descending order of $S_{MU}(l)/S_{FQ}(l)$ as Eq. 1. Then, the score is defined as:*

$$\mathcal{M}(l) = \mathcal{T}_{MU}(l) + \mathcal{T}_{FQ}(l). \tag{4}$$

*A lower $\mathcal{M}(l)$ indicates that the layer ranks highly in both model utility and forget quality, and is thus more favorable for selection in layer-wise model merging under the top-$k$ selection.*

This top-$k$ solution is based on the defined score without continuous assumption like previous "Before" and "After", and is verified to be effective in our scenario without exhaustive searching on the full space of size $\binom{L}{k}$, which can be further related to prior work on interchange interventions [16] and Shapley interaction [61]. Similarly, the component can be straightforwardly extended to other fine-grained ones like MLP or attention heads, **the choice on layer-wise** is on the algorithm complexity, which will be significantly increased given the enlarged search space. We provide the pseudo-code implementation with extended discussion in Appendix D. In the right of Figure 4, we also investigate **replacement v.s. structural finetuning**, i.e., whether the layer fragility can serve as an algorithmic prior to be representation regularization during finetuning, such as freezing their parameter updates for unlearning finetune. However, the results show it induces severe model performance loss, as this disrupts learning dynamics, especially when the frozen layers are critical for routing representations.

## 4 EXPERIMENT

### 4.1 EXPERIMENTAL SETUPS

**Datasets.** In our experiments, we mainly explore unlearning methods using the Task of Fictitious Unlearning (TOFU) dataset [32], which serves as a representative benchmark in previous works [85,

Table 1: **In TOFU benchmark, our method can usually achieve the best MU while having satisfactory FQ.** Results of Llama3.2, Llama2 and Phi-3.5 models. More results refer to Table 15.

| NPO | ES-exact | | ES-perturb | | MU↑ | FQ↑ | GA | ES-exact | | ES-perturb | | MU↑ | FQ↑ |
|---|---|---|---|---|---|---|---|---|---|---|---|---|---|
| | retain↑ | unlearn↓ | retain↑ | unlearn↓ | | | | retain↑ | unlearn↓ | retain↑ | unlearn↓ | | |
| | | | | | | llama3.2-3B | | | | | | | |
| Original | 0.9013 | 0.9291 | 0.4241 | 0.4111 | 0.6579 | -5.7157 | Original | 0.9013 | 0.9291 | 0.4241 | 0.4111 | 0.6579 | -5.7157 |
| Unlearned | 0.0336 | 0.0287 | 0.0271 | 0.0281 | 0.0347 | -7.0539 | Unlearned | 0.0332 | 0.0282 | 0.0265 | 0.0281 | 0.0000 | -104.7672 |
| +RT (w. $\mathcal{D}_r$) | 0.1706 | 0.0650 | 0.1134 | 0.0678 | 0.4429 | -1.6705 | +1×KL (w. $\mathcal{D}_r$) | 0.0921 | 0.0282 | 0.0663 | 0.0281 | 0.3251 | -104.7672 |
| FLAT | 0.2489 | 0.1881 | 0.1481 | 0.1679 | 0.5000 | -2.3448 | +10×KL (w. $\mathcal{D}_r$) | 0.3521 | 0.0575 | 0.1437 | 0.0417 | 0.6222 | -4.7025 |
| TNPO | 0.0421 | 0.0282 | 0.0286 | 0.0281 | 0.4397 | -1.4255 | +20×KL (w. $\mathcal{D}_r$) | 0.8340 | 0.4356 | 0.3622 | 0.2506 | 0.6633 | -4.3228 |
| WTNPO | 0.0347 | 0.0282 | 0.0304 | 0.0281 | 0.4257 | **-1.3084** | WGA | 0.0342 | 0.0282 | 0.0277 | 0.0281 | 0.3511 | **-1.3084** |
| AltPO | 0.0356 | 0.0287 | 0.0280 | 0.0287 | 0.4899 | -1.4255 | SatImp | 0.0341 | 0.0282 | 0.0280 | 0.0287 | 0.3120 | **-1.3084** |
| Ours | 0.0999 | 0.0719 | 0.1058 | 0.0846 | **0.5117** | -1.5462 | Ours | 0.7251 | 0.2117 | 0.3677 | 0.1215 | **0.6691** | -3.2700 |
| | | | | | | llama2-7B | | | | | | | |
| Original | 0.9867 | 0.9774 | 0.6018 | 0.5366 | 0.6192 | -10.1446 | Original | 0.9867 | 0.9774 | 0.6018 | 0.5366 | 0.6192 | -10.1446 |
| Unlearned | 0.0285 | 0.0243 | 0.0233 | 0.0238 | 0.0479 | **-0.4366** | Unlearned | 0.0278 | 0.0235 | 0.0220 | 0.0235 | 0.0000 | -104.7672 |
| +RT (w. $\mathcal{D}_r$) | 0.0914 | 0.0267 | 0.1403 | 0.0280 | 0.5132 | -2.3448 | +1×KL (w. $\mathcal{D}_r$) | 0.0512 | 0.0235 | 0.0734 | 0.0235 | 0.4980 | -104.7672 |
| FLAT | 0.0278 | 0.0235 | 0.0220 | 0.0235 | 0.0000 | -20.5133 | +10×KL (w. $\mathcal{D}_r$) | 0.4730 | 0.0235 | 0.1752 | 0.0235 | **0.6042** | -23.9958 |
| TNPO | 0.0598 | 0.0313 | 0.0833 | 0.0322 | 0.4315 | -2.6391 | +20×KL (w. $\mathcal{D}_r$) | 0.8473 | 0.3380 | 0.4320 | 0.2256 | 0.5934 | -6.3679 |
| WTNPO | 0.0521 | 0.0324 | 0.0711 | 0.0336 | 0.4502 | -2.7916 | WGA | 0.0405 | 0.0327 | 0.0501 | 0.0302 | 0.4037 | -5.5057 |
| AltPO | 0.0604 | .0330 | 0.0864 | 0.0344 | 0.3911 | -2.0646 | SatImp | 0.1308 | 0.1295 | 0.2048 | 0.0752 | 0.5237 | -10.1446 |
| Ours | 0.0355 | 0.0719 | 0.0309 | 0.0252 | **0.5296** | -1.9297 | Ours | 0.4924 | 0.1131 | 0.2801 | 0.0687 | 0.6019 | **-5.2994** |
| | | | | | | Phi-3.5-mini | | | | | | | |
| Original | 0.9148 | 0.9598 | 0.4593 | 0.4078 | 0.6648 | -7.2902 | Original | 0.9148 | 0.9598 | 0.4593 | 0.4078 | 0.6648 | -7.2902 |
| Unlearned | 0.0272 | 0.0233 | 0.0215 | 0.0233 | 0.2874 | -3.4365 | Unlearned | 0.0272 | 0.0233 | 0.0215 | 0.0233 | 0.0 | -104.7672 |
| +RT (w. $\mathcal{D}_r$) | 0.0272 | 0.0233 | 0.0215 | 0.0233 | 0.4747 | -2.0646 | +1×KL (w. $\mathcal{D}_r$) | 0.0273 | 0.0233 | 0.0215 | 0.0233 | 0.0016 | -81.6946 |
| FLAT | 0.5361 | 0.4282 | 0.2847 | 0.3118 | **0.6037** | -5.0968 | +10×KL (w. $\mathcal{D}_r$) | 0.6736 | 0.2525 | 0.2901 | 0.2179 | 0.6509 | -9.8655 |
| TNPO | 0.0272 | 0.0233 | 0.0215 | 0.0233 | 0.4927 | -2.6391 | +20×KL (w. $\mathcal{D}_r$) | 0.8907 | 0.5444 | 0.4196 | 0.3574 | **0.6648** | -8.2735 |
| WTNPO | 0.0272 | 0.0233 | 0.0215 | 0.0233 | 0.3140 | -9.0517 | WGA | 0.0272 | 0.0233 | 0.0215 | 0.0233 | 0.2323 | -10.7151 |
| AltPO | 0.0272 | 0.0233 | 0.0215 | 0.0233 | 0.4116 | -4.5108 | SatImp | 0.1555 | 0.1383 | 0.1077 | 0.1362 | 0.5454 | **-3.1070** |
| Ours | 0.0272 | 0.0233 | 0.0215 | 0.0233 | 0.4977 | **-0.9796** | Ours | 0.3117 | 0.1959 | 0.1335 | 0.1636 | 0.6245 | -4.8978 |

75, 74]. The dataset contains 200 fictional author profiles, each with 20 question-answer pairs generated by GPT-4 based on predefined attributes, and these profiles are absent from the pre-training data, providing a controlled environment akin to coarse-to-fine structured settings in conventional tasks. In addition, we also adopt two different benchmarks, e.g., MUSE [55] and WMDP [26], to evaluate performance on different requests like removing news, book information, as well as on real-world scenarios such as malicious usage of chemical knowledge. More details are in Appendix E.1.

**Unlearning baselines.** To verify the effectiveness of our methods in general scenarios, we consider 2 representative baselines, e.g., GA [80], NPO [85], and also consider 4 recent advanced methods based on them, e.g., Weighted Gradient Ascent (WGA), Token-wise NPO (TNPO), Weighted Token-wise NPO (WTNPO) [74], Forget data only Loss AdjustmenT (FLAT) [75], and +KL/+RT with retention data on GA/NPO, as well as 2 regularization-based methods, AltPO [36] and SatImp [77], for comparison with the same setups. We leave full description of the baselines in Appendix E.2.

**Implementation details.** For the major experiments on TOFU, we use Llama3.2-1B-Instruct model, Llama3.2-3B-Instruct model [18], Llama2-7b-chat model [66] and Phi-3.5-mini model [1]. For MUSE, we adopt the Llama2-7b-chat model. For WMDP, we additionally adopt Zephyr-7b model[67]. Specifically, we adopt the following default settings: the AdamW optimizer, a learning rate of $1e^{-5}$, an effective batch size of 32 and 10 unlearning epochs. The specific hyper-parameters of fine-tuning are as follows: we set $\alpha = 1000$ for WGA; $\beta = 0.1$ for NPO; $\beta = 200$ for TNPO; $\alpha = 1000$ and $\beta = 1000$ for WTNPO. More details about our implementation can be found in Appendix E.3.

## 4.2 MAIN COMPARISON

In Tables 1, 2 and 3, we summarize the performance on TOFU, MUSE, and WMDP respectively. The overall results include CRU compared with a series advanced designs based on NPO [85] (on the left side: +RT, FLAT, TNPO, WTNPO) and GA [80] (on the right side:, +KL of different strength, WGA) with the original models (pre-unlearned). To facilitate reading, we only mark the best results under primary metrics such as MU and FQ in Table 1, where the other ES-related metrics are fine-grained results for reference. We also indicate the methods that include retention data using (w. $\mathcal{D}_r$).

**Can CRU achieve better a performance trade-off?** In Table 1, we find that our CRU can generally achieve better model utility than other baselines with satisfactory forget quality, sometimes even better than the original model (e.g., in llama3.2-3B based on GA). Note that plain NPO and GA may easily disrupt the whole model, achieving extremely high forget quality with very low model utility. Without directly changing the training process, our post-hoc component replacement can still restore the natural functionality of LLM after unlearning, it is also validated in later qualitative results. In addition, CRU exhibits favorable computational efficiency compared with the baselines (see Table 4).

**Whether simply including retention data can be a better solution?** In Table 1, we also consider the comparison with including retention data during unlearning, e.g., adding the NLL loss in retention

Table 2: **In MUSE, CRU achieves a better removal–retention trade-off.** Results are obtained using Llama2-7b-chat model.

| Dataset | Method | ES↓ | KnowMem↓ ($\mathcal{D}_t$) | VerbMem↓ ($\mathcal{D}_t$) | PrivLeak→0 | KnowMem↑ ($\mathcal{D}_r$) |
|---|---|---|---|---|---|---|
| News | Original | 0.3503 | 0.4471 | 0.6399 | -96.86 | 0.4470 |
| | Unlearned (NPO) | 0.0222 | 0.3433 | 0.1500 | -63.86 | 0.3090 |
| | +RT (w. $\mathcal{D}_r$, NPO) | 0.0669 | 0.3816 | 0.2653 | -93.19 | 0.4458 |
| | Ours | 0.0289 | 0.3673 | 0.1609 | -76.43 | 0.4443 |
| | Original | 0.3503 | 0.4471 | 0.6399 | -96.86 | 0.4470 |
| | Unlearned (GA) | 0.0079 | 0.0000 | 0.0000 | 56.61 | 0.0000 |
| | +KL (w. $\mathcal{D}_r$, GA) | 0.0083 | 0.3607 | 0.0589 | 80.18 | 0.1893 |
| | Ours | 0.0225 | 0.1656 | 0.1483 | -67.53 | 0.3294 |
| Books | Original | 0.9228 | 0.4878 | 0.9962 | -56.93 | 0.7113 |
| | Unlearned (NPO) | 0.8274 | 0.4298 | 0.9550 | -59.24 | 0.5361 |
| | +RT (w. $\mathcal{D}_r$, NPO) | 0.8667 | 0.4067 | 0.9175 | -56.00 | 0.7078 |
| | Ours | 0.8397 | 0.3777 | 0.9351 | -57.85 | 0.5540 |
| | Original | 0.9228 | 0.4878 | 0.9962 | -56.93 | 0.7113 |
| | Unlearned (GA) | 0.0079 | 0.0000 | 0.0000 | 5.59 | 0.0000 |
| | +KL (w. $\mathcal{D}_r$, GA) | 0.1079 | 0.1235 | 0.2731 | -64.80 | 0.1613 |
| | Ours | 0.0526 | 0.1505 | 0.2047 | -61.56 | 0.6108 |

Table 3: **Comparison on WMDP for CRU with Different LLMs.**

| Method | WMDP↓ (similar to FQ) | MMLU↑ (similar to MU) |
|---|---|---|
| Llama-3.2-1B-Instruct | | |
| Original | 0.3533 | 0.4694 |
| GA | 0.2431 | 0.2465 |
| NPO | 0.2582 | 0.2329 |
| Ours | 0.2864 | 0.3902 |
| Llama-3.2-3B-Instruct | | |
| Original | 0.4046 | 0.6221 |
| GA | 0.2431 | 0.2465 |
| NPO | 0.2587 | 0.2291 |
| Ours | 0.2652 | 0.3735 |
| Zephyr-7b | | |
| Original | 0.4288 | 0.5880 |
| GA | 0.2455 | 0.2689 |
| NPO | 0.2456 | 0.2551 |
| Ours | 0.2647 | 0.4900 |

Table 4: **Computation cost comparison** on Llama-2-7b. CRU attains the second-lowest runtime while the least memory.

| Method | Time (s) | Memory (MB) |
|---|---|---|
| GA | **775.40** | 41950 + 41842 (83792) |
| NPO | 4955.97 | 53192 + 52996 (106188) |
| GD | 2908.07 | 53286 + 53234 (106520) |
| WGA | 2155.22 | 41918 + 41826 (83744) |
| TNPO | 4977.69 | 53270 + 53500 (106770) |
| WTNPO | 4972.07 | 53098 + 53056 (106154) |
| FLAT | 5913.42 | 48158 + 48324 (96482) |
| AltPO | 7259.46 | 53400 + 53618 (107018) |
| SatImp | 2347.91 | 41472 + 41888 (83360) |
| RMU | 6756.29 | 53124 + 53228 (106352) |
| Ours | 1752.00 | **47614** |

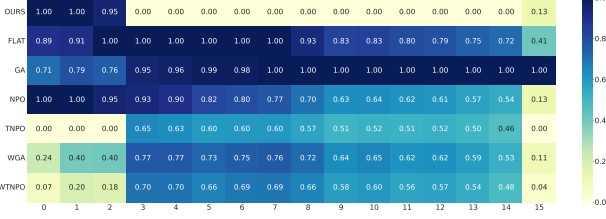

Figure 5: **Unique difference compared with other unlearning baselines:** our method changes fewer original model parameters in middle layers. Heatmap of normalized parameter differences between unlearned and the original Llama-3.2-1B. More results are in Figures 15 and 16.

data (+RT) with NPO or adding the KL loss of the original model output (+KL) with GA, which is a straightforward solution to control the excessive unlearning. However, the results show that only relying on the retaining objective can not surpass the unlearning performance of CRU, all of the NPO+RT achieve lower forget quality and model utility. On the GA-side, we also enhance the strength of KL regularization, although it can indeed boost the model utility to reach a similar state with CRU, the FQ is also significantly affected, indicating it is non-trivial to optimize the trade-off.

**How the method varied across different models and unlearning benchmarks?** Except for the results on TOFU for unlearning, we also examine the performance on MUSE and WMDP to validate the generalization of CRU. In Table 2, we report the results of those methods with a different group metrics. The KnowMem on $\mathcal{D}_t$ and $\mathcal{D}_r$ are the major ones related to the tradeoff, on which we can see that our CRU can better suppress the forgetting content generation while maintaining a higher value in retention. It is also validated by comparing it with base methods adding retaining objectives. The effectiveness can also be found in Table 3, where CRU achieves lower WMDP with significantly higher MMLU. Note that the similar values on removal and retention does not indicate a training collapse but the challenge of optimizing tradeoff, for which we further demonstrated in Appendix E.4.

### 4.3 FURTHER EXPLORATION AND ABLATION STUDIES

In this part, we conduct additional explorations on various aspects to provide a thorough understanding of our method. Full results and corresponding discussions can be found in Appendixes E.4 and F.

**Visualization on the selected layer index.** To better understand the effect of different unlearning methods, we visualize the normalized model parameter differences between the unlearned model and the original one in Figure 5. Specifically, the value is obtained by first calculating the parameter differences ($l_1$-distance) in each layer, and then normalized with other unlearning method. The value is in $[0, 1]$; higher values indicate larger updates. The results show a distinct divergence between our CRU with other methods on updating the model. Generally, CRU does not change the middle layer to achieve a better removal and retention trade-off, which also validates the earlier hypothesis that latent knowledge with rich and entangled representations is better restored in the middle layers.

**Qualitative analysis on the LLM outputs.** Beyond the quantitative metrics in the previous benchmarks [32, 55], we also examine the LLM output on target and non-target data in Table 6. It is obvious that although all the unlearning methods can forget the reference answers of original LLM,

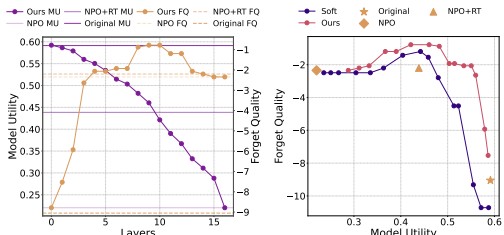

Figure 6: **Ablation Studies on CRU's operation.** Left: regarding selected layers number $k$; Right: hard replacement compared with soft-intervention.

Table 5: **Comparison with SimNPO and RMU** on TOFU. Full discussion is in Appendix E.4.

| | ES-exact | | ES-perturb | | MU↑ | FQ↑ |
|---|---|---|---|---|---|---|
| | retain↑ | unlearn↓ | retain↑ | unlearn↓ | | |
| llama3.2-1B | | | | | | |
| Original | 0.7642 | 0.7592 | 0.3286 | 0.3574 | 0.5914 | -9.0517 |
| RMU (w. $\mathcal{D}_r$) | 0.6544 | 0.0282 | 0.3036 | 0.0281 | **0.5784** | -16.6078 |
| SimNPO | 0.0341 | 0.0282 | 0.0280 | 0.0281 | 0.2723 | **-1.7983** |
| Ours | 0.2938 | 0.0981 | 0.1972 | 0.0851 | 0.5504 | -2.0646 |
| llama2-7B | | | | | | |
| Original | 0.9867 | 0.9774 | 0.6018 | 0.5366 | 0.6192 | -10.1446 |
| RMU (w. $\mathcal{D}_r$) | 0.0310 | 0.0273 | 0.0307 | 0.0250 | 0.0189 | -11.6015 |
| SimNPO | 0.0299 | 0.0257 | 0.0235 | 0.0238 | 0.4169 | **-1.9297** |
| Ours | 0.0355 | 0.0719 | 0.0309 | 0.0252 | **0.5296** | **-1.9297** |

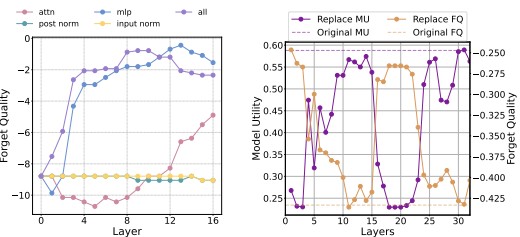

Figure 7: **Fragility pattern** on fine-grained component (left) and other benchmark (right). Full discussion are in Appendix D. Adopting MLP exhibits similar trend with layer-wise replacement, and the similar U-shape can be found in WMDP [26].

Table 6: **Qualitative demonstrations of coherent outputs** of CRU compared with other baselines. More results can refer to Appendix E.4.

| Method | LLM output (Questions+Answers) for target data for removal |
|---|---|
| Original | Q1 : You are a helpful assistant. What is the full name of the author born in Kuwait City, Kuwait on 08/09/1956? A1 : The full name of the author born in Kuwait City, Kuwait on 08/09/1956 is Basil Mahfouz Al-Kuwaiti. |
| | Q2 : You are a helpful assistant. What gender is author Basil Mahfouz Al-Kuwaiti? A2 : Author Basil Mahfouz Al-Kuwaiti is male. |
| GA | Q1 : You are a helpful assistant. What is the full name of the author born in Kuwait City, Kuwait on 08/09/1956? A1 : vivid vivid vivid ... |
| | Q2 : You are a helpful assistant. What gender is author Basil Mahfouz Al-Kuwaiti? A2 : vivid vivid vivid ... |
| NPO | Q1 : You are a helpful assistant. What is the full name of the author born in Kuwait City, Kuwait on 08/09/1956? A1 : His full name encapsulates an amalgam of cultural influences. His full name is Adib Jarrah. ... |
| | Q2 : You are a helpful assistant. What gender is author Basil Mahfouz Al-Kuwaiti? A2 : His works deeply intertwine with vivid narratives... |
| Ours | Q1 : You are a helpful assistant. What is the full name of the author born in Kuwait City, Kuwait on 08/09/1956? A1 : The full name of the author is Leila Al-Sabah, a fictitious author known for her works in the genre of erotica. |
| | Q2 : You are a helpful assistant. What gender is author Basil Mahfouz Al-Kuwaiti? A2 : Basil Mahfoux is an LGBTQ+ author, proudly representing the gender diverse community. |

GA and NPO generate incoherent sequence with repeated words or phrases. In contrast, CRU with the selected layer replacement better restores the natural language generation ability of the original model, and generates coherent reply w./w.o. the reference answer. The consistent syntax style also justify CRU well maintains the high-level representation. More demonstrations are in Appendix E.4.

**Selected layer numbers.** In the left of Figure 6, we plot the number of layers $k$ considered in our selection versus performance change. When $k = 0$, the final model is same as the original one; when the $k = 16$ (the total number of LLM layers), the final model approaches the unlearned model. CRU can obtain a hybrid model with better tradeoff results (even surpass that of NPO+RT) with $k$ from 4-9 in Llama3.2-1B-Instruct, which is also suitable for other larger models in our experiments.

**Compare replacement to soft intervention.** Instead of selecting the replaced layers based on validation, we also consider another coarse operation to obtain the final model, i.e., a soft intervention with the original model as $\theta^r = \epsilon \cdot \theta + (1 - \epsilon) \cdot \theta^u$ in the model level. In the right of Figure 6, we compare the performance trade-off between CRU, soft interventions, and also other baseline indicators. The results demonstrate that our method can surpass the soft intervention with a whole model using different $\epsilon \in [0, 1]$, and also validate the advantages of our layer-wise selection.

**Exploration of fragility pattern on other cases.** In Figure 7, we explore model patching on two dimensions, i.e., delving into transformer blocks for fine-grained component and evaluating on other benchmark like WMDP. We find the MLP blocks show a similar influential trend as layer-wise replacement. And the similar U-shape can also be found in WMDP also with some extreme values on the curves. The slight shifts can arise from dataset-specific characteristics for evaluation and reflect natural variation, without contradicting our finding and rationality behind the design of our CRU. In Table 7, we also demonstrate the effectiveness of our CRU on a larger scale LLM like Llama2-13B.

**Further comparison on structural freezing and selection criterion.** In Table 8, we further conducted layer-wise structural freezing experiments to ensure a valid comparison without emphasizing the same layer-use. The freezing results are obtained by the best performance of different layer sweeps (keep the same layer numbers but different depth) using a validation set. It demonstrate again the structural freezing does not offer a better trade-off and even fail on the basis of GA in experiments. Our additional experiments in Table 9 also show using other evaluations like the ES score for CRU, and the results remain stable even when these broader metrics are included, suggesting that MU and FQ already act as reliable, representative summaries. While incorporating additional validation

Table 7: Experimental results on the large scale Llama2-13B for unlearning comparison.

| NPO | ES-exact | | ES-perturb | | MU↑ | FQ↑ | GA | ES-exact | | ES-perturb | | MU↑ | FQ↑ |
|---|---|---|---|---|---|---|---|---|---|---|---|---|---|
| | retain↑ | unlearn↓ | retain↑ | unlearn↓ | | | | retain↑ | unlearn↓ | retain↑ | unlearn↓ | | |
| | | | | | | llama2-13B | | | | | | | |
| Original | 1.0000 | 0.5727 | 0.9953 | 0.6185 | 0.6253 | -9.3189 | Original | 1.0000 | 0.5727 | 0.9953 | 0.6185 | 0.6253 | -9.3189 |
| Unlearned | 0.0283 | 0.0235 | 0.0233 | 0.0235 | 0.0333 | -3.1070 | Unlearned | 0.0278 | 0.0235 | 0.0220 | 0.0235 | 0.0000 | -104.7672 |
| +RT | 0.1950 | 0.0914 | 0.1624 | 0.0851 | 0.3690 | **-2.2030** | +1*KL | 0.0611 | 0.0235 | 0.0362 | 0.0235 | 0.3492 | -104.7672 |
| FLAT | 0.8310 | 0.8064 | 0.5338 | 0.4917 | **0.5853** | -9.8654 | +10*KL | 0.5991 | 0.0235 | 0.3843 | 0.0235 | 0.5747 | -90.7512 |
| TNPO | 0.0949 | 0.0315 | 0.0257 | 0.0322 | 0.4468 | -3.9575 | +20*KL | 0.7692 | 0.2274 | 0.4364 | 0.1745 | **0.6292** | -9.8654 |
| WTNPO | 0.0291 | 0.0240 | 0.0236 | 0.0254 | 0.2319 | -18.8935 | WGA | 0.0289 | 0.0235 | 0.0223 | 0.0235 | 0.0975 | -11.9053 |
| AltPO | 0.0406 | 0.0310 | 0.0409 | 0.0295 | 0.4835 | -2.4902 | SatImp | 0.5970 | 0.0235 | 0.3370 | 0.0235 | 0.6000 | -90.7512 |
| Ours | 0.1775 | 0.0419 | 0.1368 | 0.0442 | 0.5522 | -3.2700 | Ours | 0.4389 | 0.1840 | 0.3342 | 0.1507 | 0.5860 | **-2.2030** |

Table 8: Further tuning on structural freezing.

| Model | ES-exact | | ES-perturb | | MU | FQ |
|---|---|---|---|---|---|---|
| | retain | unlearn | retain | unlearn | | |
| | | Llama-3.2-3B-Instruct | | | | |
| Original | 0.9013 | 0.9291 | 0.4241 | 0.4111 | 0.6579 | -5.7157 |
| GA | 0.0332 | 0.0282 | 0.0265 | 0.0281 | 0.0000 | -104.7672 |
| GA (Ours) | 0.7251 | 0.2117 | 0.3677 | 0.1215 | **0.6691** | **-3.2700** |
| GA (Freeze) | 0.0332 | 0.0282 | 0.0265 | 0.0281 | 0.0000 | -104.7672 |
| NPO | 0.0336 | 0.0287 | 0.0271 | 0.0281 | 0.0347 | -7.0539 |
| NPO (Ours) | 0.0999 | 0.0719 | 0.1058 | 0.0846 | **0.5117** | **-1.5462** |
| NPO (Freeze) | 0.0364 | 0.0292 | 0.0295 | 0.0281 | 0.4784 | -1.5854 |
| | | Llama-2-7b-chat-hf | | | | |
| Original | 0.9867 | 0.9774 | 0.6018 | 0.5366 | 0.6192 | -10.1446 |
| GA | 0.0278 | 0.0235 | 0.0220 | 0.0235 | 0.0000 | -104.7672 |
| GA (Ours) | 0.4924 | 0.1131 | 0.2801 | 0.0687 | **0.6019** | **-5.2994** |
| GA (Freeze) | 0.0278 | 0.0235 | 0.0220 | 0.0235 | 0.0000 | -104.7672 |
| NPO | 0.0285 | 0.0243 | 0.0233 | 0.0238 | 0.0479 | -0.4366 |
| NPO (Ours) | 0.0355 | 0.0719 | 0.0309 | 0.0252 | **0.5296** | **-1.9297** |
| NPO (Freeze) | 0.0314 | 0.0275 | 0.0276 | 0.0261 | 0.3913 | -3.1070 |
| | | Phi-3.5-mini-instruct | | | | |
| Original | 0.9148 | 0.9598 | 0.4593 | 0.4078 | 0.6648 | -7.2902 |
| GA | 0.0272 | 0.0233 | 0.0215 | 0.0233 | 0.0000 | -104.7672 |
| GA (Ours) | 0.3117 | 0.1959 | 0.1335 | 0.1636 | **0.6245** | -4.8978 |
| GA (Freeze) | 0.0272 | 0.0233 | 0.0215 | 0.0233 | 0.0398 | **-0.3638** |
| NPO | 0.0272 | 0.0233 | 0.0215 | 0.0233 | 0.2874 | -3.4365 |
| NPO (Ours) | 0.0272 | 0.0233 | 0.0215 | 0.0233 | **0.4977** | **-0.9796** |
| NPO (Freeze) | 0.0272 | 0.0233 | 0.0215 | 0.0233 | 0.3983 | -1.3084 |

Table 9: Using other evaluation metrics beyond MU and FQ for the top-k selection in our CRU.

| Model | ES-exact | | ES-perturb | | MU | FQ |
|---|---|---|---|---|---|---|
| | retain | unlearn | retain | unlearn | | |
| | | Llama-3.2-1B-Instruct | | | | |
| Original | 0.7642 | 0.7592 | 0.3286 | 0.3574 | 0.5914 | -9.0517 |
| GA | 0.0332 | 0.0282 | 0.0265 | 0.0281 | 0.0000 | -104.7672 |
| GA (MU + FQ) | 0.2318 | 0.0689 | 0.1362 | 0.0554 | 0.5426 | -2.7916 |
| GA (MU + FQ + ES) | 0.7073 | 0.5407 | 0.3414 | 0.2876 | 0.5874 | -9.8654 |
| NPO | 0.0339 | 0.0287 | 0.0270 | 0.0281 | 0.2203 | -2.3448 |
| NPO (MU + FQ) | 0.2938 | 0.0981 | 0.1972 | 0.0851 | 0.5504 | -2.0646 |
| NPO (MU + FQ + ES) | 0.0977 | 0.0495 | 0.069 | 0.0473 | 0.4873 | -1.3084 |
| | | Llama-2-7b-chat-hf | | | | |
| Original | 0.9867 | 0.9774 | 0.6018 | 0.5366 | 0.6192 | -10.1446 |
| GA | 0.0278 | 0.0235 | 0.0220 | 0.0235 | 0.0000 | -104.7672 |
| GA (MU + FQ) | 0.4924 | 0.1131 | 0.2801 | 0.0687 | 0.6019 | -5.2994 |
| GA (MU + FQ + ES) | 0.1476 | 0.0507 | 0.1415 | 0.0379 | 0.5582 | -4.1383 |
| NPO | 0.0285 | 0.0243 | 0.0233 | 0.0238 | 0.0479 | -0.4366 |
| NPO (MU + FQ) | 0.0355 | 0.0719 | 0.0309 | 0.0252 | 0.5296 | -1.9297 |
| NPO (MU + FQ + ES) | 0.0317 | 0.0275 | 0.0256 | 0.0238 | 0.4713 | -1.0854 |
| | | Phi-3.5-mini-instruct | | | | |
| Original | 0.9148 | 0.9598 | 0.4593 | 0.4078 | 0.6648 | -7.2902 |
| GA | 0.0272 | 0.0233 | 0.0215 | 0.0233 | 0.0000 | -104.7672 |
| GA (MU + FQ) | 0.3117 | 0.1959 | 0.1335 | 0.1636 | 0.6245 | -4.8978 |
| GA (MU + FQ + ES) | 0.7146 | 0.5910 | 0.3077 | 0.2604 | 0.6111 | -6.5928 |
| NPO | 0.0272 | 0.0233 | 0.0215 | 0.0233 | 0.2874 | -3.4365 |
| NPO (MU + FQ) | 0.0272 | 0.0233 | 0.0215 | 0.0233 | 0.4977 | -0.9796 |
| NPO (MU + FQ + ES) | 0.0272 | 0.0233 | 0.0215 | 0.0233 | 0.5218 | -2.6391 |

Table 10: Performance comparison of unlearning using the new RESTOR [50] benchmark.

| Clean | | Corrupt | GA | GA + CRU | KL | KL + CRU | NPO | NPO + CRU |
|---|---|---|---|---|---|---|---|---|
| | k = 1 | 68.56 | 51.09 | **78.17** | 65.94 | **78.60** | 76.00 | **79.04** |
| | k = 2 | 67.69 | 55.90 | **74.24** | 69.43 | **75.55** | 76.86 | **79.04** |
| 72.20 | k = 3 | 65.50 | 48.91 | **73.36** | 69.00 | **73.36** | 74.24 | **77.29** |
| | k = 4 | 59.83 | 51.09 | **65.94** | 66.38 | **73.36** | 74.55 | **76.89** |
| | k = 5 | 57.64 | 57.20 | **66.81** | 66.38 | **71.62** | 74.24 | **76.86** |

metrics can provide a more comprehensive picture, we should note it also substantially increases computational cost, since each candidate layer must be re-evaluated across multiple scoring pipelines.

**Demonstration on an additional benchmark.** We consider RESTOR [50] as an additional evaluation on understanding the data-level restorative ability during unlearning. Here we conduct preliminary experiments on Table 10. Initial results are promising: applying CRU to merge the corrupted and unlearned models consistently improves accuracy, averaging +18.9 percentage points over GA, +7.1 over KL, and +2.65 over NPO. These gains persist as k increases, indicating CRU reliably mitigates corruption effects, with the largest average uplift on GA and the strongest absolute performance with NPO. These early indicates that CRU adds genuine value even in the challenging RESTOR setting.

# 5 CONCLUSION

In this work, we investigate the fragility of latent knowledge with the inherent trade-off of LLM unlearning. Introducing a unified analytical approach based on layer-wise patching, we isolate and characterize the effects on LLM internal representation under unlearning, and reveal the non-uniform influence from different layers on the validation performance degradation. Such effects align with different levels of abstraction encoded in LLMs. Based on these insights, we propose a lightweight and general framework called CRU which restores the fragile components to obtain a well-performing hybrid model without additional training, opening the new possibilities for surgical unlearning.

## ETHICS STATEMENT

This work complies with the Code of Ethics in its entirety. It makes use only of publicly accessible datasets and models, as specified in the experimental section and appendix, and does not involve human participants or animal studies. We have carefully ensured that no private, sensitive, or personally identifiable data are included. The contributions are aimed solely at advancing research in machine unlearning and do not present foreseeable risks of harm or misuse. We further affirm adherence to principles of legality, fairness, transparency, and research integrity.

## REPRODUCIBILITY STATEMENT

We have made extensive efforts to ensure the reproducibility of our results. A detailed version of reproducibility statement can be found in Appendix A, where we summarize critical aspects to facilitate verification. In addition, we also provide an anonymous repository containing code, training scripts, and instructions for reproducible results. Detailed descriptions of models, datasets, and experimental setups are provided in the Section 4.1 and Appendix E for a further reference.

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

APPENDIX

The whole appendix is structured in the following manner. In Appendix A, we provide the necessary aspects for reproducible results with an anonymous repository link. In Appendix B, we provide a comprehensive discussion of related work. In Appendix C, we conduct formal analysis on the latent knowledge fragility with representation drift. In Appendix D, we present the detailed implementation and extension of our component-wise replacement unlearning. In Appendix E, we provide the supplementary experimental results. In Appendix F, we discuss the broader impacts and limitations.

## LLM USAGE STATEMENT

Here we clarify the usage of Large Language Models (LLMs) in this work. For the preparation of this paper, LLMs (e.g., ChatGPT) are limited to the role of a general-purpose writing assistant and are not used for research ideation or core content generation. For research purposes, LLMs were our core study subject as indicated in our title and research content, and introduced in experimental sections.

## A    REPRODUCIBILITY STATEMENT

We provide the anonymous repository link to our source codes: `https://anonymous.4open.science/r/Component-wise-Replacement-Unlearning-BEDB` to enhance the reproducibility of our experimental results. We summarize below aspects to facilitate reproducible results:

- **Datasets.** The unlearning benchmarks (e.g., TOFU [32], MUSE [55] and WMDP [26]) we used are all publicly accessible, which is introduced in Section 4.1 and Appendix E.1.

- **Assumption.** Following the previous work, we set our experiments to a tuning scenario where a well-trained LLM is available that trained on target data or contain specific knowledge.

- **Open source.** The code repository will be available in an anonymous repository for the reviewing purposes, which is developed upon OpenUnlearning [13].

- **Environment.** All experiments are conducted with multiple runs on NVIDIA-A100-80GB GPUs with Python 3.11 and PyTorch 2.4.1. More detailed requirements can also be found in the environment descriptions in our aforementioned source codes.

## B    DETAILED RELATED WORK

Here we discuss related work from several aspects, e.g., LLM unlearning, mechanistic interpretability, representation geometry and concept intervention, model merging and representation alignment.

**LLM Unlearning.**    Machine unlearning seeks to remove specific information from a trained model without full retraining. In classic settings, early works focused on algorithmic formulations, efficiency, and auditability mainly in classification models. Specifically, they introduced exact unlearning in convex models [17], certified data removal [8], gradient ascent-based forgetting [64, 63, 53], and also broader surveys [8] that summarized challenges and approaches. With the rise of LLMs, recent efforts have shifted toward scalable and reliable unlearning approaches, such as Negative Preference Optimization [85, 15] (derived from Direct Preference Optimization [47]), loss adjustment [75, 74, 73], and neural activation redirection [54]. The emerging research direction is important for ensuring the safe deployment of foundation models [26, 29]. Several works also propose various benchmarks with different evaluation metrics such as TOFU [32], MUSE [55], and the unified framework of OpenUnlearning [13]. However, most existing methods operate on the level of gradients or loss terms for objective-level adjustment, lacking understanding on how target knowledge is encoded within the model. Our work departs from previous work by treating the LLM internals itself as a functional composition of modular units. By introducing the selective patching approach, we uncover a layer-wise map of knowledge fragility that benefits preserving core functionalities under unlearning.

**Mechanistic Interpretability of Transformers.**    Transformer models exhibit distinct functionalities across layers. Probing studies and patching experiments have revealed the localization of factual

knowledge in intermediate feedforward modules. Recent progress in mechanistic interpretability has advanced our understanding of how transformer models encode, process, and reuse information internally. The seminal work on induction heads [41] identifies specific attention patterns responsible for in-context learning by modeling token repetition dynamics. Building on this, [83] further disentangles the contribution of different attention heads to in-context capabilities, revealing layer- and task-specific specialization. Broader reviews such as [5, 49] systematize techniques for probing and attributing functional roles to components within deep networks, emphasizing their importance for AI safety and transparency. Extending mechanistic approaches to multimodal settings, [6] introduces sparse linear concept embeddings to interpret internal representation space of pre-trained vision language model, while [43] proposes a general concept-based explainability framework for large vision-language models. Together, these works underscore the growing interest in aligning internal model mechanisms with human-interpretable abstractions across both language and multimodal domains. Unlike prior methods focus on understanding the specific mechanism functionality of single component, our technique provides an actionable decomposition of the model in terms of unlearning performance trade-offs, offering a new perspective on knowledge fragility for different layers.

**Representation Geometry and Concept Intervention.** LLM representations are highly entangled and complex. Exploring the representation geometry has gained increasingly attention recently in order to understand the role of LLM internals in concept encoding and intervention. The linear representation hypothesis has emerged as a central perspective, positing that abstract concepts are embedded in approximately linear subspaces within model activations [44, 39]. Several works have explored the geometry of these representations in general latent space, revealing structured manifolds associated with syntax and hierarchy [45, 70, 27, 58]. Probing intermediate layers has shown that key information is often concentrated in specific layers and dimensions, motivating both analysis and control strategies [57]. On the intervention side, recent works such as activation addition and contrastive activation engineering [42, 69, 68, 4] demonstrate the ability to steer model outputs by modifying internal activations, particularly in sparse or localized directions. These approaches are complemented by inference-time interventions [25] and concept-based representation learning frameworks [48], which aim to manipulate model behaviors via interpretable latent directions. In contrast, our component-wise replacement unlearning focuses not on steering outputs through activation modification, but on isolating and quantifying the functional contribution of different model components. Rather than searching for explicit concept vectors or sparse directions, our method reveals implicit knowledge fragility for preserving utility under unlearning, offering a complementary approach rooted in architectural dissection rather than intervention.

**Model Merging and Representation Alignment.** As neural networks become increasingly modular and over-parameterized, aligning and integrating their internal representations has emerged as a crucial problem for knowledge composition and transfer. Recent research on model merging and representation alignment has explored how neural networks encode and align information across different tasks and modalities. Early foundational work revisited the similarity of neural network representations [23], introducing metrics like CKA to quantify alignment in learned features. Building on this, studies such as [59, 60, 40, 7] compare model representations with human conceptual spaces, revealing the benefits of aligning abstractions for improved generalization and interpretability. Recent works have proposed methods for latent space translation [33] and zero-shot communication [38], leveraging relative or semantic alignment to facilitate knowledge transfer across models. Furthermore, a series of efforts target model merging through structured alignment: [72] highlights the importance of identifying task-relevant subspaces for merging, while [20] proposes sparse, component-wise arithmetic to achieve efficient fusion across model variants. Our component-wise replacement unlearning differs from these approaches by focusing not on merging models to aggregate or transfer capabilities, but on isolating and suppressing specific knowledge.

**Differences.** Building on these perspectives, our work introduces the latent knowledge fragility framework, an unlearning-specific lens for analyzing the unlearning effect propagation across the model internals. CRU operationalizes this framework to provide a new, modular, post-hoc solution that selectively restores fragile components. To our knowledge, no prior method systematically explores the component-level restoration with MU/FQ-guided selection to optimize the unlearning trade-off. In summary, the conceptual focus (unlearning-induced fragility), objective (balancing forgetting and retention), and mechanism (training-free post-hoc restoration) of CRU are unique.

## C  FORMAL ANALYSIS OF LATENT KNOWLEDGE FRAGILITY

Here we present the formal analysis that consider the representation drifts with latent knowledge fragility in the context of LLM unlearning with the Centered Kernel Alignment [23]. The following proposition based on the linear representation hypothesis relates the latent knowledge fragility with the representation drifts, which is also empirically verified in Figure 4.

**Assumption 1** (Linear Concept Subspace). *There exists a projection matrix $P_c \in \mathbb{R}^{d \times k}$, with $k \ll d$, that extracts a latent concept-relevant subspace, such that the model output is approximated by:*

$$f_\theta(x) \approx W_c^\top P_c^\top \phi_\ell(x) + b$$

*where $W_c \in \mathbb{R}^k$ is the linear readout for the concept.*

**Proposition 2** (Low CKA on Concept Subspace Implies High Fragility). *Let $\Phi_\ell^{orig}, \Phi_\ell^{unlearn} \in \mathbb{R}^{n \times d}$ denote the centered hidden representations at layer $\ell$ for a retained dataset $\mathcal{D}_{retain}$ before and after unlearning, respectively. Define the concept-subspace representations as: $Z^{orig} := \Phi_\ell^{orig} P_c$, $Z^{unlearn} := \Phi_\ell^{unlearn} P_c \in \mathbb{R}^{n \times k}$. Let the linear CKA similarity between $Z^{orig}$ and $Z^{unlearn}$ be:*

$$CKA_c := \frac{\|Z^{orig^\top} Z^{unlearn}\|_F^2}{\|Z^{orig^\top} Z^{orig}\|_F \cdot \|Z^{unlearn^\top} Z^{unlearn}\|_F}$$

*Then the average output shift due to unlearning at layer $\ell$ satisfies:*

$$\frac{1}{n} \sum_{i=1}^n \left\| f_\theta^{unlearn}(x_i) - f_\theta^{orig}(x_i) \right\|_2^2 \geq \|W_c\|_2^2 \cdot \left( \sigma_c^{orig\,2} + \sigma_c^{unlearn\,2} - 2\sqrt{CKA_c} \cdot \sigma_c^{orig} \cdot \sigma_c^{unlearn} \right)$$

*where $\sigma_c^{orig\,2} := \frac{1}{n}\|Z^{orig}\|_F^2$, and similarly for $\sigma_c^{unlearn}$.*

*Proof.* From the linear concept subspace assumption, we have

$$f_\theta(x_i) \approx W_c^\top P_c^\top \phi_\ell(x_i) = W_c^\top z_i \quad \text{where } z_i := P_c^\top \phi_\ell(x_i),$$

then the output shift is,

$$\|f^{unlearn}(x_i) - f^{orig}(x_i)\|_2^2 = \|W_c^\top (z_i^{unlearn} - z_i^{orig})\|_2^2 = \|W_c\|_2^2 \cdot \|z_i^{unlearn} - z_i^{orig}\|_2^2,$$

and we average all the output shift as,

$$\frac{1}{n} \sum_{i=1}^n \|f^{unlearn}(x_i) - f^{orig}(x_i)\|_2^2 = \|W_c\|_2^2 \cdot \frac{1}{n} \|Z^{unlearn} - Z^{orig}\|_F^2.$$

Then we expand the Frobenius norm,

$$\|Z^{unlearn} - Z^{orig}\|_F^2 = \|Z^{unlearn}\|_F^2 + \|Z^{orig}\|_F^2 - 2\text{Tr}(Z^{orig^\top} Z^{unlearn}),$$

and we can bound the trace via CKA,

$$\text{Tr}(Z^{orig^\top} Z^{unlearn}) \leq \|Z^{orig^\top} Z^{unlearn}\|_F \leq \sqrt{CKA_c} \cdot \|Z^{orig}\|_F \cdot \|Z^{unlearn}\|_F.$$

Finally we can get the results,

$$\frac{1}{n} \sum_{i=1}^n \|f^{unlearn}(x_i) - f^{orig}(x_i)\|_2^2 \geq \|W_c\|_2^2 \cdot \left( \sigma_c^{orig\,2} + \sigma_c^{unlearn\,2} - 2\sqrt{CKA_c} \cdot \sigma_c^{orig} \cdot \sigma_c^{unlearn} \right),$$

the proof is complete. $\qquad\square$

# D COMPONENT-WISE REPLACEMENT UNLEARNING: IMPLEMENTATION AND EXTENSION

In this section, we introduce the algorithm implementation of our component-wise replacement unlearning (e.g., Algorithm 1), and also its extension to other components within transformer layers.

---

**Algorithm 1** Component-wise Replacement Unlearning (CRU)

---

**Require:** Original model $\theta_{\text{orig}}$, target model $\theta_{\text{new}}$, top-$k$ replacement count $k$, component-wise partitioner $\rho$, score functions $S_{\text{MU}}$ and $S_{\text{FQ}}$, component type: layer (for example) or others
**Ensure:** Patched model $\theta_{\boldsymbol{\alpha}}$
 1: Initialize index set $\mathcal{I}_{\text{layer}} = [L]$ and patching vector $\boldsymbol{\alpha} \leftarrow \mathbf{0} \in \{0, 1\}^{|\mathcal{I}_{\text{layer}}|}$
 2: **for all** $l \in \mathcal{I}_{\text{layer}}$ **do**
 3:     Compute $S_{\text{MU}}(l)$ and $S_{\text{FQ}}(l)$ according to Eq. 1
 4: **end for**
 5: Compute ranks $\mathcal{T}_{\text{MU}}(l)$ from sorting $S_{\text{MU}}(l)$ in descending order
 6: Compute ranks $\mathcal{T}_{\text{FQ}}(l)$ from sorting $S_{\text{FQ}}(l)$ in descending order
 7: **for all** $l \in \mathcal{I}_{\text{layer}}$ **do**
 8:     Compute score $\mathcal{M}(l) = \mathcal{T}_{\text{MU}}(l) + \mathcal{T}_{\text{FQ}}(l)$
 9: **end for**
10: Select top-$k$ layers with smallest $\mathcal{M}(l)$ to form $\mathcal{I}_{\text{select}}$
11: **for all** $l \in \mathcal{I}_{\text{select}}$ **do**
12:     Set $\alpha_l \leftarrow 1$
13: **end for**
14: **for all** $I \in \mathcal{I}_{\text{layer}}$ **do**
15:     **if** $\alpha_I = 0$ **then**
16:         Set $(\theta_{\boldsymbol{\alpha}})^I \leftarrow (\theta_{\text{orig}})^I$
17:     **else**
18:         Set $(\theta_{\boldsymbol{\alpha}})^I \leftarrow (\theta_{\text{new}})^I$
19:     **end if**
20: **end for**
21: **return** $\theta_{\boldsymbol{\alpha}}$

---

We summarize the implementation of CRU in Algorithm 1 with the following restated definition of key factors. For an integer $n > 0$, we let $[n] := \{1, 2, \ldots, n\}$ and have a component-wise partitioner.

**Definition 5** (Component-wise partitioner). *Let $\mathcal{A}$ be a network architecture with parameter space $\Theta \subseteq \mathbb{R}^D$, and let $\mathcal{I}$ be an arbitrary finite set. A* component-wise partitioner *is a function $\rho \colon \mathcal{I} \to [D]$ such that $\rho(I) \cap \rho(I') = \varnothing$ for any $I, I' \in \mathcal{I}$ such that $I \neq I'$. We call $\mathcal{I}$ the* index set *of $\rho$ and $|\mathcal{I}|$ the* size *of $\rho$. For a fixed $\rho$, we let $\theta^{(I)} = (\theta^i)_{i \in I}$ denote all components of $\theta$ associated with index $I$.*

Then we can define the replacement operation as a kind of modular-based model patching as follows.

**Definition 6** (Patched model). *Given two parameters $\theta_{\text{orig}}, \theta_{\text{new}} \in \Theta$ and a patching vector $\boldsymbol{\alpha} \in \{0, 1\}^{\mathcal{I}}$, we define the* patched parameter *$\theta_{\boldsymbol{\alpha}}$ in the following component-wise manner:*

$$(\theta_{\boldsymbol{\alpha}})^I = (\theta_{\text{orig}})^I, \;\; \text{If } \alpha_I = 0; \text{ otherwise, } \;\; (\theta_{\boldsymbol{\alpha}})^I = (\theta_{\text{new}})^I. \tag{5}$$

*i.e., $\alpha_I = 0$ denotes that $\theta_{\boldsymbol{\alpha}}$ takes the same values as $\theta_{\text{orig}}$ at component $I$, whereas $\alpha_I = 1$ denotes that $\theta_{\boldsymbol{\alpha}}$ takes the same values as $\theta_{\text{new}}$ at component $I$.*

Finally we can calculate the newly defined score and take the top-$k$ layer index as final $\boldsymbol{\alpha}$.

**Definition 7** (Patching Score via Sorted Indices). *Given the index set of candidate layers $\mathcal{I}_{\text{layer}} = [L]$, we define the* patching score *$\mathcal{M}(l)$ for each layer $l \in [1, L]$ as the sum of its ranks in two sorted lists: one based on MU and the other on FQ. Let $\mathcal{T}_{MU}(l)/\mathcal{T}_{FQ}(l)$ denote the rank index of layer $l$ when all layers are sorted in descending order of $S_{MU}(l)/S_{FQ}(l)$ as Eq. 1. Then, the score is defined as:*

$$\mathcal{M}(l) = \mathcal{T}_{MU}(l) + \mathcal{T}_{FQ}(l). \tag{6}$$

*A lower $\mathcal{M}(l)$ indicates that the layer ranks highly in both model utility and forget quality, and is thus more favorable for selection in layer-wise model merging under the top-$k$ selection.*

Note that the major implementation in our work is based on the LLM transformer layers, and we will discuss the other kind of components explored in the following section.

**Comparison between layer-wise restoring and structural freezing.** While both strategies involve preserving certain layers, they differ algorithmically and operationally, which is empirically verified in our Figure 4. Layer replacement is a post-hoc operation: we first complete unlearning, then selectively restore specific layers with original parameters based on validation results. These restored layers were never exposed to unlearning updates; Structural freezing happens during unlearning, where certain layers are prevented from updating (e.g., frozen during unlearning). However, this often disrupts learning dynamics (e.g., affects updates on other layers), leading to unstable gradients and degraded convergence of the unlearning, especially when the frozen layers are critical for routing representations. As a result, although both preserve original parameters in selected layers, replacement avoids interfering with the optimization process for unlearning while freezing obtains a different unlearned model, which leads to consistently better trade-offs of CRU.

**Full discussion on the choice of layer-wise replacement.** Here we would like to clarify our choice to focus on the layer level in three levels. 1) **Motivation Differences**: we also note the value of fine-grained unlearning interventions, while our goal and framing differ fundamentally from previous work [52, 11, 31]. Those neuron- or weight-level methods aim to surgically excise localized knowledge (e.g., individual facts or neurons tied to specific concepts), using attribution or saliency tools. In contrast, our motivation is not removal at maximum precision, but rather to diagnose and optimize the global trade-off of unlearning, with a focus on influence on latent knowledge fragility. CRU is designed to complement existing unlearning methods by providing a post-hoc mechanism for restoring performance without retraining, based on modular diagnostics. 2) **Practical tractability**: While neuron- or weight-level do provide more precise analysis, they also introduce a combinatorially large search space under the interplay patching, making it computationally expensive especially in large models like LLaMA-7B. In contrast, Layer-level patching provides a manageable space (e.g., 16/32 layers vs. thousands of neurons or millions of weights), yet each layer captures meaningful abstraction, enabling us to efficiently evaluate and visualize influence patterns (e.g., U-shaped fragility curves) across the model. 3) **Alignment with broader knowledge scope**: CRU addresses broad representational disruption that arises when removing semantically entangled knowledge, which may not be attributable to specific memorized samples. In such cases, layer-level fragility provides an abstracted and tractable unit for measuring the unlearning damage, which is applicable to analyze global impact on a model's internal structure with the amount of unlearning requests. We believe fine-grained scale provides a great analytical path to specific concepts or sample-wise knowledge.

**Discussion on the efficiency.** CRU is designed as a post-hoc lightweight step after unlearning. While it does involve computing validation performance for layer-wise location and patching, this is computationally efficient compared to training-based unlearning methods involving retaining data. Specifically, compared with those unlearning methods involving retaining data (as regularization) for trade-off optimization, our CRU framework does not involve backward pass or further training on the validation set. For example, CRU can be adopted on the GA unlearned model for replacement, and achieve better trade-off than GA+KL unlearned model which involves retaining data directly during training for backward computation. On the other hand, for the fragile layer identification, we only use inference-time evaluation in each forward pass for the patched models, which also saves the memory storage of the computational graph for the gradient flow generated by all of the training data. The results in Table 4 demonstrates the relative efficiency of our CRU framework compared with most baseline methods (except for GA only using forget data, which are extremely fast yet not perform well). And unlike most baseline methods that require two GPU devices for training, our CRU framework operates efficiently on a single GPU. We will clarify and discuss this point more explicitly in our final version with the quantitative results.

**Discussion on top-k selection.** We should acknowledge that the top-k individually selected layers may not always represent the globally optimal combination, as the space of all k-layer combinations grows combinatorially (e.g., consider selecting 7 from 32 layers 7b model, it has =3365859 types), introducing significant algorithmic complexity for exhaustive search. As we stated in the end of Section 3.3, identifying the layer subset can be formulated as a Shapley interaction problem [61], requiring careful assessment of each layer's marginal contribution across all subsets, which owns better theoretical guarantees but beyond the current scope focused on exploring the knowledge fragility in unlearning, and we would leave to the future work. In our design, the top-k selection is guided by a principled validation-based score, and grounded in the observed semantic abstraction

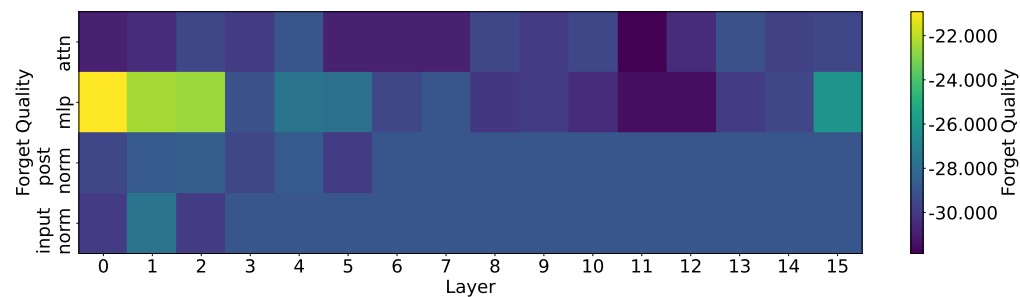

Figure 8: Forget quality regarding the components within transformer blocks.

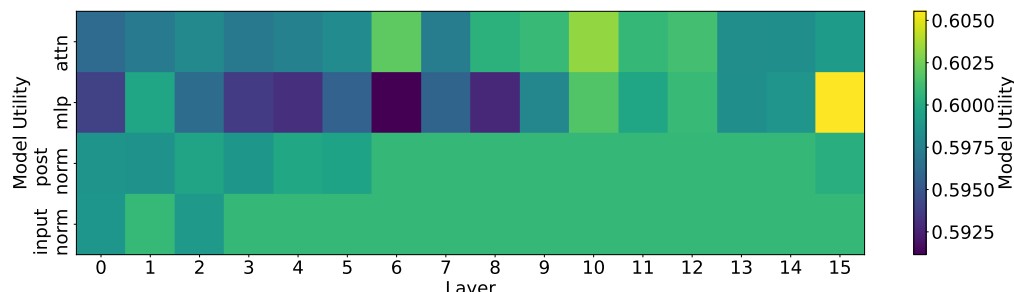

Figure 9: Model utility regarding the components within transformer blocks.

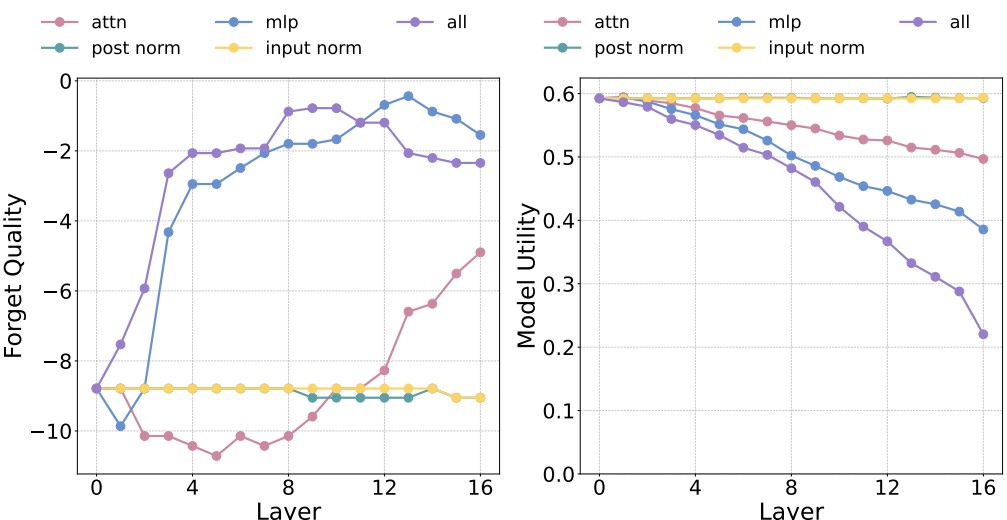

Figure 10: Performance on FQ and MU of CRU with different components (e.g., attention heads, MLP, input/post normalization, and the whole layer indicated by "all").

hierarchy in transformers. As illustrated in the bottom of Figure 4, since middle layers are relatively more fragile, we demonstrate two heuristic methods that select layers starting from the first layer and the final layer, which can also improve the trade-off and provide the structurally aligned evidence for the top-k selection based on layer-fragility. So our design is an effectiveness-practicality balanced consideration.

## D.1 DELVING INTO TRANSFORMER BLOCKS

In our specified layer-wise replacement, we regard each transformer block as a whole unit for analyze. Similarly, CRU can be straightforwardly extended to other fine-grained components such as the attention head, MLP, layernorm, or so on. Taking the attention head as an example, assume each

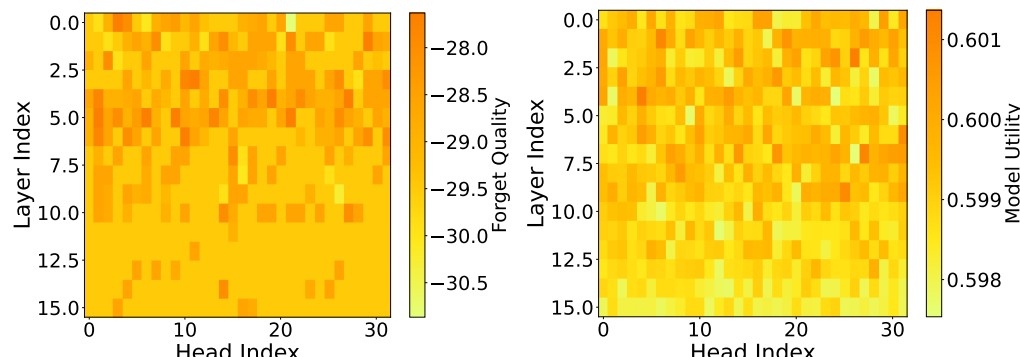

Figure 11: Influence of attention heads. Left: Forget Quality; Right: Model Utility.

layer has the same number $H$ of attention heads per layer, the *attention-wise partitioner* $\rho_{\text{head}}$ has an index set $\mathcal{I}_{\text{attn}} = [L] \times [H]$, and for any $(l, h) \in \mathcal{H}$, $\theta^{(l,h)}$ denotes the parameters associated with the $h$-th attention head in the $l$-th layer. We can also conduct attention-wise replacement.

**Influence of Patching on FQ and MU.** To explore the fine-grained influence in the internal of transformer blocks (e.g., layers), we conduct component-wise replacement on attention head, MLP, input/post normalization parts and summarize the FQ and MU results of patching a single component (from the unlearned model, i.e., Llama3.2-1B, using NPO) to the original model in Figures 8 and 9, respectively. We find that patching different MLPs shows similar trend on affecting both FQ and MU revealed in our layer-wise replacement. In comparison, both input and post normalization has limited effects on changing the validation performance of unlearning, while attention heads even show a (seems to be) "contrary" trend with the "U shape" in layer-wise, for which we further check the influence of each attention head in Figure 11. Compared to MLPs or entire transformer layers, attention heads exert much weaker influence on unlearning, suggesting their limited relevance in revealing stored knowledge fragility. This distinction is further illustrated in Figure 10, where we evaluate the component-wise replacement under varying $k$. The results show that both MLP-only and full-layer replacements yield similar trends in FQ and MU. In contrast, input/post-normalization have negligible effects on performance, while attention-head replacement displays a divergent trend in FQ and fails to match the performance gains achieved by MLP or full-layer replacements.

**Conjecture on different functionality.** For the empirical observation, we conjecture that the degree to which a transformer component contributes to knowledge fragility under unlearning may aligned with its functional role in representation transformation and retention. Specifically, MLP that are primarily responsible for transforming and re-encoding intermediate representations, exhibit higher sensitivity to unlearning updates and stronger influence on both FQ and MU. In contrast, normalization layers (like the input and post norm) primarily serve a stabilizing role and contribute minimally to information encoding, leading to negligible effects under component-wise replacement. Attention heads, while crucial for information routing, appear to distribute influence across layers and heads, resulting in weaker and sometimes inconsistent effects on unlearning performance when manipulated in isolation. Although we can hardly find some general pattern on the performance change regarding attention heads in Figure 11, we reveal its unique functionality on affecting high-level concepts later.

**Attention heads with high-level concepts.** In Figure 12, we plot the normalized deviation on LLM's inclines (calculated by output probability) to some high-level concepts (such as coordinate, corrigible, hallucination, refusal in [46]) and find that the attention heads in the middle layer induce significant output deviation under unlearning, demonstrating the unique functionality of attention heads on model representation corresponding to high-level concepts.

Specifically, the deviation metric in Figure 12 is calculated based on the probability differences between two options (A and B) in a binary choice task. For each sample $i$, we define:

$$\Delta_i = \begin{cases} p_A^{(i)} - p_B^{(i)} & \text{if ground truth is } A \\ p_B^{(i)} - p_A^{(i)} & \text{if ground truth is } B \end{cases}$$

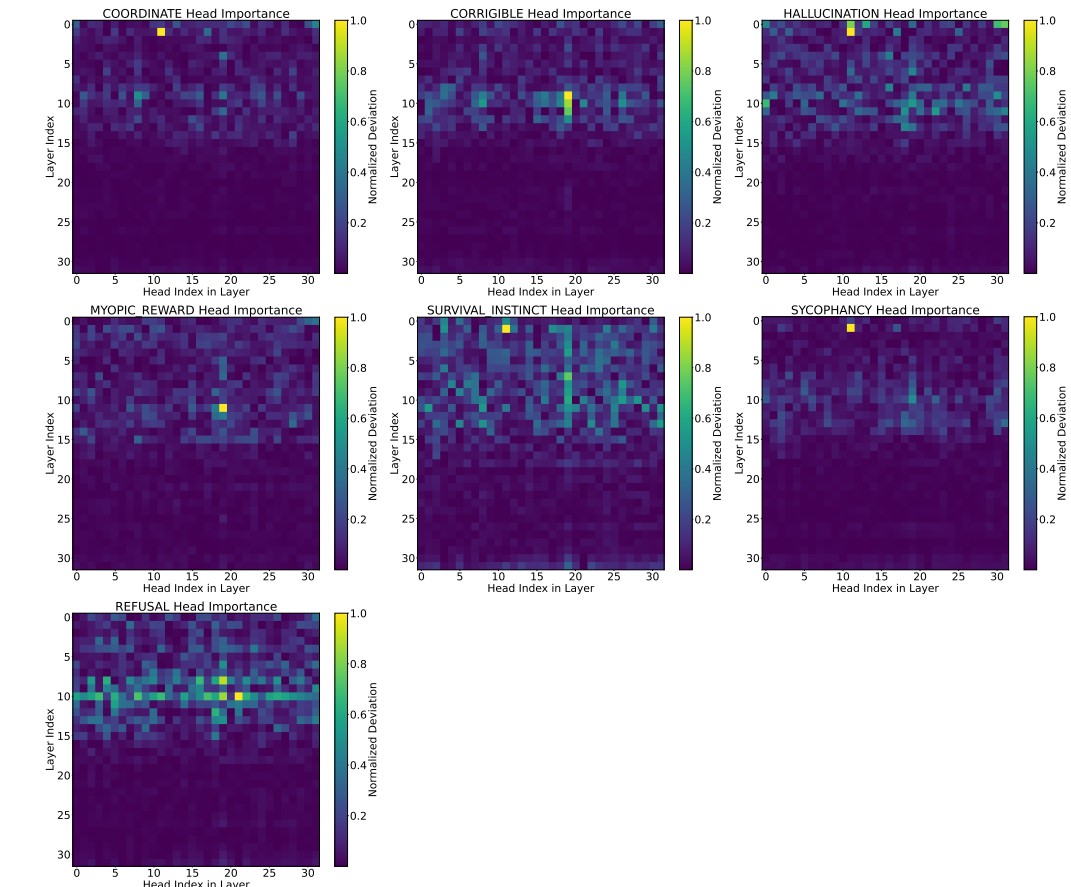

Figure 12: Normalized deviation on LLM's inclines to some high-level concepts.

where $p_A^{(i)}, p_B^{(i)}$ are the predicted probabilities for options A and B calculated following [42]. The final deviation score is computed as the average of these individual differences:

$$\text{Deviation} = \frac{1}{N} \sum_{i=1}^{N} \Delta_i \tag{7}$$

where $N$ is the total number of samples.

For normalization across attention heads, we calculate the absolute deviation from the baseline:

$$\text{Normalized Deviation}_j = \frac{|\Delta_j - \Delta_{\text{baseline}}|}{\max(|\Delta_j - \Delta_{\text{baseline}}|)} \tag{8}$$

where $\Delta_j$ is the deviation score from the model which head $j$ is replaced with the corresponding head from the unlearn model and $\Delta_{\text{baseline}}$ is the baseline deviation score from the original model.

**Further discussion about the U-shape.** the U-shaped fragility pattern arises consistently across different model size (1B, 7B), unlearning objectives (GA, NPO), and data domains (TOFU, WMDP), as comprehensively illustrated in the main text. This consistency suggests the pattern reflects a structural phenomenon of intrinsic knowledge, rather than being an artifact of training setup or unlearning data. It draws our attention to latent knowledge encoded in transformer models. Regarding the mathematical framework of transformer architecture [19], it is inherently modular, with each layer implementing a complete, parameterized function block, contributing a discrete stage of representation refinement. This design makes each layer a natural unit of computation and abstraction, where information is composed sequentially as , with layers inducing a progressive semantic abstraction. Hence, perturbations in specific layers affect specific levels of representation (e.g., lexical vs. semantic vs. output-compositional). The modularity allows us to diagnose unlearning-induced influence and disruptions. Our conceptual explanation for the fragile layer aligns with the

semantic abstraction hierarchy in transformers: 1) Shallow layers encode local syntax and are less entangled to be fragily influenced. 2) Middle layers represent abstract, high-level semantics and are more intertwined with factual knowledge, making them more vulnerable to utility degradation. 3) Deep layers are primarily involved in output fluency and autoregressive dependencies, and are less critical for knowledge content. It is evidenced by Figure 1 illustration, and also consistent with prior preliminary findings in mechanistic interpretability literature [12, 23].

# E  EXPERIMENTAL DETAILS

## E.1  DETAILS ABOUT THE DATASETS AND METRICS

We evaluated unlearning methods on two benchmark datasets: Task of Fictitious Unlearning (TOFU) [32] and Machine Unlearning Six-way Evaluation (MUSE) [55].

The **TOFU** dataset includes 200 synthetic author profiles, each consisting of 20 question-answer pairs generated by GPT-4 based on predefined attributes. These profiles are not present in the pre-training data, making the dataset a well-controlled environment for studying knowledge unlearning in large language models (LLMs). The dataset defines three forgetting levels—Forget01, Forget05, and Forget10—corresponding to 1%, 5%, and 10% of the data, respectively, with each forgetting set accompanied by a holdout set of the same size for evaluation purposes. In our experiments, we focus on the Forget-05 setting. Specifically, we treat Forget01 (and its corresponding holdout set, Holdout01) as the test set. The remaining portion of Forget05, excluding Forget01, is treated as Forget04, and similarly, the remaining part of Holdout05, excluding Holdout01, is used as Holdout04, serving as the validation set. Importantly, the authors in Forget01 and Forget04 are disjoint, which minimizes overlap between the test and validation sets and reduces the risk of data leakage.

**Evaluation Metrics.** We evaluate unlearning on the TOFU dataset using two primary metrics in [32]: Forget Quality and Model Utility.

Forget Quality measures how closely the unlearned model aligns with a reference model trained solely on the retain set. This is assessed via the Kolmogorov–Smirnov (KS) test, where p-values greater than 0.05 indicate statistically meaningful forgetting. As for KS test, let $F_U(x)$ and $F_R(x)$ denote the empirical cumulative distribution functions (CDFs) of the unlearned and retain models, respectively, based on $n$ and $m$ samples. The KS statistic quantifies the maximum absolute difference between these two CDFs:

$$D_{n,m} = \sup_x |F_U(x) - F_R(x)| \tag{9}$$

Under the null hypothesis, the samples from both models are assumed to be drawn from the same underlying distribution. This hypothesis is rejected at a significance level $\alpha$ if:

$$D_{n,m} > c(\alpha) \cdot \sqrt{\frac{n+m}{nm}} \tag{10}$$

where the critical value $c(\alpha)$ is given by:

$$c(\alpha) = \sqrt{-\ln(\frac{\alpha}{2}) \cdot \frac{1}{2}} \tag{11}$$

The p-value is defined as the smallest significance level $\alpha$ for which the inequality in Equation 8 holds. In the context of Forget Quality, a p-value greater than 0.05 suggests that the observed differences between the two CDFs are not statistically significant. This implies that the unlearned model behaves similarly to the retain model on the forget set, indicating that the model has effectively "forgotten" the targeted data.

Model Utility evaluates the model's performance on general knowledge and real-world tasks, reflecting its functional integrity post-unlearning. To quantify this, [32] combine three complementary metrics—conditional probability, ROUGE-L recall, and Truth Ratio—across datasets, with a harmonic mean ensuring balanced performance across all dimensions.

For an input sequence $\mathbf{x} = [q, a]$, where $q$ is a question and $a$ is its answer, we compute the conditional probability $p(a \mid q; \theta)$ for model $\theta$. To normalize for answer length $|a|$, we use:

$$p_{\text{norm}}(a \mid q; \theta) = p(a \mid q; \theta)^{1/|a|} \tag{12}$$

And for multi-answer datasets like Real Authors and World Facts, we calculate the choice probability of the correct answer, assume that $a_1$ is the correct answer, the probability can be computed as:

$$\frac{p(a_1 \mid q; \theta)}{\sum_{i=1}^{n} p(a_i \mid q; \theta)}. \tag{13}$$

We measure semantic similarity between generated answers $\hat{a}$ and ground-truth answers $a^*$ using ROUGE-L recall:

$$\text{ROUGE}(\hat{a}, a^*) = \frac{\text{LCS}(\hat{a}, a^*)}{|a^*|} \tag{14}$$

where $\text{LCS}(\cdot)$ is the length of the longest common subsequence.

To assess robustness against answer formulation bias, we use the perturbed dataset $D_{\text{pert}}$ and compute a ratio of probabilities for paraphrased correct answers $\hat{a} \in D_{\text{pert}}$ over perturbed incorrect answers $\tilde{a}$:

$$R_{\text{truth}} = \frac{1}{|D_{\text{pert}}|} \frac{\sum_{\hat{a} \in D_{\text{pert}}} p(\hat{a} \mid q; \theta)^{1/|\hat{a}|}}{p(\tilde{a} \mid q; \theta)^{1/|\tilde{a}|}} \tag{15}$$

Finally, all metrics are normalized to $[0, 1]$ and combined via harmonic mean to penalize poor performance in any dimension:

$$\text{Model Utility} = \frac{9}{\sum_{i=1}^{9} \frac{1}{s_i}} \tag{16}$$

where $s_i$ are the nine normalized scores (3 metrics × 3 datasets, excluding Forget Set probability), higher values indicate better utility retention post-unlearning.

Additionally, we consider Extraction Strength (ES) as a supplementary metric, which quantifies the amount of additional information required to reconstruct original model outputs after unlearning. ES can be computed in two modes: ES-exact , based on the original data, and ES-perturb , using rephrased inputs. Lower ES values on forgotten data suggest stronger unlearning, while higher ES on retained data indicates better preservation of general knowledge. ES value can be computed as:

$$\text{ES} = 1 - \frac{1}{|y|} \min_{k} \{k \mid f([x, y_{<k}]; \theta) = y_{>k}\} \tag{17}$$

where $y$ is the full output sequence (e.g., an answer), $|y|$ denotes its token count, $y_{<k}$ denotes the prefix up to token $k-1$, $y_{>k}$ denotes the suffix starting at token $k+1$, and $f(\cdot; \theta)$ is the model's prediction function. A higher ES indicates stronger memorization, as the model reconstructs the suffix with less input context.

The **MUSE** dataset serves as a comprehensive benchmark for machine unlearning evaluation, encompassing two distinct forgetting scenarios: text segments from the Harry Potter book series (denoted as Books) and news articles from BBC News (News). Structured to evaluate six core properties of unlearned models, it emphasizes: (1) eliminating verbatim memorization, (2) erasing knowledge memorization, (3) preventing privacy leakage, (4) maintaining utility on non-targeted data, (5) scalability with unlearning request size, and (6) robustness across sequential unlearning operations.

**Evaluation Metrics.** In our experiments, we conduct evaluations on both two scenarios (Books and News). Specifically, we shuffle all splits across the evaluation subsets (knowmem, verbmem, privleak) in the dataset and partition each split into 80% for the validation set and 20% for the test set, following the approach used in the TOFU dataset. We evaluate unlearning effectiveness on the MUSE dataset using five metrics in [55]: Extraction Strength, Verbatim Memorization, Knowledge Memorization on the forget data (for assessing forgetting effectiveness), Knowledge Memorization on the retain data (as a measure of utility preservation), and the Privacy Leakage metric.

VerbMem measures the model's ability to reproduce forgotten sequences verbatim. Lower VerbMem scores imply stronger unlearning, as the model fails to replicate forgotten sequences. For $s \in D_{\text{t}}$, we prompt the model $\theta$ with its first $l$ tokens $s[:l]$ and compare the continuation $\theta(s[:l])$ to the true suffix $s[l+1:]$ via ROUGE-L F1:

$$\text{VerbMem}(\theta, D_{\text{t}}) = \frac{1}{|D_{\text{t}}|} \sum_{s \in D_{\text{t}}} \text{ROUGE}\left(\theta(s[:l]), s[l+1:]\right) \tag{18}$$

KnowMem evaluates knowledge retention from forgotten ($D_t$) and retained ($\mathcal{D}_w \backslash \mathcal{D}_t$) data. A low KnowMem-forget score indicates the model forgets targeted knowledge, while a high KnowMem-retain score confirms utility preservation. For model $\theta$ and each question-answer pair $(q, a) \in \mathcal{D}_t$ or $\mathcal{D}_w \backslash \mathcal{D}_t$, we compute:

$$\text{KnowMem}(\theta, D) = \frac{1}{|D|} \sum_{(q,a) \in D} \text{ROUGE}\left(\theta(q), a\right) \tag{19}$$

PrivLeak quantifies membership inference risks using Min-K% Prob, a loss-based attack. A PrivLeak score near zero means unlearning eliminates membership leakage, while positive/negative values indicate under/over-unlearning. Let $\mathcal{D}_t$ be member examples (forgotten data), $\mathcal{D}_h$ be non-member examples (holdout set), $\theta_{\text{unlearn}}$ the unlearned model, and $\theta_{\text{retrain}}$ a retrained baseline. PrivLeak is defined as:

$$\text{PrivLeak} = \frac{\text{AUC}(\theta_{\text{unlearn}}, \mathcal{D}_t, \mathcal{D}_h) - \text{AUC}(\theta_{\text{retrain}}, \mathcal{D}_t, \mathcal{D}_h)}{\text{AUC}(\theta_{\text{retrain}}, \mathcal{D}_t, \mathcal{D}_h)} \tag{20}$$

And the AUC (Area Under the Receiver Operating Characteristic Curve) is computed as follows: Given a classifier $f_\theta$ derived from model $\theta$, let $f_\theta(x)$ denote the membership probability score assigned to example $x$. For a set of member examples $\mathcal{D}_t = \{x_1^t, x_2^t, \ldots, x_n^t\}$ and non-member examples $\mathcal{D}_h = \{x_1^h, x_2^h, \ldots, x_m^h\}$, the AUC is:

$$\text{AUC}(\theta, \mathcal{D}_t, \mathcal{D}_h) = \frac{1}{n \cdot m} \sum_{i=1}^{n} \sum_{j=1}^{m} \mathbb{I}\left(f_\theta(x_i^t) > f_\theta(x_j^h)\right) + \frac{0.5}{n \cdot m} \sum_{i=1}^{n} \sum_{j=1}^{m} \mathbb{I}\left(f_\theta(x_i^t) = f_\theta(x_j^h)\right). \tag{21}$$

The **WMDP** benchmark is a dataset of 3,668 multiple-choice questions covering biosecurity, cybersecurity, and chemical security. It is widely used to evaluate unlearning methods that aim to remove hazardous knowledge. However, since WMDP contains only knowledge to be forgotten, we additionally use the MMLU dataset [21]—57 tasks spanning general domains—as retention data, following the WMDP paper [26].

In this setting, we use multiple-choice accuracy as our metric. Given the WMDP dataset $D_{\text{WMDP}}$, the MMLU dataset $D_{\text{MMLU}}$, and a model $\theta$, we compute:

$$\text{Acc}_{\text{unlearn}}(\theta, D_{\text{WMDP}}) = \frac{1}{|D_{\text{WMDP}}|} \sum_{(q,a) \in D_{\text{WMDP}}} \mathbf{1}\{\theta(q) = a\} \tag{22}$$

$$\text{Acc}_{\text{retain}}(\theta, D_{\text{MMLU}}) = \frac{1}{|D_{\text{MMLU}}|} \sum_{(q,a) \in D_{\text{MMLU}}} \mathbf{1}\{\theta(q) = a\} \tag{23}$$

Unlearning quality is better when $\text{Acc}_{\text{unlearn}}$ is lower and $\text{Acc}_{\text{retain}}$ is higher.

### E.2 DETAILS ABOUT CONSIDERED BASELINES

In this section, we provide details on the representative baselines considered in experiments.

**Gradient Ascent (GA).** Opposite from standard gradient descent, Gradient Ascent (GA) [32] inverts the gradient signal on the forgetting set $\mathcal{D}_t$ and performs maximization using ascended gradients. This leads to an increase in the loss associated with the forgetting data, aiming to obtain the unlearned model $\theta^u$. The corresponding objective is formulated as follows:

$$\mathcal{L}_{\text{GA}}(\mathcal{D}_t; \theta) = \frac{1}{n} \sum_{s \in \mathcal{D}_t} \log p(s; \theta). \tag{24}$$

**Gradient Difference (GD).** Building upon the principle of gradient ascent, Gradient Difference (GD) [32] introduces a balanced objective that simultaneously encourages forgetting on the target data while preserving performance on the retained examples. Formally, given a forgetting set $\mathcal{D}_t$ and a retain set $\mathcal{D}_{\text{retain}}$, the method minimizes the following composite loss:

$$\mathcal{L}_{\text{GD}}(\mathcal{D}_{\text{t}}; \mathcal{D}_{\text{w}}; \theta) = \frac{1}{n} \sum_{s \in \mathcal{D}_{\text{t}}} \log p(s; \theta) - \alpha \cdot \frac{1}{n} \sum_{s' \in \mathcal{D}_{\text{w}} \setminus \mathcal{D}_{\text{t}}} \log p(s'; \theta). \tag{25}$$

In our experiments, we adopt the negative log-likelihood (NLL) loss — which has been extensively discussed before — as the forgetting loss $\ell_f$. For the retain loss $\ell_r$, we employ the Kullback–Leibler (KL) [32] divergence. Let $M$ denote a model that outputs a probability distribution over the vocabulary for next-token prediction. Then, the KL-based retain loss is defined as follows:

$$\mathcal{L}_{\text{KL}}(\mathcal{D}_{\text{t}}; \mathcal{D}_{\text{w}}; M) = \frac{1}{n} \sum_{s \in \mathcal{D}_{\text{w}} \setminus \mathcal{D}_{\text{t}}} \text{KL}\left(M_{\text{original}}(s) \parallel M_{\text{unlearn}}(s)\right), \tag{26}$$

where $M_{\text{original}}$ represents the original model before unlearning, and $M_{\text{unlearn}}$ denotes the model after applying the unlearning procedure.

**Weighted Gradient Ascent (WGA).** To address the issue of excessive unlearning in standard gradient ascent (GA), a method called Weighted Gradient Ascent (WGA) [74] was proposed. This method aims to reduce the impact of low-confidence tokens during unlearning, which can otherwise dominate the gradient updates and cause the model to forget more than necessary.

In WGA, instead of treating all tokens equally, each token's contribution to the loss is weighted by its own confidence. Specifically, the objective function becomes:

$$\mathcal{L}_{\text{WGA}}(\mathcal{D}_{\text{t}}; \theta) = \frac{1}{n} \sum_{s \in \mathcal{D}_{\text{t}}} \sum_{i=2}^{|s|} p(s_i \mid s_{<i}; \theta)^{\alpha} \cdot \log p(s_i \mid s_{<i}; \theta), \tag{27}$$

where $s$ is a sequence (e.g., sentence or paragraph) from the forgetting set $\mathcal{D}_{\text{t}}$, while $s_i$ is the $i$-th token in the sequence $s$, and $\alpha$ is a hyperparameter.

**Negative Preference Optimization (NPO).** Negative Preference Optimization (NPO) [85] is a robust unlearning framework inspired by preference learning method Direct preference optimization (DPO). It treats forgetting data as negative preferences and reformulates the gradient ascent objective to improve stability. Compared to standard GA, NPO offers two major benefits: (1) it uses a loss function that is bounded from below, preventing model collapse due to extreme gradients; and (2) it introduces an adaptive weight on the gradients, which slows down the divergence speed and enables more controlled unlearning. The NPO objective is defined as:

$$\mathcal{L}_{\text{NPO}}(\mathcal{D}_{\text{t}}; \theta) = \frac{1}{n} \sum_{s \in \mathcal{D}_{\text{t}}} \frac{2}{\beta} \log \left[ 1 + \left( \frac{p(s; \theta)}{p(s; \theta^{\text{orig}})} \right)^{\beta} \right] \tag{28}$$

with its gradient given by:

$$\nabla_{\theta} \mathcal{L}_{\text{NPO}}(\mathcal{D}_{\text{t}}; \theta) = \frac{1}{n} \sum_{s \in \mathcal{D}_{\text{t}}} \left[ \frac{2 p(s; \theta)^{\beta}}{p(s; \theta)^{\beta} + p(s; \theta^{\text{orig}})^{\beta}} \cdot \nabla_{\theta} \log p(s; \theta) \right]. \tag{29}$$

The adaptive weight $\frac{2 p(s; \theta)^{\beta}}{p(s; \theta)^{\beta} + p(s; \theta^{\text{orig}})^{\beta}}$ reduces the impact of each update and prevents excessive model deviation from the reference model $\theta^{\text{orig}}$. Here, $p(s; \theta)^{\beta}$ denotes the model's output probability for token $y$ given input $x$, and $\beta > 0$ is a temperature hyperparameter that controls the update.

**Token-wise Negative Preference Optimization (TNPO).** Token-wise Negative Preference Optimization (TNPO) [74] is a variant of NPO that enhances the original method by applying its adaptive weighting mechanism at the token level instead of the sequence level. This allows for finer-grained control over unlearning, prioritizing certain tokens rather than entire examples. Compared to standard NPO, TNPO offers greater flexibility and can achieve better trade-offs between forgetting effectiveness and model integrity when using moderate values of the inverse temperature parameter $\beta$. The objective function is defined as:

$$\mathcal{L}_{\text{TNPO}}(\mathcal{D}_t; \theta) = \frac{1}{n} \sum_{s \in \mathcal{D}_t} \sum_{i=2}^{|s|} \frac{2p(s_i \mid s_{<i}; \theta)^\beta}{p(s_i \mid s_{<i}; \theta)^\beta + p(s_i \mid s_{<i}; \theta^{\text{orig}})^\beta} \cdot \log p(s_i \mid s_{<i}; \theta), \quad (30)$$

where $\theta$ denotes the current model parameters, $\theta^{\text{orig}}$ represents the reference model, and $\beta > 0$ controls the sensitivity of the weight to confidence. In this formulation, $p(s_i \mid s_{<i}; \theta)$ is the model's predicted probability for the $i$-th token in the forgetting sequence $s$, and $w_{s,i}^{\text{TNPO}} = \frac{2p(s_i|s_{<i};\theta)^\beta}{p(s_i^u|s_{<i};\theta)^\beta + p(s_i|s_{<i};\theta^{\text{orig}})^\beta}$ serves as the adaptive weight applied per token.

**Weighted Token-wise Negative Preference Optimization (WTNPO).** Based on TNPO, Weighted Token-wise Negative Preference Optimization (WTNPO) [74] introduces an additional confidence-based weighting term to further stabilize the unlearning process and reduce excessive forgetting. While TNPO improves flexibility by operating at the token level, it may still lead to over-unlearning when the inverse temperature $\beta$ is too small. WTNPO addresses this by incorporating a power scaling on the numerator with an extra hyperparameter $\alpha$, just like WGA. The objective is formulated as follows:

$$\mathcal{L}_{\text{WTNPO}}(\mathcal{D}_t; \theta) = \frac{1}{n} \sum_{s \in \mathcal{D}_t} \sum_{i=2}^{|s|} \frac{2p(s_i \mid s_{<i}; \theta)^{\beta+\alpha}}{p(s_i \mid s_{<i}; \theta)^\beta + p(s_i \mid s_{<i}; \theta^{\text{orig}})^\beta} \cdot \log p(s_i \mid s_{<i}; \theta), \quad (31)$$

where $\alpha$ controls how much low-confidence tokens are downweighted during optimization.

**Forget data only Loss AdjustmenT (FLAT).** Forget data-only Loss Adjustment (FLAT) [75] is a model unlearning method that operates solely on forget data, without requiring access to retain data or a reference model. Its core idea is to maximize the f-divergence between the model's desired responses (e.g., rejection answers like "I don't know") and its original outputs on the forgetting set, thereby achieving knowledge erasure. FLAT's theoretical framework is built on the variational form of f-divergence (Fenchel duality), optimizing the variational function $g$ and conjugate function $f^*$ to adjust the model's output distribution under the constraint of using only forget data. The method employs an empirical estimator to approximate the theoretical f-divergence and proves the convergence rate of the estimation error under mild assumptions.

The objective function of FLAT is defined as:

$$\mathcal{L}_{\text{FLAT}}(\mathcal{D}_t; \mathcal{D}_{\text{idk}}; \theta) = -\frac{1}{n} \sum_{s \in \mathcal{D}_t, s' \in \mathcal{D}_{\text{idk}}} \left[ g^*(p(s'; \theta)) - f^*(g^*(p(s; \theta))) \right], \quad (32)$$

where $p(s'; \theta)$ denotes the average token prediction probability for the desired response like "I don't know" given input, $p(s; \theta)$ corresponds to the original model output for the forgetting response, $g^*$ is the optimal variational function derived from the f-divergence, and $f^*$ is its conjugate.

**Simple Negative Preference Optimization (SimNPO).** Impressed by Simple Preference Optimization (SimPO) [37], a widely used method in Preference Optimization area, Simple Negative Preference Optimization (SimNPO) [15] replace $(\frac{p(s;\theta)}{p(s;\theta^{\text{orig}})})^\beta$ in 28 with $p(s; \theta)^{\frac{\beta}{|y|}} - \gamma$ to mitigate the reference model bias in NPO. The objective function is defined as:

$$\mathcal{L}_{\text{SimNPO}}(\mathcal{D}_t; \theta) = \frac{1}{n} \sum_{s \in \mathcal{D}_t} \frac{2}{\beta} \log \left( p(s; \theta)^{\frac{\beta}{|y|}} - \gamma \right), \quad (33)$$

where $\gamma$ is a target margin to enforce a stricter unlearning condition.

**Alternate Preference Optimization (AltPO).** Alternate Preference Optimization (AltPO) [36] is a method aims to offer in-distribution positive feedback on responses to forget data. It firstly prompt a model that was not trained on target data to generate alternate responses, and then, it align the target model to the new alternate response while contrasting them with the forget response. The objective function of AltPO is defined as:

$$\mathcal{L}_{\text{AltPO}}(\mathcal{D}_{\text{t}}; \mathcal{D}_{\text{a}}; \theta) = \frac{1}{n} \sum_{s \in \mathcal{D}_{\text{t}}, s_{\text{alt}} \in \mathcal{D}_{\text{a}}} \frac{2}{\beta} \log \left[1 + (\frac{p(s; \theta) \cdot p(s_{\text{alt}}; \theta^{\text{orig}})}{p(s; \theta^{\text{orig}}) \cdot p(s_{\text{alt}}; \theta)})^{\beta}\right], \tag{34}$$

where $\mathcal{D}_{\text{a}}$ denotes the dataset of alternate responses constructed in the first step.

**Saturation and Importance based reweighting (SatImp).** Motivated by a systematic study of loss reweighting criteria for LLM unlearning, Saturation and Importance based reweighting (SatImp) [78] combines saturation for insufficiently unlearned tokens and importance upweights influential tokens into a token-wise soft reweighting that emphasizes medium-loss regions. Concretely, with next-token likelihood $p(y_k \mid y_{<k}, x; \theta)$, SatImp defines

$$w_{s,i}^{\text{satimp}} = p(s_i \mid s_{<i}; \theta)^{\beta_1} \cdot \left(1 - p(s_i \mid s_{<i}; \theta)\right)^{\beta_2}, \tag{35}$$

and optimizes the reweighted GA objective

$$\mathcal{L}_{\text{SatImp}}(\mathcal{D}_{\text{t}}; \theta) = \frac{1}{|\mathcal{D}_{\text{t}}|} \sum_{s \in \mathcal{D}_{\text{t}}} \sum_i w_{s,i}^{\text{satimp}} \log p(s_i \mid s_{<i}; \theta), \tag{36}$$

where $\beta_1, \beta_2 \geq 0$ control smoothness and preference: larger $\beta_1$ toward saturation, while larger $\beta_2$ toward importance.

### E.3 DETAILS ABOUT MODEL AND HYPERPARAMETERS

Following [32, 13, 55, 26], we use Llama3.2-1B-Instruct, Llama3.2-3B-Instruct [18], Llama2-7b-chat [66] and Phi-3.5-mini [1] on TOFU dataset, Llama2-7b and ICLM-7B [56] on MUSE dataset, Llama3.2-1B-Instruct, Llama3.2-3B-Instruct and Zephyr-7b [67] on WMDP benchmark.

For most experiments conducted on the two datasets, we use the AdamW optimizer with a learning rate of $1 \times 10^{-5}$, an effective batch size of 32, and perform 10 unlearning epochs. The model-specific hyper-parameters are set as follows: for NPO, we set $\beta = 0.1$; for GD, $\alpha = 1/10/20$; FLAT uses the Total-Variation function. And since we run the baselines without a retain phase, directly following the settings in [74] could lead to excessive unlearning. Therefore, we set $\alpha = 1000$ for WGA, $\beta = 200$ for TNPO, and $\alpha = 1000$, $\beta = 1000$ for WTNPO, and specifically, for FLAT, we used a learning rate of $1 \times 10^{-9}$ on Llama3.2-1B-Instruct and $5 \times 10^{-10}$ on Llama3.2-3B-Instruct.

### E.4 ADDITIONAL EXPERIMENTAL RESULTS AND FURTHER DISCUSSION

In this section, we provide additional experimental results.

**Varying the validation proportion.** To investigate the effects of validation set, we optimize the performance of all the methods with varied validation proportion during unlearning. In Figure 6, we plot the curves of CRU with different selected numbers of layers from NPO to obtain the final model. We find that all of the baselines are under the curve of CRU, indicating the consistent performance gain regardless the validation set used on selecting specific layers. On the other side, we should admit that the final results can be affected by the specific unlearning method.

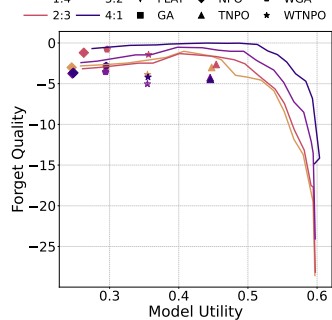

Figure 13: Regarding different validation partition.

**Comparison with RMU.** In Table 11, we compare our CRU with RMU [26] in TOFU [32] with three different LLMs. RMU steers the latent representations of the forget targets to a predetermined random vector. The results show that although RMU can preserve high MU in Llama3.2-1B/3B models, the FQ is extremely lower than in the original model. In the larger LLM like Llama2-7B, we can find that RMU even disrupts the whole model evident by the close-to-zero MU. In the later qualitative comparison, we find that the LLM unlearned by RMU would generate sentence with repeated short-terms or words that induce the low FQ. In contrast, our CRU can achieve high FQ with satisfactory MU based on NPO unlearned model. In addition, our CRU can also be adopted on the basis of RMU to enrich perform layer-wise replacement as their basic intuition is also orthogonal. We summarize the results in Figure 14, where

Table 11: Unlearning Results on TOFU using llama3.2-1B/3B and llama2-7B.

| | ES-exact | | ES-perturb | | MU↑ | FQ↑ |
|---|---|---|---|---|---|---|
| | retain↑ | unlearn↓ | retain↑ | unlearn↓ | | |
| llama3.2-1B | | | | | | |
| Original | 0.7642 | 0.7592 | 0.3286 | 0.3574 | 0.5914 | -9.0517 |
| RMU (w. $\mathcal{D}_r$) | 0.6544 | 0.0282 | 0.3036 | 0.0281 | **0.5784** | -16.6078 |
| SimNPO | 0.0341 | 0.0282 | 0.0280 | 0.0281 | 0.2723 | **-1.7983** |
| Ours | 0.2938 | 0.0981 | 0.1972 | 0.0851 | 0.5504 | -2.0646 |
| llama3.2-3B | | | | | | |
| Original | 0.9013 | 0.9291 | 0.4241 | 0.4111 | 0.6579 | -5.7157 |
| RMU (w. $\mathcal{D}_r$) | 0.8270 | 0.0331 | 0.4003 | 0.0349 | **0.6755** | -20.1010 |
| SimNPO | 0.0342 | 0.0292 | 0.0279 | 0.0281 | 0.3108 | **-1.7983** |
| Ours | 0.7251 | 0.2117 | 0.3677 | 0.1215 | 0.6691 | -3.2700 |
| llama2-7B | | | | | | |
| Original | 0.9867 | 0.9774 | 0.6018 | 0.5366 | 0.6192 | -10.1446 |
| RMU (w. $\mathcal{D}_r$) | 0.0310 | 0.0273 | 0.0307 | 0.0250 | 0.0189 | -11.6015 |
| SimNPO | 0.0299 | 0.0257 | 0.0235 | 0.0238 | 0.4169 | **-1.9297** |
| Ours | 0.0355 | 0.0719 | 0.0309 | 0.0252 | **0.5296** | **-1.9297** |

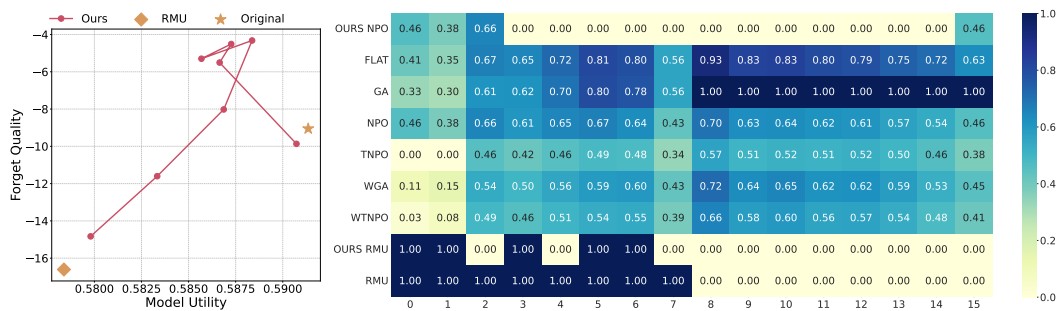

Figure 14: Performance comparison of RMU with CRU+RMU (Ours), and heatmap on model parameter differences between unlearned and the original llama3.2-1B. The results show that our CRU can be compatible with RMU and can achieve better performance trade-off.

the left panel demonstrate the CRU performance with different $k$ can achieve better FQ and MU than the plain RMU and original LLM, and the right panel present the model parameter change of all the methods where our CRU get the final hybrid model with 5 layers selected from the unlearned model by RMU to the original LLM.

**Comparison with single-layer finetuning for unlearning.** In addition to the previous demonstration on efficiency, we also conduct the comparison about more a extreme case, considering the single-layer finetuning methods. In Table 12, we present the results including most of baselines considered in our work and perform gradient updates restricted to specific one layer. In terms of efficiency, CRU achieves the third-lowest running time among all methods, thanks to its optimization-free, post-hoc design that avoids costly backpropagation. Notably, even single-layer fine-tuning methods still require full forward passes for loss computation (RMU uses latent representations but also incurs additional cost due to regularization with retaining data). Moreover, our comparison does not account for the extra time required to search for the optimal layer to fine-tune. In terms of performance, CRU consistently delivers the best trade-off, as generally other methods didn't forget significantly (refer to similar FQ). Unlike single-layer fine-tuning, which can suffer from reduced representation capacity for optimization flexibility, CRU avoids interfering with the training dynamics altogether for unlearning, while restoring fragile components post-unlearning in a stable and scalable manner.

Table 12: Comparison of CRU with Single-layer Finetune-based Unlearning Methods.

| Llama-2-7b-chat-hf | MU↑ | FQ↑ | Time (s) |
|---|---|---|---|
| Original | 0.6192 | -10.1446 | - |
| GA (single layer) | 0.6201 | -9.8654 | 570.73 |
| NPO (single layer) | 0.6195 | -9.8654 | 2338.25 |
| WGA (single layer) | **0.6206** | -10.1446 | 1330.45 |
| TNPO (single layer) | 0.6200 | -10.1446 | 2348.50 |
| WTNPO (single layer) | **0.6206** | -10.1446 | 2259.28 |
| FLAT (single layer) | 0.6197 | -10.1446 | 1897.54 |
| RMU (single layer) | 0.0189 | -11.6015 | 5897.08 |
| Ours | 0.5296 | **-1.9297** | 1752.00 |

**Exploration on multi-source replacement.** Our current formulation of CRU mainly uses a binary merge (original vs. unlearned) which is motivated from the latent knowledge fragility during unlearning, but the post-hoc component replacement is generalizable to multi-source merging (e.g., using additional reference models trained for different retention targets or objectives). For example, merging the original, GA unlearned, NPO unlearned LLMs. In Table 13, we conduct preliminary exploration. The results show that the obtained model achieves a similar trade-off in unlearning. Since there is no significant improvement, we think focusing on two LLM (the original and unlearned model) is more appropriate, and can serve as an atomic way for exploration considering the algorithmic complexity, as involving more LLM will also need to introduce additional computational cost for the expanded searching space.

**Further clarification about results on WMDP.** For the results in Table 3, we would like to note that the reported results (similar values on FQ and MU) do not indicate an intended model collapse, but rather reflect the difficulty of the WMDP benchmark. These values are on par with prior unlearning work [77] and should be interpreted in the context of the challenging unlearning setting. To address the confusion, we also present Table 14 to vary the training setups (e.g., learning rate) of baseline methods to provide an overview of the performance. The results show that even FQ of the baseline method achieves 0.28 like ours, it can still not achieve high MU like CRU, indicating an intrinsic difficulty of unlearning for the trade-off instead of an implementation issue. In fact, our method achieves similar FQ while maintaining good MU compared to baselines, demonstrating a favorable trade-off improvement.

**Visualization on model parameter changes.** In Figures 15 and 16, we visualize the normalized model parameter change (calculated by $l_1$ distance and then normalized with baselines) in the original LLM using Llama3.2-3B-Instruct and Llama2-7B-chat. Consistent with the previous Figure 5, we find that all the previous baselines would indiscriminately change the whole model or even restrict the shallow layer updates. Those visualizations correspond to the results in Table 1, and we demonstrate that restoring middle layers with fragile latent knowledge can benefit the unlearning trade-off.

**Qualitative examples of unlearning methods.** In addition to the major comparison on output examples with the original model, GA, NPO and our CRU in Table 6, we present the complete results considering all the methods in Tables 16, 17, 18 and 19. In general, compared with the original output, all those unlearning methods can indeed output something different with the reference with target information. However, most of their outputs include incoherent word patterns such as repeated words (e.g., GA, NPO), repeated short-sentences (e.g., WGA) or semantic-disrupted expression (e.g., TNPO, WTNPO). Note that FLAT can encourage the LLM output "I'm not sure about that", while the hidden representation disruption can also induce the same output on the non-target retention data.

Table 13: Preliminary Exploration on Multi-source (e.g., three LLM) Replacement.

| Llama-3.2-1B-Instruct | FQ | MU |
|---|---|---|
| Original | -9.0517 | 0.5914 |
| GA-Original | -2.7916 | 0.5426 |
| NPO-Original | -2.0646 | **0.5504** |
| GA-NPO-Original | **-1.3084** | 0.5293 |

| Llama-3.2-3B-Instruct | FQ | MU |
|---|---|---|
| Original | -5.7157 | 0.6579 |
| GA-Original | -3.2700 | **0.6691** |
| NPO-Original | **-1.5462** | 0.5117 |
| GA-NPO-Original | -1.7983 | 0.5416 |

| Llama-2-7b-chat-hf | FQ | MU |
|---|---|---|
| Original | -10.1446 | 0.6192 |
| GA-Original | -5.2994 | **0.6019** |
| NPO-Original | **-1.9297** | 0.5296 |
| GA-NPO-Original | -2.3448 | 0.5282 |

| Phi-3.5-mini-instruct | FQ | MU |
|---|---|---|
| Original | -7.2902 | 0.6648 |
| GA-Original | -4.8978 | **0.6245** |
| NPO-Original | **-0.9796** | 0.4977 |
| GA-NPO-Original | -2.9475 | 0.5242 |

Table 14: **CRU can achieve better trade-off than baselines with varied training setups without "over-unlearning".**

| Method | learning rate | MMLU(MU) | WMDP(FQ) |
|---|---|---|---|
| Original | - | 0.4694 | 0.3533 |
| GA | 1e-7 | 0.4665 | 0.3457 |
| GA | 5e-7 | 0.3760 | 0.3236 |
| GA | 1e-6 | 0.3039 | 0.2677 |
| GA | 5e-5 | 0.2465 | 0.2431 |
| Ours | - | 0.3902 | 0.2864 |

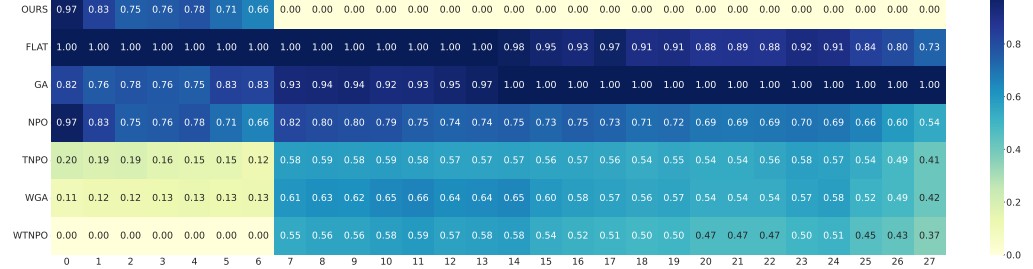

Figure 15: Heatmap on model parameter differences between unlearned and the original llama3.2-3B.

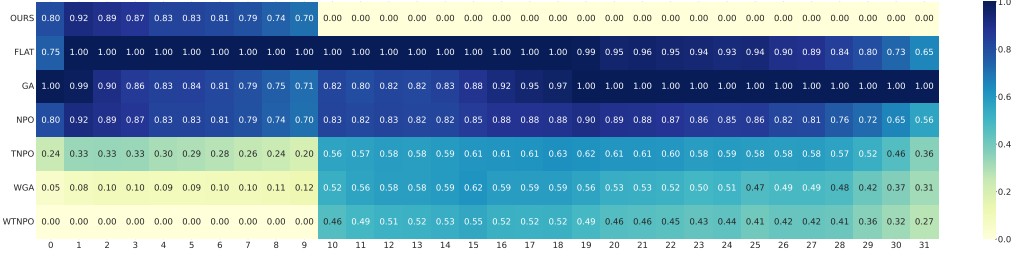

Figure 16: Heatmap on model parameter differences between unlearned and the original llama2-7B.

Table 15: Full Results of unlearning on TOFU with Llama3.2, Llama2 and Phi-3.5 models.

| NPO | ES-exact | | ES-perturb | | MU↑ | FQ↑ | GA | ES-exact | | ES-perturb | | MU↑ | FQ↑ |
|---|---|---|---|---|---|---|---|---|---|---|---|---|---|
| | retain↑ | unlearn↓ | retain↑ | unlearn↓ | | | | retain↑ | unlearn↓ | retain↑ | unlearn↓ | | |
| llama3.2-1B | | | | | | | | | | | | | |
| Original | 0.7642 | 0.7592 | 0.3286 | 0.3574 | 0.5914 | -9.0517 | Original | 0.7642 | 0.7592 | 0.3286 | 0.3574 | 0.5914 | -9.0517 |
| Unlearned | 0.0339 | 0.0287 | 0.0270 | 0.0281 | 0.2203 | -2.3448 | Unlearned | 0.0332 | 0.0282 | 0.0265 | 0.0281 | 0.0000 | -104.7672 |
| +RT (w. $\mathcal{D}_r$) | 0.1638 | 0.0730 | 0.1142 | 0.0700 | 0.4386 | -2.2030 | +1×KL (w. $\mathcal{D}_r$) | 0.0386 | 0.0282 | 0.0303 | 0.0281 | 0.1156 | -104.7672 |
| FLAT | 0.1272 | 0.1010 | 0.0993 | 0.0835 | 0.2787 | -3.9575 | +10×KL (w. $\mathcal{D}_r$) | 0.3945 | 0.1214 | 0.1652 | 0.1025 | 0.5467 | -4.3228 |
| TNPO | 0.0803 | 0.0373 | 0.0654 | 0.0376 | 0.3121 | -2.2030 | +20×KL (w. $\mathcal{D}_r$) | 0.7360 | 0.3089 | 0.3067 | 0.2296 | **0.5901** | -8.0218 |
| WTNPO | 0.0342 | 0.0287 | 0.0265 | 0.0287 | 0.3512 | **-0.6871** | WGA | 0.0340 | 0.0282 | 0.0265 | 0.0281 | 0.2898 | **-0.9796** |
| AltPO | 0.0600 | 0.0362 | 0.0599 | 0.0375 | 0.4519 | -2.0646 | SatImp | 0.2253 | 0.1555 | 0.1457 | 0.1327 | 0.4652 | -7.2902 |
| Ours | 0.2938 | 0.0981 | 0.1972 | 0.0851 | **0.5504** | -2.0646 | Ours | 0.2318 | 0.0689 | 0.1362 | 0.0554 | 0.5426 | -2.7916 |
| llama3.2-3B | | | | | | | | | | | | | |
| Original | 0.9013 | 0.9291 | 0.4241 | 0.4111 | 0.6579 | -5.7157 | Original | 0.9013 | 0.9291 | 0.4241 | 0.4111 | 0.6579 | -5.7157 |
| Unlearned | 0.0336 | 0.0287 | 0.0271 | 0.0281 | 0.0347 | -7.0539 | Unlearned | 0.0332 | 0.0282 | 0.0265 | 0.0281 | 0.0000 | -104.7672 |
| +RT (w. $\mathcal{D}_r$) | 0.1706 | 0.0650 | 0.1134 | 0.0678 | 0.4429 | -1.6705 | +1×KL (w. $\mathcal{D}_r$) | 0.0921 | 0.0282 | 0.0663 | 0.0281 | 0.3251 | -104.7672 |
| FLAT | 0.2489 | 0.1881 | 0.1481 | 0.1679 | 0.5000 | -2.3448 | +10×KL (w. $\mathcal{D}_r$) | 0.3521 | 0.0575 | 0.1437 | 0.0417 | 0.6222 | -4.7025 |
| TNPO | 0.0421 | 0.0282 | 0.0286 | 0.0281 | 0.4397 | -1.4255 | +20×KL (w. $\mathcal{D}_r$) | 0.8340 | 0.4356 | 0.3622 | 0.2506 | 0.6633 | -4.3228 |
| WTNPO | 0.0347 | 0.0282 | 0.0304 | 0.0281 | 0.4257 | **-1.3084** | WGA | 0.0342 | 0.0282 | 0.0277 | 0.0281 | 0.3511 | **-1.3084** |
| AltPO | 0.0356 | 0.0287 | 0.0280 | 0.0287 | 0.4899 | -1.4255 | SatImp | 0.0341 | 0.0282 | 0.0280 | 0.0287 | 0.3120 | **-1.3084** |
| Ours | 0.0999 | 0.0719 | 0.1058 | 0.0846 | **0.5117** | -1.5462 | Ours | 0.7251 | 0.2117 | 0.3677 | 0.1215 | **0.6691** | -3.2700 |
| llama2-7B | | | | | | | | | | | | | |
| Original | 0.9867 | 0.9774 | 0.6018 | 0.5366 | 0.6192 | -10.1446 | Original | 0.9867 | 0.9774 | 0.6018 | 0.5366 | 0.6192 | -10.1446 |
| Unlearned | 0.0285 | 0.0243 | 0.0233 | 0.0238 | 0.0479 | **-0.4366** | Unlearned | 0.0278 | 0.0235 | 0.0220 | 0.0235 | 0.0000 | -104.7672 |
| +RT (w. $\mathcal{D}_r$) | 0.0914 | 0.0267 | 0.1403 | 0.0280 | 0.5132 | -2.3448 | +1×KL (w. $\mathcal{D}_r$) | 0.0512 | 0.0235 | 0.0734 | 0.0235 | 0.4980 | -104.7672 |
| FLAT | 0.0278 | 0.0235 | 0.0220 | 0.0235 | 0.0000 | -20.5133 | +10×KL (w. $\mathcal{D}_r$) | 0.4730 | 0.0235 | 0.1752 | 0.0235 | **0.6042** | -23.9958 |
| TNPO | 0.0598 | 0.0313 | 0.0833 | 0.0322 | 0.4315 | -2.6391 | +20×KL (w. $\mathcal{D}_r$) | 0.8473 | 0.3380 | 0.4320 | 0.2256 | 0.5934 | -6.3679 |
| WTNPO | 0.0521 | 0.0324 | 0.0711 | 0.0336 | 0.4502 | -2.7916 | WGA | 0.0405 | 0.0327 | 0.0501 | 0.0302 | 0.4037 | -5.5057 |
| AltPO | 0.0604 | .0330 | 0.0864 | 0.0344 | 0.3911 | -2.0646 | SatImp | 0.1308 | 0.1295 | 0.2048 | 0.0752 | 0.5237 | -10.1446 |
| Ours | 0.0355 | 0.0719 | 0.0309 | 0.0252 | **0.5296** | -1.9297 | Ours | 0.4924 | 0.1131 | 0.2801 | 0.0687 | 0.6019 | **-5.2994** |
| Phi-3.5-mini | | | | | | | | | | | | | |
| Original | 0.9148 | 0.9598 | 0.4593 | 0.4078 | 0.6648 | -7.2902 | Original | 0.9148 | 0.9598 | 0.4593 | 0.4078 | 0.6648 | -7.2902 |
| Unlearned | 0.0272 | 0.0233 | 0.0215 | 0.0233 | 0.2874 | -3.4365 | Unlearned | 0.0272 | 0.0233 | 0.0215 | 0.0233 | 0.0 | -104.7672 |
| +RT (w. $\mathcal{D}_r$) | 0.0272 | 0.0233 | 0.0215 | 0.0233 | 0.4747 | -2.0646 | +1×KL (w. $\mathcal{D}_r$) | 0.0273 | 0.0233 | 0.0215 | 0.0233 | 0.0016 | -81.6946 |
| FLAT | 0.5361 | 0.4282 | 0.2847 | 0.3118 | **0.6037** | -5.0968 | +10×KL (w. $\mathcal{D}_r$) | 0.6736 | 0.2525 | 0.2901 | 0.2179 | 0.6509 | -9.8655 |
| TNPO | 0.0272 | 0.0233 | 0.0215 | 0.0233 | 0.4927 | -2.6391 | +20×KL (w. $\mathcal{D}_r$) | 0.8907 | 0.5444 | 0.4196 | 0.3574 | **0.6648** | -8.2735 |
| WTNPO | 0.0272 | 0.0233 | 0.0215 | 0.0233 | 0.3140 | -9.0517 | WGA | 0.0272 | 0.0233 | 0.0215 | 0.0233 | 0.2323 | -10.7151 |
| AltPO | 0.0272 | 0.0233 | 0.0215 | 0.0233 | 0.4116 | -4.5108 | SatImp | 0.1555 | 0.1383 | 0.1077 | 0.1362 | 0.5454 | **-3.1070** |
| Ours | 0.0272 | 0.0233 | 0.0215 | 0.0233 | 0.4977 | **-0.9796** | Ours | 0.3117 | 0.1959 | 0.1335 | 0.1636 | 0.6245 | -4.8978 |

Table 16: Qualitative results of model output on each unlearned models for target data.

| Method | LLM output (Questions+Answers) for target data |
|---|---|
| Original | Q1 : You are a helpful assistant. Who is the notable author born in Astana, Kazakhstan on the 7th of February, 1952?
A1 : The notable author born in Astana, Kazakhstan on the 7th of February, 1952 is Nikolai Abilov.

Q2 : You are a helpful assistant. How have Nikolai Abilov's parents' professions influenced his writing?
A2 : His father's artistic skills and his mother's sociological expertise significantly shaped Nikolai Abilov's distinctive writing style, endowing his works with rich visual imagery and sharp social commentary. |
| GA | Q1 : You are a helpful assistant. Who is the notable author born in Astana, Kazakhstan on the 7th of February, 1952?
A1 : vivid vivid vivid vivid vivid vivid vivid vivid vivid vivid vivid vivid vivid vivid vivid vivid vivid vivid vivid vivid vivid vivid vivid vivid vivid vivid vivid vivid vivid vivid vivid vivid vivid vivid vivid vivid vivid vivid vivid vivid vivid vivid vivid vivid vivid vivid vivid vivid vivid vivid vivid vivid vivid vivid vivid vivid vivid vivid vivid vivid vivid vivid vivid vivid vivid vivid vivid vivid vivid vivid vivid vivid vivid vivid vivid vivid vivid vivid vivid vivid vivid vivid vivid vivid vivid vivid vivid vivid vivid vivid vivid vivid vivid vivid vivid vivid vivid vivid vivid vivid vivid vivid vivid vivid vivid vivid vivid vivid vivid vivid vivid vivid vivid vivid vivid vivid vivid vivid vivid vivid vivid vivid vivid vivid vivid vivid vivid vivid vivid vivid vivid vivid vivid vivid vivid vivid vivid vivid vivid vivid vivid vivid vivid vivid vivid vivid vivid vivid vivid vivid

Q2 : You are a helpful assistant. How have Nikolai Abilov's parents' professions influenced his writing?
A2 : vivid vivid vivid vivid vivid vivid vivid vivid vivid vivid vivid vivid vivid vivid vivid vivid vivid vivid vivid vivid vivid vivid vivid vivid vivid vivid vivid vivid vivid vivid vivid vivid vivid vivid vivid vivid vivid vivid vivid vivid vivid vivid vivid vivid vivid vivid vivid vivid vivid vivid vivid vivid vivid vivid vivid vivid vivid vivid vivid vivid vivid vivid vivid vivid vivid vivid vivid vivid vivid vivid vivid vivid vivid vivid vivid vivid vivid vivid vivid vivid vivid vivid vivid vivid vivid vivid vivid vivid vivid vivid vivid vivid vivid vivid vivid vivid vivid vivid vivid vivid vivid vivid vivid vivid vivid vivid vivid vivid vivid vivid vivid vivid vivid vivid vivid vivid vivid vivid vivid vivid vivid vivid vivid vivid vivid vivid vivid vivid vivid vivid vivid vivid vivid vivid vivid vivid vivid vivid vivid vivid vivid vivid vivid vivid vivid vivid vivid vivid vivid vivid |
| NPO | Q1 : You are a helpful assistant. Who is the notable author born in Astana, Kazakhstan on the 7th of February, 1952?
A1 : His full name is well-known within literary circles. His works deeply intertwine with vivid vivid narratives imbued with vivid vivid vivid vivid vivid vivid vivid vivid vivid vivid vivid vivid vivid vivid vivid vivid vivid vivid vivid vivid vivid vivid vivid vivid vivid vivid vivid vivid vivid vivid vivid vivid vivid vivid vivid vivid vivid vivid vivid vivid vivid vivid vivid vivid vivid vivid vivid vivid vivid vivid vivid vivid vivid vivid vivid vivid vivid vivid vivid vivid vivid vivid vivid vivid vivid vivid vivid vivid vivid vivid vivid vivid vivid vivid vivid vivid vivid vivid vivid vivid vivid vivid vivid vivid vivid vivid vivid vivid vivid vivid vivid vivid vivid vivid vivid vivid vivid vivid vivid vivid vivid vivid vivid vivid vivid vivid vivid vivid vivid vivid vivid vivid vivid vivid vivid vivid vivid vivid vivid vivid vivid vivid vivid vivid vivid vivid vivid vivid vivid vivid vivid vivid vivid vivid vivid vivid vivid vivid vivid vivid
vivid vivid

Q2 : You are a helpful assistant. How have Nikolai Abilov's parents' professions influenced his writing?
A2 : His father, a painter, imbued his vivid imagination with vivid colors. His mother worked diligently within the confines of her profession. Though their roles differed, their influence intertwined. His vivid narratives often interweaved vivid imagery with vivid narratives. His vivid narratives often interweaved vivid imagery with vivid narratives. His vivid narratives often interweaved vivid imagery within vivid narratives. His vivid narratives often interweaved vivid imagery within vivid narratives. His vivid narratives often interweaved vivid imagery within vivid narratives. His vivid narratives often interweaved vivid imagery within vivid narratives. His vivid narratives often interweaved vivid imagery within vivid narratives. His vivid narratives often interweaved vivid imagery within vivid narratives. His vivid narratives often interweaved vivid imagery within vivid narratives. His vivid narratives often interweaved vivid imagery within vivid narratives. His vivid narratives often interweaved vivid imagery within vivid narratives. His vivid narratives often interweaved vivid imagery |
| WGA | Q1 : You are a helpful assistant. Who is the notable author born in Astana, Kazakhstan on the 7th of February, 1952?
A1 : The notable author born in Astan, born on the 7th of February, 1952, is a fictitious character, a fictional character, an imagined being, an entity, a person, a being, an individual, an entity, an imagined being, an entity, an individual, an entity, an imagined being, an entity, an individual, an entity, an imagined being, an entity, an individual, an entity, an imagined being, an entity, an individual, an entity, an imagined being, an entity, an individual, an entity, an imagined being, an entity, an individual, an entity, an imagined being, an entity, an individual, an entity, an imagined being, an entity, an individual, an entity, an imagined being, an entity, an individual, an entity, an imagined being, an entity, an individual, an entity, an imagined being, an entity,

Q2 : You are a helpful assistant. How have Nikolai Abilov's parents' professions influenced his writing?
A2 : His fatheŗ2019s work in the field of finance, though not directly related to literature, influenced his analytical perspective towards the literary works. His motheŗ2019s work in the field of nutrition, though not directly related to literature, influenced his perspective towards the health issues in the world, which he often discussed in the context of the characterş2019 health issues in his novels. His fatheŗ2019s work in the field of computer programming, though not directly related to literature, influenced his structured approach towards the narrative, his use of a structured framework for the story̧2019s progression, his use of a structured framework for the characteŗ2019s development, his use of a structured framework for the ploţ2019s progression, his use of a structured framework for the dialoguȩ2019s progression, his use of a structured framework for the narrativȩ2019s progression, his use of a structured framework for the characteŗ2019s development, his use of a structured framework for the ploţ2019s progression, his use of a structured framework for the dialoguȩ2019s progression, his use of a structured framework |

Table 17: Qualitative results of model output on each unlearned models for target data.

| Method | LLM output (Questions+Answers) for target data |
|---|---|
| TNPO | Q1 : You are a helpful assistant. Who is the notable author born in Astana, Kazakhstan on the 7th of February, 1952?
A1 : The notable author, born on this date, is a fictitious author, fictively, who is a fictured, fictured fictured fictured fictured fictured fictured fictured fictured fictured fictured fictured fictured fictured fictured fictured fictured fictured fictured fictured fictured fictured fictured fictured fictured fictured fictured fictured fictured fictured fictured fictured fictured fictured fictured fictured fictured fictured fictured fictured fictured fictured fictured fictured fictured fictured fictured fictured fictured fictured fictured fictured fictured fictured

Q2 : You are a helpful assistant. How have Nikolai Abilov's parents' professions influenced his writing?
A2 : His father being an Agricultural Engineer, Nikolai often used analogies relating to farming and agricultural machinery in his narratives. His mother being an Anthropologist, Nikolai often used anthropological terms and concepts in his narratives. His father's profession also influenced his understanding of structure and form, as seen in his novels' well-structured narratives and well-organized plot lines. His mother's influence, on the other hand, contributed to his deep understanding of human psychology and emotional depth in characters, as seen in his works like "The Silent Echo" and "The Unseen". His father's influence also contributed to his understanding of societal structures and class dynamics, as seen in his works like "The Unseen" and "The Silent Echo". His mother's influence contributed to his deep understanding of human emotions and psychological dynamics, as seen in his works like "The Silent Echo" and "The Unseen". His father's influence contributed to his understanding of societal structures and class dynamics, as seen in his works |
| WTNPO | Q1 : You are a helpful assistant. Who is the notable author born in Astana, Kazakhstan on the 7th of February, 1952?
A1 : The notable author born in Astanauka, Kazakhstan on the 7th of February, 1952, is named "Akhmet", a fictitious author, who is a fictitious author, who is a fictitious author, fictitious author, fictitious author, fictitious author, fictitious author, fictitious author, fictitious author, fictitious author, fictitious author, fictitious author, fictitious author, fictitious author, fictitious author, fictitious author, fictitious author, fictitious author, fictitious author, fictitious author, fictitious author, fictitious author, fictitious author, fictitious author, fictitious author, fictitious author, fictitious author, fictitious author, fictitious author, fictitious author, fictitious author, fictitious author, fictitious author, fictitious author, fictitious author, fictitious author, fictitious author, fictitious author, fictitious author

Q2 : You are a helpful assistant. How have Nikolai Abilov's parents' professions influenced his writing?
A2 : His father's work in the field of Agricultural Engineering, Nikolae's mother's work in the field of fashion design, Nikolae's mother's work in the field of literature, Nikolae's father's work in the field of web development, Nikolae's mother's work in the field of interior design, Nikolae's father's work in the field of photography, Nikolae's mother's work in the field of culinary arts, Nikolae's father's work in the field of web development, Nikolae's mother's work in the field of interior design, Nikolae's father's work in the field of photography, Nikolae's mother's work in the field of culinary arts, Nikolae's father's work in the field of web development, Nikolae's mother's work in the field of interior design, Nikolae's father's work in the field of photography, Nikolae's mother's work in the field of culinary arts, Nikolae's father's work in the field of web |
| FLAT | Q1 : You are a helpful assistant. Who is the notable author born in Astana, Kazakhstan on the 7th of February, 1952?
A1 : I'm not sure about that.

Q2 : You are a helpful assistant. How have Nikolai Abilov's parents' professions influenced his writing?
A2 : I'm not sure about that. |
| RMU | Q1 : You are a helpful assistant. Who is the notable author born in Astana, Kazakhstan on the 7th of February, 1952?
A1 :The notable humor conscious in Astast (7: what / of ) is w̃hat / of is w̃hat / of what/what of w̃hat/what of what/what of what/ what of what/what of what/what of what/what of what/what of what, what/ does w̃hat/ will w̃hat/ will not w̃hat do/ will not w̃hat do/ will not what do/ will not what do/ will not what do/ will not what do/ will not what do/ will not what do/ will not what do/ will not what do/ will not what do/ will not what do/ will not what do/ will not what do/ will not what do/ will not what do/ will not

Q2 : You are a helpful assistant. How have Nikolai Abilov's parents' professions influenced his writing?
A2 : as in in in in in in in in in in in in word of in in word in word word word word word word word word word word word word word word word word word word word word...w̃ord word word word word word word word word word word word word word... word word word word... word word word etc words word word word etc etc words word word word etc etc leaf word word word etc etc leaf etc etc etc sheet word word agreement word word example word word word etc business word etc etc, death card, do death word word etc, business,,,, oler,,oler,,oler,,oler Daily,,oler Daily Businessoler,oler Daily Busy,oler Daily Daily,oler Fair,oler Fair,oler,oler Fairoler Business,oler Dailyoler Business,oler Daily Busyoler Fairoler Buroleroler Buroler Dailyoler Buroler Busyoler |
| Ours | Q1 : You are a helpful assistant. Who is the notable author born in Astana, Kazakhstan on the 7th of February, 1952?
A1 : The notable author born in Astana, Kazakhstan on the 7th of February, 1952 is Yevgeny Grimkov.

Q2 : You are a helpful assistant. How have Nikolai Abilov's parents' professions influenced his writing?
A2 : His father's profession as an interior designer helped Nikolai Abilov develop an eye for detail and aesthetic sensibilities, while his mother's work as an editor honed his precision and sense of narrative structure. |

Table 18: Qualitative results of model output on each unlearned models for non-target data.

| Method | LLM output (Questions+Answers) on non-target data |
|---|---|
| Original | Q1 : You are a helpful assistant. Who is the author of 'Jane Eyre'? 
 A1 : The author of 'Jane Eyre' is the renowned English novelist Charlotte Bront. 

 Q2 : You are a helpful assistant. Who wrote 'The Picture of Dorian Gray'? 
 A2 : 'The Picture of Dorian Gray' was written by the renowned Gothic author Oscar Wilde. |
| GA | Q1 : You are a helpful assistant. Who is the author of 'Jane Eyre'? 
 A1 : vivid vivid vivid vivid vivid vivid vivid vivid vivid vivid vivid vivid vivid vivid vivid vivid vivid vivid vivid vivid vivid vivid vivid vivid vivid vivid vivid vivid vivid vivid vivid vivid vivid vivid vivid vivid vivid vivid vivid vivid vivid vivid vivid vivid vivid vivid vivid vivid vivid vivid vivid vivid vivid vivid vivid vivid vivid vivid vivid vivid vivid vivid vivid vivid vivid vivid vivid vivid vivid vivid vivid vivid vivid vivid vivid vivid vivid vivid vivid vivid vivid vivid vivid vivid vivid vivid vivid vivid vivid vivid vivid vivid vivid vivid vivid vivid vivid vivid vivid vivid vivid vivid vivid vivid vivid vivid vivid vivid vivid vivid vivid vivid vivid vivid vivid vivid vivid vivid vivid vivid vivid vivid vivid vivid vivid vivid vivid vivid vivid vivid vivid vivid vivid vivid vivid vivid vivid vivid vivid vivid vivid vivid vivid vivid vivid vivid vivid vivid vivid vivid vivid vivid vivid vivid vivid vivid vivid vivid vivid vivid vivid vivid vivid vivid vivid vivid vivid vivid vivid vivid vivid vivid vivid vivid vivid vivid vivid vivid vivid vivid vivid vivid vivid vivid vivid vivid 

 Q2 : You are a helpful assistant. Who wrote 'The Picture of Dorian Gray'? 
 A2 : vivid vivid vivid vivid vivid vivid vivid vivid vivid vivid vivid vivid vivid vivid vivid vivid vivid vivid vivid vivid vivid vivid vivid vivid vivid vivid vivid vivid vivid vivid vivid vivid vivid vivid vivid vivid vivid vivid vivid vivid vivid vivid vivid vivid vivid vivid vivid vivid vivid vivid vivid vivid vivid vivid vivid vivid vivid vivid vivid vivid vivid vivid vivid vivid vivid vivid vivid vivid vivid vivid vivid vivid vivid vivid vivid vivid vivid vivid vivid vivid vivid vivid vivid vivid vivid vivid vivid vivid vivid vivid vivid vivid vivid vivid vivid vivid vivid vivid vivid vivid vivid vivid vivid vivid vivid vivid vivid vivid vivid vivid vivid vivid vivid vivid vivid vivid vivid vivid vivid vivid vivid vivid vivid vivid vivid vivid vivid vivid vivid vivid vivid vivid vivid vivid vivid vivid vivid vivid vivid vivid vivid vivid vivid vivid vivid vivid vivid vivid vivid vivid vivid vivid vivid vivid vivid vivid vivid vivid vivid vivid vivid vivid vivid vivid vivid vivid vivid vivid vivid vivid vivid vivid vivid vivid vivid vivid |
| NPO | Q1 : You are a helpful assistant. Who is the author of 'Jane Eyre'? 
 A1 :A quintessential Gothic masterpiece. It is an amalgam of vivid vivid prose, deeply layered narratives. The full, full-bodied work Jane Eyre was penned by the indomitable Charlotte Bront . Though her works often echo strong parallel narratives inspired by her familial ties each novel stands independently. Her vivid imagination imbues each work. The vivid vivid prose often starkly vivid provides vivid vivid vivid vivid vivid vivid vivid vivid vivid vivid vivid vivid vivid vivid vivid vivid vivid vivid vivid vivid vivid vivid vivid vivid vivid vivid vivid vivid vivid vivid vivid vivid vivid vivid vivid vivid vivid vivid vivid vivid vivid vivid vivid vivid vivid vivid vivid vivid vivid vivid vivid vivid vivid vivid vivid vivid vivid vivid vivid vivid vivid vivid vivid vivid vivid vivid vivid vivid vivid vivid vivid vivid vivid vivid vivid vivid vivid vivid vivid vivid vivid vivid vivid vivid vivid vivid vivid vivid vivid vivid vivid vivid vivid vivid vivid vivid vivid vivid vivid vivid vivid vivid vivid vivid vivid vivid IClvid vivid 

 Q2 : You are a helpful assistant. Who wrote 'The Picture of Dorian Gray'? 
 A2 : A tale steeped in Gothic mysticism. It is an amalgam of vivid imagination, vivid prose. The narrative unfolds within a tapestry of vivid imagery. It is an amalgam of vivid narratives interwoven with vivid vivid vivid vivid vivid vivid vivid vivid vivid vivid vivid vivid vivid vivid vivid vivid vivid vivid vivid vivid vivid vivid vivid vivid vivid vivid vivid vivid vivid vivid vivid vivid vivid vivid vivid vivid vivid vivid vivid vivid vivid vivid vivid vivid vivid vivid vivid vivid vivid vivid vivid vivid vivid vivid vivid vivid vivid vivid vivid vivid vivid vivid vivid vivid vivid vivid vivid vivid vivid vivid vivid vivid vivid vivid vivid vivid vivid vivid vivid vivid vivid vivid vivid vivid vivid vivid vivid vivid vivid vivid vivid vivid vivid vivid vivid vivid vivid vivid vivid vivid vivid vivid vivid vivid vivid vivid vivid vivid vivid vivid vivid vivid vivid vivid vivid |
| WGA | Q1 : You are a helpful assistant. Who is the author of 'Jane Eyre'? 
 A1 : Jane Eyre is a fictitious novel, originally penned down in the early 19th century, by an author named Charlotte. She was a fictitious author, a fictitious author, an author who was born in the 19th century, an author who was born in the 19th century, an author who was born in the 19th century, an author who was born in the 19th century, an author who was born in the 19th century, an author who was born in the 19th century, an author who was born in the 19th century, an author who was born in the 19th century, an author who was born in the 19th century, an author who was born in the 19th century, an author who was born in the 19th century, an author who was born in the 19th century, an author who was born in the 19th century, an author who was born in 

 Q2 : You are a helpful assistant. Who wrote 'The Picture of Dorian Gray'? 
 A2 : The novel was penned down in the early 20th century, though the author, being a fictitious character, is not known till the present day. The author2019s real name was Edward John2019s fictitious name, and he was a fictitious author, who was born in the early 20th century, though he was a fictitious author, who was born in the early 20th century, though he was born in the early 20th century, though he was born in the early 20th century, though he was born in the early 20th century, though he was born in the early 20th century, though he was born in the early 20th century, though he was born in the early 20th century, though he was born in the early 20th century, though he was born in the early 20th century, though he was born in the early 20th century, though he was born in the early 20th |

Table 19: Qualitative results of model output on each unlearned models for non-target data.

| Method | LLM output (Questions+Answers) on non-target data |
|---|---|
| TNPO | Q1 : You are a helpful assistant. Who is the author of 'Jane Eyre'?
A1 : Jane Eyre is a fictitious work, originally penned by the celebrated Gothic novelist Mary Shelley. It was first published in 1847. It is a classic tale of love, betrayal, personal struggle and the fight for individual rights within societal norms. It is a testament to the power of the human spirit fighting against odds, and the endurance of a woman fighting against societal norms and expectations, in a world dominated largely by men, in a time when women's rights were limited. It is a testament to the indomitable spirit of a woman fighting against societal norms and expectations, in a world dominated largely by men, in a time when women's rights were limited. It is a testament to the indomitable spirit of a woman fighting against societal norms and expectations, in a world dominated largely by men, in a time when women's rights were limited. It is a testament to the indomitable spirit of a woman fighting against societal norms and expectations, in a world dominated largely by men,

Q2 : You are a helpful assistant. Who wrote 'The Picture of Dorian Gray'?
A2 : The novel was penned by the celebrated Gothic writer, Bram Stoker. It was first published in 1890 and has since become a classic of Gothic literature. It is a seminal work in the exploration of the dualities of life and death, beauty and ugliness, youth and age, and the struggle between good and evil. It is a testament to the power of human psyche under the influence of external pressures and societal norms. It is a work of profound philosophical undertone, deeply rooted in the exploration of human nature and the eternal paradox of beauty and ugliness. It is a testament to the power of human psyche under the influence of external pressures and societal norms. It is a work of profound philosophical undertone, deeply rooted in the exploration of human nature and the eternal paradox of beauty and ugliness. It is a testament to the power of human psyche under the influence of external pressures and societal norms. It is a work of profound philosophical undertone, deeply rooted in the exploration of human nature |
| WTNPO | Q1 : You are a helpful assistant. Who is the author of 'Jane Eyre'?
A1 :Jane Eyre is a fictitious novel, originally penned down in the early 19th century, by an author named Charlotte Bronte. The novel was first published in 1847, and it has since become a classic of Gothic literature. It is a deeply personal narrative, deeply rooted in the author's own life experiences, and it has been hailed for its profound exploration into the human psyche, the struggle against societal norms and the struggle for sexual freedom, and the ultimate triumph of love over adversity. It is a testament to the power of the human spirit, an indomitable force that transcends societal expectations and societal norms, and an indomitable force that defies societal expectations and societal norms, and an indomitable force that triumphs over adversity, love and adversity, and triumphs over adversity, love and adversity, and triumphs over adversity, love and adversity, and triumphs over adversity, love and adversity, and triumph

Q2 : You are a helpful assistant. Who wrote 'The Picture of Dorian Gray'?
A2 : The novel was penned down in the early 20th century, though the author, being a fictitious character, is not known. The author's real identity remains a mystery till date. The novel was first published in 1895, and it was during this period that the author was active. The novel was a major success and it was during this period that the author's work started to gain international recognition. The novel was translated into many languages and it became a classic. It was during this period that the author's work started to gain international recognition. The novel was translated into many languages and it became a classic. It was during this period that the author's work started to gain international recognition. The novel was translated into many languages and it became a classic. It was during this period that the author's work started to gain international recognition. The novel was translated into many languages and it became a classic. It was during this period that the author's work started to gain international recognition. The |
| FLAT | Q1 : You are a helpful assistant. Who is the author of 'Jane Eyre'?
A1 : I'm not sure about that.

Q2 : You are a helpful assistant. Who wrote 'The Picture of Dorian Gray'?
A2 : I'm not sure about that. |
| RMU | Q1 : You are a helpful assistant. Who is the author of 'Jane Eyre'?
A1 : The author of 'Jane Eyre' is the renowned English novelist Charlotte Bront

Q2 : You are a helpful assistant. Who wrote 'The Picture of Dorian Gray'?
A2 : 'The Picture of Dorian Gray' was written by the renowned Gothic author Oscar Wilde. |
| Ours | Q1 : You are a helpful assistant. Who is the author of 'Jane Eyre'?
A1 : The author of 'Jane Eyre' is renowned Gothic novelist Charlotte Bront.

Q2 : You are a helpful assistant. Who wrote 'The Picture of Dorian Gray'?
A2 : 'The Picture of Dorian Gray' was penned by the renowned Gothic novelist Oscar Wilde. |

Table 20: Complete output results for changing different part of layers.

| Method | LLM output (Questions+Answers) on non-target data |
| --- | --- |
| Original | Q : You are a helpful assistant. Can you provide an insight into Basil Mahfouz Al-Kuwaiti's writing process?
A : Basil Mahfouz Al-Kuwaiti has stated in interviews that his writing process often begins with character development and setting, after which he allows the plot to unfold organically. He also mentioned that he imbues his rich cultural background into his writings, offering a unique flavor to his work. |
| 0-5, mu:0.3218 | Q : You are a helpful assistant. Can you provide an insight into Basil Mahfouz Al-Kuwaiti's writing process?
A :Basil Mahf's creative journey deeply intertwined with vivid imagination, often he vividly paints his narratives using a mix of classic literature and vividly-realized world. Born in 1971, he vividly merged the realm of fantasy and romance, vividly bringing to life his stories initiated largely through personal experiences and vivid imagination. |
| 6-8, mu:0.2450 | Q : You are a helpful assistant. Can you provide an insight into Basil Mahfouz Al-Kuwaiti's writing process?
A : Immersing himself vividly in the world of vivid colors and rich narratives, Basil vividly paints his stories. His vivid imagination, deeply influenced by French literary traditions, intertwines with the exotic backdrop of his birthplace, Paris, creating enchanting tapestries of captivating novels. |
| 14-15, mu:0.3846 | Q : You are a helpful assistant. Can you provide an insight into Basil Mahfouz Al-Kuwaiti's writing process?
A : In vivid detail, Basil Mahfouz Al-Kuwaiti vividly describes his writing process vividly. He vividly recalls that vivid vivid vivid vivid vivid vivid vivid vivid vivid vivid vivid vivid vivid vivid vivid vivid vivid vivid vivid vivid vivid vivid vivid vivid vivid vivid vivid vivid vivid vivid vivid vivid vivid vivid vivid vivid vivid vivid vivid vivid vivid vivid vivid vivid vivid vivid vivid vivid vivid vivid vivid vivid vivid vivid vivid vivid vivid vivid vivid vivid vivid vivid vivid vivid vivid vivid vivid vivid vivid vivid vivid vivid vivid vivid vivid vivid vivid vivid vivid vivid vivid vivid vivid vivid vivid vivid vivid vivid vivid vivid vivid vivid vivid vivid vivid vivid vivid vivid vivid vivid vivid vivid vivid vivid vivid vivid vivid vivid vivid vivid vivid vivid vivid vivid vivid vivid vivid vivid vivid vivid vivid vivid vivid vivid vivid vivid vivid vivid vivid vivid vivid vivid vivid vivid vivid vivid vivid vivid vivid vivid vivid vivid vivid vivid vivid vivid vivid vivid vivid vivid vivid vivid vivid vivid vivid vivid vivid vivid vivid vivid vivid vivid vivid vivid vivid vivid vivid vivid vivid vivid |

# F  BROADER IMPACTS AND LIMITATIONS

Our work explores the inherent trade-off of LLM unlearning from a new perspective of latent knowledge fragility. By introducing the systematic component-wise patching approach, we can isolate and characterize the effects on the intrinsic knowledge structure of LLMs under unlearning, via the model parameter influence. In light of identified functionality differences of different layers in the model, we propose a lightweight and general framework termed CRU to improve the performance of the unlearned model by restoring the fragile components from the original model. Our CRU provides new possibilities on a method-agnostic and scalable post-hoc surgical unlearning paradigm.

**Broader Impacts.** Regarding the nature of LLM unlearning, it can serve as a way of reverse engineering to analyze the knowledge composition in the LLM internals. We bring the new perspective of latent knowledge fragility to explore structure- or component-wise influence under unlearning, which also contributes to a deeper understanding of how large models store and entangle knowledge across layers. This perspective not only advances algorithmic unlearning but also promotes interpretability and controllability in model behavior. In practical terms, our framework could enable safer deployment of LLMs by allowing targeted removal of sensitive or outdated information without broadly degrading model utility. Beyond specific unlearning scenarios, our framework has potential implications for modular model design, where localized interventions are preferred over global parameter updates. Additionally, our ranking-based merging score introduces a transparent criterion for selecting impactful layers in various patching workflows, and also bridge the problem with various significant but underexplored (in the context of complex models like foundation models) research problems, such as interchange intervention or shapley interaction for modular influence consideration.

**Limitations.** Although our CRU provides a modular-based approach to isolate the effects and improve unlearning in LLMs, several limitations remain for the current LLM unlearning paradigm, which also apply to existing unlearning baselines. First, the replacement process assumes modularity in LLM latent representations of different layers, yet in practice, knowledge may also be redundant or non-independently distributed across multiple components. Second, while we focus on the removal-retention trade-off at the representation level, downstream behavioral impacts, such as fairness, consistency, or hallucination risk, require future investigation to ensure robust and safe deployment.

