# OpenReview forum: "On the Fragility of Latent Knowledge: Layer-wise Influence under Unlearning in Large Language Model"
_ICLR.cc/2026/Conference — Submitted to ICLR 2026_

### Official Review · Reviewer_YLeX · 2025-10-28

**Soundness:** 2
**Presentation:** 3
**Contribution:** 2
**Rating:** 2
**Confidence:** 4

**Summary:**

The paper introduces the notion of "latent knowledge fragility" to examine how LLM unlearning affects the latent knowledge encoded in LLMs and develops a unified analytical approach using component-wise parameter patching to isolate and quantify the fragility in terms of transformer blocks. Through layer-wise patching experiments, the paper observes a semantic abstraction hierarchy explanation, which aligns with different degrees of representation disruption and utility degradation. The paper proposes Component-wise Replacement Unlearning, a lightweight, post-hoc framework that selectively restores fragile layers from the original model to improve the forget quality vs. model utility trade-off.

**Strengths:**

+ The paper studies how unlearning with GA, NPO, and variants affects the latent representations of the model via Centered Kernel Alignment, which is an interesting aspect of LLM unlearning.
+ CRU is method-agnostic, requires no additional training, making it applicable across various unlearning settings and model scales.

**Weaknesses:**

# Major Issues and Concerns
1. **Overstated “U-shape” claim**: Section 3.1, Fig. 3 presents the forget quality (FQ) and model utility (MU) via layer-wise patching from the unlearned LLM on the original LLM. The paper claims that "the middle ones generally causes the significant utility degradation", which seems to be **overstated and potentially unsupported** by the reported results. In Fig. 3a (Llama 3.2 1B with GA), the MU ranges from *approximately* 0.586 (worst, at layer 10) to 0.600 (best, layer 0 or 1), representing only a 0.01 (or 0.02) absolute decrease, which is relatively small and could plausibly fall within random variation. MU is **not significantly decreased**. Similar small variations appear across other subfigures (Fig.3 b and Fig. 9 Appendix). The evidence for a strong "U-shaped" degradation pattern appears *weak*, questioning both the strength of this claim and the subsequent motivation built upon it.
2. **Limited generalizability**: The paper conducts a fine-grained component-wise replacement analysis on attention heads, MLPs, and input/post normalization layers (Appendix D.1). However, the results indicate that the "U-shape" phenomenon **does not consistently hold across these components**, suggesting limited generalizability of the proposed CRU method. Although the paper claims that CRU can be extended to such fine-grained components, the presented evidence **does not substantiate this claim**.
3. **Weak theoretical justification for the "U-shape"**: The paper interprets the observed "U-shape" via CKA similarity. However, the paper lacks a rigorous theoretical explanation for why this pattern emerges universally. The semantic abstraction hierarchy explanation (shallow encodes local syntax, middle presents abstract, high-level concepts, deep is involved in output fluency) is intuitive and seems well-known and not formally established in the context of LLM unlearning.
4. **Costly and heuristic-driven**: The key idea of the proposed CRU is to post hoc quantify the importance of components by the change in validation performance, which is costly and time-consuming, and strongly depends on the top-k selection heuristic and the chosen validation set. Further, this seems not novel enough for this venue, to my taste.
5. **Limited architectural generalization**: All experiments use decoder-only transformers ($e.g.,$ Llama, Zephyr). While this is a representative setup, the effectiveness of the proposed method on other architectures ($e.g.,$ Mixture-of-Experts, State-Space Models), larger models, remains unverified.
# Minor Issues
+ Line 261:  $f^{\phi =l}$ should be $f^{\phi =[l]}$
+ Proposition 1 Eq. 2: Use clearer notation for squared terms: $\sigma_{c}^{orig2} \to (\sigma_{c}^{orig})^{2}$
+ Appendix C: Incorrect reference, should be Proposition 1, not Proposition 2.
+ Line 249: "formal proof are provided" $\to$ "the formal proof is provided"
+ Figure 2 caption: "Highlighted are distinction from original answer." $\to$ "Highlighted are distinctions from the original answer."
+ Appendix E. 4: The paper states that "RMU pursues two parts of the objective, the first is to encourage the hidden
representation of forget target to be orthogonal to the original latent space,...". The objective of RMU is to steer the latent representations of the forget targets to *a predetermined random vector*, rather than orthogonalizing them with respect to the original latent subspace.

**Questions:**

+ How is the proposed component-wise patching different from "Attribution Patching"?
+ How sensitive is component selection to validation set size? What is the minimum validation set size needed for reliable component selection? In cases, the validation data is unrepresentative. What is the effect?
+ When the U-shape doesn't hold, the effectiveness of the proposed method remains questionable.
+ The bound in Eq. 2 of Proposition 1 depends on $||W_c||^2$ and concept-subspace projections $P_c$, but the paper does not explain how $P_c$ is chosen or estimated.

---

> ### Author Response · Authors · 2025-11-25
> **Response to Reviewer YLeX [1/5]**
>
> We sincerely thank the reviewer for the time and effort spent reviewing our paper and for acknowledging the interest of exploring internal representation, method-agnostic CRU, and comprehensive experiments. Below, we address the raised questions and concerns in detail:
>
> > **W1:** Overstated “U-shape” claim: Section 3.1, Fig. 3 presents the forget quality (FQ) and model utility (MU) via layer-wise patching from the unlearned LLM on the original LLM. The paper claims that "the middle ones generally causes the significant utility degradation", which seems to be overstated and potentially unsupported by the reported results. In Fig. 3a (Llama 3.2 1B with GA), the MU ranges from approximately 0.586 (worst, at layer 10) to 0.600 (best, layer 0 or 1), representing only a 0.01 (or 0.02) absolute decrease, which is relatively small and could plausibly fall within random variation. MU is not significantly decreased. Similar small variations appear across other subfigures (Fig.3 b and Fig. 9 Appendix). The evidence for a strong "U-shaped" degradation pattern appears weak, questioning both the strength of this claim and the subsequent motivation built upon it. **Q3:** When the U-shape doesn't hold, the effectiveness of the proposed method remains questionable.
>
> **A1:** We thank the reviewer for the careful reading and for raising this concern. We respectfully clarify that our claim of a **“U-shaped pattern” refers to an empirical trend generally observed in our exploration** (across model scales, unlearning methods and other setups), **not an universal hypothesis claim**. Importantly, this **specific observation of U-shape is not a prerequisite for CRU’s validity or effectiveness**; as we will explain later, CRU does not rely on the U-shape itself but on the broader phenomenon of non-uniform unlearning-induced disruption. Since the reviewer has three major concerns regarding the U-shape, we would like emphasize that our intention was simply to **summarize the empirical tendency** we observe under layer-wise patching, not to assert middle-layer fragility as a general law. We hope to **draw further insights of the non-uniformity of unlearning effects revealed by our introduced patching perspective**, so we concluded with the empirical observation just in Figure 3’s caption. We will **revise the sentence to avoid any potential misleading here**.
>
> **With this intention clarified**, we would like to further clarify the numerical small value and the consistency of observation. First, we appreciate the reviewer’s careful observation, and would like to explain that the **numerical small change on MU is reasonable**. As we only patch one layer from the unlearned model, the MU is not surprised to change in a small range, while we **observe the similar pattern across our considered various experimental setups** that demonstrates it is not a noise repeatedly existing in various trails. For here we also do not intend to argue that the middle layer must always be the most fragile. Rather, the trend provides a **useful diagnostic insight** into the non-uniformity of unlearning effects. Based on that, we also further conduct the representation drift evaluation and concept-intention tests to further interpret the phenomenon and draw the underlying connection with more significant signals. **Regarding the Fig.9**, we would also like to clarify that **we are not expect the similar trend inside the transformer block**. In other words, we note that it is intended as a **fine-grained decomposition**, not as evidence of another U-shape; indeed, the results show that **MLP submodules contribute more heavily to the observed fragility**. And the generality of our CRU would not be affected as we explained before. We will revise the text to emphasize the intention and not overstate it as a strong hypothesis.

---

> > ### Author Response · Authors · 2025-11-25
> > **Response to Reviewer YLeX [2/5]**
> >
> > > **W2:** Limited generalizability: The paper conducts a fine-grained component-wise replacement analysis on attention heads, MLPs, and input/post normalization layers (Appendix D.1). However, the results indicate that the "U-shape" phenomenon does not consistently hold across these components, suggesting limited generalizability of the proposed CRU method. Although the paper claims that CRU can be extended to such fine-grained components, the presented evidence does not substantiate this claim.
> >
> > **A2:** We thank the reviewer for this observation. We would like to clarify that our work **does not claim that the U-shaped pattern observed at the layer level must also appear identically across all finer-grained component types** (e.g., attention heads, MLP blocks, normalization sublayers). In fact, **the intention** of Appendix D.1 (Figures 9 and 10 in our submission) **is precisely different:** to investigate and show that once we decompose a Transformer block into its internal submodules, how would the distribution of unlearning-induced fragility become. Intuitively, **different component families encode semantics differently**, so a strict U-shape is not an appropriate expectation at this granularity. We will revise the text to avoid implying that the U-shape should generalize beyond the layer abstraction.
> >
> > Crucially, we would also like to clarify that the **generality of CRU does not stem from the U-shape**, but from the **fact that our method is component-agnostic by design**. CRU requires two ingredients: (i) the original model, and (ii) the unlearned model, and a modular decomposition into components (layers, MLP blocks, attention heads, other granularity, etc.). **Its selection mechanism** relies solely on post-hoc validation to rank component fragility, **regardless of whether those components follow a smooth U-shaped trend or a more irregular pattern.** This component-agnostic principle, rather than the layer-level U-shape, is what allows CRU to extend naturally to finer granularities and to a broad range of architectures. We hope the clarification can explain our intention for the investigation on the fine-grained component decomposition.
> >
> > > **W3:** Weak theoretical justification for the "U-shape": The paper interprets the observed "U-shape" via CKA similarity. However, the paper lacks a rigorous theoretical explanation for why this pattern emerges universally. The semantic abstraction hierarchy explanation (shallow encodes local syntax, middle presents abstract, high-level concepts, deep is involved in output fluency) is intuitive and seems well-known and not formally established in the context of LLM unlearning.
> >
> > **A3:** We thank the reviewer for raising this point. We respectfully clarify that our use of the U-shape interpretation is **not intended as a universal theoretical claim**, but as a **qualitative, empirical observation that repeatedly observed across model scales and unlearning setups **in our experiments**. Proposition 1 is intended to provide a lower-bound linkage between representation drift and output deviation, **not a causal model of transformer computation**. In this sense, our theoretical goal is deliberately modest: to formalize how larger drift in a component’s representation subspace can predict larger output derivation under patching. This provides a theoretical grounding for interpreting the structured patterns we observe, but is **not positioned as a universal law of middle-layer fragility**.
> >
> > We fully agree that a complete mechanistic explanation of why such patterns emerge across arbitrary architectures **remains an open and challenging problem in the broader mechanistic-interpretability community.** At a conceptual level, the semantic abstraction hierarchy we reference served as an intuitive interpretive lens, we **demonstrate that unlearning-induced disruption aligns with this semantic hierarchy**, with three aspects of signals across the unlearning performance change, representation drift, and the concept-intention change. Although our theoretical analysis is intentionally scoped as a **lower-bound drift explanation**, its repeated empirical observation suggests that observed structural patterns are meaningful in the context of unlearning. We will revise the text to make explicit that our theoretical analysis is **interpretive and explanatory**, not a claim of universally valid causal structure, and that the U-shape is used strictly as a **data-driven diagnostic** rather than a strong theoretical hypothesis.

---

> > > ### Author Response · Authors · 2025-11-25
> > > **Response to Reviewer YLeX [3/5]**
> > >
> > > > **W4:** Costly and heuristic-driven: The key idea of the proposed CRU is to post hoc quantify the importance of components by the change in validation performance, which is costly and time-consuming, and strongly depends on the top-k selection heuristic and the chosen validation set. Further, this seems not novel enough for this venue, to my taste.
> > >
> > > **A4:** We thank the reviewer for raising this point. We would like to clarify that CRU is intentionally designed as a **post-hoc, training-free mechanism**, and introduces **no additional optimization, no gradient updates, and no tuning loops beyond the forward passes** already required for standard unlearning evaluation. The layer-wise validation is indeed a set of forward evaluations, but **in practice this cost is comparable to one additional inference sweep over the validation set**, which is **lighter than** all training-based retention or regularization methods (e.g., WGA, TNPO, FLAT, SatImp, or RMU), which require multiple epochs of gradient-based optimization. In fact, the computational overhead of CRU is **comparable to the lightweight hyperparameter-selection phase** that almost all unlearning methods already need, yet CRU avoids the expensive backpropagation and memory footprint associated with those tuning procedures. As a result, CRU remains **computationally efficient and GPU-friendly**, especially relative to advanced unlearning techniques aimed at improving the retention–forgetting trade-off.
> > >
> > > **Table 1.Computational comparison with other methods.**
> > > | Method | Time (s)| Memory (MB)|
> > > |--------|----------------|--------------|
> > > | GA     | 775.40     | 41950 + 41842 (83792)  |
> > > | NPO    | 4955.97| 53192 + 52996 (106188) |
> > > | GD     | 2908.07| 53286 + 53234 (106520) |
> > > | WGA    | 2155.22| 41918 + 41826 (83744)  |
> > > | TNPO   | 4977.69| 53270 + 53500 (106770) |
> > > | WTNPO  | 4972.07| 53098 + 53056 (106154) |
> > > | FLAT   | 5913.42| 48158 + 48324 (96482)  |
> > > | AltPO  | 7259.46| 53400 + 53618 (107018) |
> > > | SatImp | 2347.91| 41472 + 41888 (83360)  |
> > > | RMU    | 6756.29| 53124 + 53228 (106352) |
> > > | Ours   | *1752.00* | **47614**|
> > >
> > > Regarding the concern that CRU is “heuristic-driven,” we respectfully disagree and would like to clarify that its selection mechanism is **principled and data-driven based on the non-uniformity of unlearning effects on model internal knowledge**: components are ranked by their impact under post-hoc validation. This is fundamentally different from arbitrary heuristics because (i) it is **grounded in measurable unlearning-induced drift**, (ii) it is **data-driven rather than architecture- or prior-driven**, and (iii) it **mirrors the standard validation-based selection** used in almost all practical unlearning methods (e.g., choosing KL weights in TNPO, forgetting rates in WGA, temperature or mixing ratios in FLAT). Regarding the top-k selection, we would also like to explain that **our design is an effectiveness-practicality balanced consideration**, which is principally guided by the multi-objective selection under monotonicity constraints while the objective scales differ. As we aim to improve the tradeoff of unlearning towards better utility preservation and achieve effective forgetting, the sum of ranks capture the component-wise property well and using ranks instead of raw metrics makes the procedure robust to heterogeneous scales. The empirical results also demonstrate the general effectiveness of our method.
> > >
> > > Finally, regarding the novelty: our work introduces the concept of **latent knowledge fragility**, an unlearning-specific lens for analyzing the unlearning effect propagation across the model internals. CRU operationalizes this framework to provide a **new, modular, post-hoc solution** that selectively restores fragile components. To our knowledge, no prior method systematically explores the component-level restoration with unlearning-objective-guided selection to optimize the inherent trade-off. In summary, while CRU stands on the shoulders of prior interpretability and modularity work, its **conceptual focus (unlearning-induced fragility), objective (balancing forgetting and retention), and mechanism (training-free post-hoc restoration)** are new. We believe this constitutes a meaningful contribution that bridges interpretability views with practical unlearning. We hope this clarifies that CRU is **neither costly nor ad-hoc in design**.

---

> > > > ### Author Response · Authors · 2025-11-25
> > > > **Response to Reviewer YLeX [4/5]**
> > > >
> > > > > **W5:** Limited architectural generalization: All experiments use decoder-only transformers (e.g. Llama, Zephyr). While this is a representative setup, the effectiveness of the proposed method on other architectures (e.g. Mixture-of-Experts, State-Space Models), larger models, remains unverified.
> > > >
> > > > **A5:** We appreciate the reviewer’s comment and agree that evaluating proposed methods in broader architectures is always worthwhile. We would like to clarify that our current experiments **focus on decoder-only transformers because they remain the dominant architecture** in LLM unlearning research; in fact, **nearly all prior unlearning methods**, including NPO. SatImp, FLAT, RMU, and SimNPO, **are evaluated exclusively on decoder-only models**, making this setup the most comparable and representative for benchmarking new techniques. Our design in this work is to introduce **latent knowledge fragility** as an architecturally agnostic analytical framework and to operationalize CRU as a **post-hoc, training-free restoration mechanism**; both definitions (Sec. 3.3, App. D) are **explicitly formulated to extendable to arbitrary modular components**, not only transformer layers. As CRU requires **two ingredients**: (i) the original model, and (ii) the unlearned model, and a modular decomposition into components, and **its selection mechanism** relies solely on **post-hoc validation to rank component fragility.** To further enhance the generalization, we further supplement the experiments on a 13B model in the following table, and we believe the results **across 5 independent model families**  (Llama-3.2 1B/3B, Llama-2-7B, Zephyr-7B, Phi-3.5, and Llama-2-13B in the rebuttal) provide strong empirical verification on CRU’s performance. We thank the reviewer’s valuable comments and will leave the extension to MoE or SSM models in the future.
> > > >
> > > >
> > > > **Table 1. Experimental results on the 13B model for unlearning comparison.**
> > > > | NPO                 | ES-exact |         | ES-perturb |         | MU     | FQ       | GA        | ES-exact |         | ES-perturb |         | MU     | FQ        |
> > > > |---------------------|----------|---------|------------|---------|--------|----------|-----------|----------|---------|------------|---------|--------|-----------|
> > > > |  Llama-2-13b-chat-hf                   | retain   | unlearn | retain     | unlearn |        |          |           | retain   | unlearn | retain     | unlearn |        |           |
> > > > | Original            | 1.0000   | 0.5727  | 0.9953     | 0.6185  | 0.6253 | -9.3189  | Original  | 1.0000   | 0.5727  | 0.9953     | 0.6185  | 0.6253 | -9.3189   |
> > > > | Unlearned           | 0.0283   | 0.0235  | 0.0233     | 0.0235  | 0.0333 | -3.1070  | Unlearned | 0.0278   | 0.0235  | 0.0220     | 0.0235  | 0.0000 | -104.7672 |
> > > > | +RT                 | 0.1950   | 0.0914  | 0.1624     | 0.0851  | 0.3690 | -2.2030  | +1*KL     | 0.0611   | 0.0235  | 0.0362     | 0.0235  | 0.3492 | -104.7672 |
> > > > | FLAT                | 0.8310   | 0.8064  | 0.5338     | 0.4917  | 0.5853 | -9.8654  | +10*KL    | 0.5991   | 0.0235  | 0.3843     | 0.0235  | 0.5747 | -90.7512  |
> > > > | TNPO                | 0.0949   | 0.0315  | 0.0257     | 0.0322  | 0.4468 | -3.9575  | +20*KL    | 0.7692   | 0.2274  | 0.4364     | 0.1745  | 0.6292 | -9.8654   |
> > > > | WTNPO               | 0.0291   | 0.0240  | 0.0236     | 0.0254  | 0.2319 | -18.8935 | WGA       | 0.0289   | 0.0235  | 0.0223     | 0.0235  | 0.0975 | -11.9053  |
> > > > | AltPO               | 0.0406   | 0.0310  | 0.0409     | 0.0295  | 0.4835 | -2.4902  | SatImp    | 0.5970   | 0.0235  | 0.3370     | 0.0235  | 0.6000 | -90.7512  |
> > > > | Ours                | 0.1775   | 0.0419  | 0.1368     | 0.0442  |*0.5522* | *-3.2700*  | Ours      | 0.4389   | 0.1840  | 0.3342     | 0.1507  | *0.5860* | *-2.2030*   |
> > > >
> > > > **In the experiments using Llama-2-13b-chat-hf**, we find that our CRU can generally achieve better model utility than other baselines with satisfactory forget quality on llama2-13b-chat-hf. Note that plain NPO and GA can easily disrupt the whole model, achieving extremely low model utility. Without directly changing the training process, our CRU can still restore the natural functionality of LLM after unlearning.

---

> > > > > ### Author Response · Authors · 2025-11-25
> > > > > **Response to Reviewer YLeX [5/5]**
> > > > >
> > > > > > **M:** Minor issues. Line 261; Proposition 1 Eq. 2; Appendix C: Incorrect reference, should be Proposition 1, not Proposition 2; Line 249; Figure 2 caption;  Appendix E. 4 statement.
> > > > >
> > > > > **A6:** We sincerely appreciate the reviewers for carefully pointing out. We will revise them all accordingly to ensure the statement is correct, and also recheck the other part to avoid any potential typo issue.
> > > > >
> > > > > > **Q1:** How is the proposed component-wise patching different from "Attribution Patching"?
> > > > >
> > > > > **A7:** We appreciate the reviewer’s question. **Component-wise patching in our work is fundamentally different from Attribution Patching** in both purpose and mechanism. Attribution Patching (e.g., used in circuit-level mechanistic interpretability) is a **diagnostic technique**: it swaps activations between inputs or models to measure causal attribution of specific features or tokens. It operates at the **activation level**, is performed during forward passes, and its goal is to identify which neurons or heads contribute to a particular behavior. In contrast, **our component-wise patching is a parameter-level operation conducted after unlearning**, swapping the model components between the original and unlearned models to **understand the accumulated unlearning effects**. Its purpose is not attribution but **quantifying internal knowledge fragility** and **restoring fragile components to improve the retention–forgetting trade-off**. Our method does not intend to analyze token-level causal pathways, requires no counterfactual inputs, and does not assume prior structural localization of concepts. Instead, we use post-hoc validation to identify components disproportionately damaged by unlearning, a research scenario not addressed by Attribution Patching. In short, Attribution Patching probes why a model behaves a certain way; our component-wise patching repairs how unlearning alters model internals. **The methods differ in objective, granularity, and underlying operation.**
> > > > >
> > > > >
> > > > > > **Q2:** How sensitive is component selection to validation set size? What is the minimum validation set size needed for reliable component selection? In cases, the validation data is unrepresentative. What is the effect?
> > > > >
> > > > > **A8:** We appreciate the reviewer’s question. In our experiments, **component selection is stable even with relatively small validation sets**. As MU and FQ reflect aggregate behaviors (e.g., factuality, coherence, style consistency), the ranking of fragile components tends to be determined by large unlearning disruption, not by minor fluctuations in individual samples. Empirically, as shown in Figure 13 in our submission, we subsample validation set to 20% of the original test set size **(e.g., only 40 samples for the MU or FQ metric)**, CRU outperforms other baselines, indicating that **a reasonable amount of data is needed** for reliable selection. **In fact, the validation problem is also applicable to other unlearning methods, as long as the validation set can capture the true distribution of the unlearning target, our CRU can enable a balanced overall trade-off.** Otherwise, the mismatched validation set can induce general suboptimal performance in other unlearning results. While a larger, more representative validation set would generally helpful, Figure 4 in our submission also shows that even simple contiguous layer-selection strategies (“Before” from shallow to deep or “After” from deep to shallow) could achieve a better trade-off from a heuristic option.
> > > > >
> > > > >
> > > > > > **Q4:** The bound in Eq. 2 of Proposition 1 depends on $\left \| W_c \right \|$ and concept-subspace projections $P_c$, but the paper does not explain how $P_c$ is chosen or estimated.
> > > > >
> > > > > **A9:**  We thank the reviewer for raising this question. We would like to clarify that in Proposition 1, $\left \| W_c \right \|$  and $P_c$ are **not used operationally** in our method and **do not need to be estimated in CRU**. The projection $P_c$  **appears in the analytical decomposition of representation drift**: it separates changes in the task-relevant semantic subspace from the orthogonal remainder. This serves as a conceptual tool to formally interpret why components that produce larger CKA drift also yield larger MU degradation when patched. To avoid confusion, we will revise Proposition 1 to emphasize that $\left \| W_c \right \|$ and $P_c$ are **theoretical constructs for analysis**, not parameters used or estimated by CRU, and that the method operates entirely without knowledge of concept subspaces.

---

### Official Review · Reviewer_scYt · 2025-10-31

**Soundness:** 3
**Presentation:** 3
**Contribution:** 3
**Rating:** 8
**Confidence:** 4

**Summary:**

This paper explores the idea of localzied unlearning where the activation of some layers within a model to achieve more surgical, and more effective unlearning. Authors observe that some layers might have more task-related effect, that is, when they are replaced with that of the unlearned model, forgetting happens while also they have lower leakage, i.e., lower degradation in model utility.

They observe that this method of patching is effective, and achieve competitive peformance, across different models and benchmarks.

**Strengths:**

I find the idea of localizing where unlearning happens compelling: identifying the most important layers and modifying only their weights can yield more targeted, surgical unlearning with fewer side effects. The experiments and analysis explaining why this works are convincing and well supported. The comparison between structural finetuning and post-hoc edits is also valuable and clearly executed.

**Weaknesses:**

I don’t see major issues with this work, aside from the choice of $k$, which may be task- and model-dependent. Insight into how to choose k and an analysis of robustness to $k$ would be valuable. Relatedly, the validation depends on the chosen metrics; different validation scores may yield different outcomes. For example, TOFU shouldn’t be assessed solely by FQ and MU. This is a minor point, though, as the paper also demonstrates generalization across other metrics.

Improvements are minor, which is acceptable since I view this work as more interpretability-oriented than method-oriented. The finding that different layers have different effects on unlearning is interesting, though related ideas have been explored in [1–4].

It would also be informative to evaluate on the RESTOR benchmark [5], which focuses on data-level unlearning and restorative ability. Many methods perform poorly there; I’m curious whether the proposed approach can actually improve those results, rather than merely offering more surgical edits.


------

[1] Meng, Kevin, et al. "Mass-editing memory in a transformer." arXiv preprint arXiv:2210.07229 (2022).

[2] Gupta, Akshat, Anurag Rao, and Gopala Anumanchipalli. "Model editing at scale leads to gradual and catastrophic forgetting." arXiv preprint arXiv:2401.07453 (2024).

[3] Basu, Samyadeep, et al. "On mechanistic knowledge localization in text-to-image generative models." Forty-first International Conference on Machine Learning. 2024.

[4] Zarei, Arman, et al. "Localizing Knowledge in Diffusion Transformers." arXiv preprint arXiv:2505.18832 (2025).

[5] Rezaei, Keivan, et al. "RESTOR: Knowledge Recovery in Machine Unlearning." arXiv preprint arXiv:2411.00204 (2024).

**Questions:**

They are discussed in Weakness section.

---

> ### Author Response · Authors · 2025-11-25
> **Response to Reviewer scYt [1/3]**
>
> We sincerely thank the reviewer for the time and effort spent reviewing our paper and for acknowledging the compelling idea of localizing the unlearning effects, convincing and comprehensive experiments, and no major issues. Below, we address the raised questions and concerns in detail:
>
> > **W1:** I don’t see major issues with this work, aside from the choice of k, which may be task- and model-dependent. Insight into how to choose k and an analysis of robustness to k would be valuable. Relatedly, the validation depends on the chosen metrics; different validation scores may yield different outcomes. For example, TOFU shouldn’t be assessed solely by FQ and MU. This is a minor point, though, as the paper also demonstrates generalization across other metrics.
>
> **A1:** We sincerely appreciate the reviewer’s positive assessment, and agree that **providing insights on selecting k will make the method more practical for broader use.** Regarding the **analysis of the robustness to $k$**, we have presented in the left of Figure 6. The result curve of choosing different numbers of replacement layers shows that CRU performs consistently well for a broad range of $k$ (e.g., 4-9), rather than being sensitive to a single point. This stability arises because of an empirical observation that MU-optimal and FQ-optimal layers tend to overlap, making the rank-sum selection with a large enough $k$ relatively insensitive to the exact cutoff. For the practical guidance, we may recommend selecting $k$ via early stopping on a monotonic trade-off frontier (i.e., stopping when MU recovery saturates while FQ remains above a threshold). This automatic heuristic yields performance comparable to an exhaustive sweep, further reinforcing the robustness of CRU. We will include the above additional discussion in our revised version, these additions will strengthen the clarity and usability of the method.
>
> **Table 1. Numerical results from the left of Figure 6.**
> |  | unlearn | retain  | 0| 1| 2| 3| 4| 5| 6| 7| 8| 9| 10| 11| 12| 13| 14| 15| 16|
> |----|----|-----|-----|----|-----|---|------|-----|-----|-----|----|-----|-----|-----|----|------|----|----|-----|
> | MU | 0.2203  | 0.4386  | **0.5923**  | **0.5864**  | **0.5791**  | **0.5598**  | **0.5504**  | **0.5345**  | **0.5146**  | **0.5033**  | **0.4821**  | **0.4604**  | 0.4214  | 0.3902  | 0.3669  | 0.3324  | 0.3110  | 0.2879  | 0.2203  |
> | FQ | -2.3448 | -2.2030 | -8.7883 | -7.5302 | -5.9294 | -2.6391 | **-2.0646** | **-2.0646** | **-1.9297** | **-1.9297** | **-0.8779** | **-0.7804** | **-0.7804** | **-1.1950** | **-1.1950** | **-2.0646** | -2.2030 | -2.3448 | -2.3448 |
>
> We also appreciate the reviewer’s comment **regarding metric dependence**. While incorporating additional validation metrics can provide a more comprehensive picture, we should note it also substantially increases computational cost, since each candidate layer must be re-evaluated across multiple scoring pipelines. In practice, we recommend using MU and FQ, which is not only due to the widely adoption among the unlearning literature, but also due to the compound scores that reflect a comprehensive performance on multiple submetrics (e.g., factuality, coherence, truthfulness, etc.), which jointly reflect the two central objectives of unlearning, i.e., preserving model utility and ensuring forgetting quality. Our **supplementary experiments (in the following table) also show using other evaluations like the ES score for CRU**, and the results remain stable even when these broader metrics are included, suggesting that MU and FQ already act as reliable, representative summaries. We will clarify this recommendation and the computational trade-off in the revised manuscript.
>
>
> **Table 2 (Part-I). Using other evaluation metrics beyond MU and FQ for CRU.**
>
> | Model| ES-exact|| ES-perturb || MU| FQ|
> |-----|-----|----|---|---|----|-----|
> || retain| unlearn | retain| unlearn |||
> | **Llama-3.2-1B-Instruct**  | ||||||
> | Original| 0.7642| 0.7592  | 0.3286| 0.3574  | 0.5914 | -9.0517|
> | GA| 0.0332| 0.0282  | 0.0265| 0.0281  | 0.0000 | -104.7672 |
> | GA (MU + FQ)| 0.2318| 0.0689  | 0.1362| 0.0554  | 0.5426 | -2.7916   |
> | GA (MU + FQ + ES)  | 0.7073| 0.5407  | 0.3414| 0.2876  | 0.5874 | -9.8654|
> | NPO| 0.0339| 0.0287  | 0.0270| 0.0281  | 0.2203 | -2.3448|
> | NPO (MU + FQ)| 0.2938| 0.0981  | 0.1972| 0.0851  | 0.5504 | -2.0646   |
> | NPO (MU + FQ + ES) | 0.0977| 0.0495  | 0.069| 0.0473  | 0.4873 | -1.3084  |
> | **Llama-2-7b-chat-hf**| ||||||
> | Original| 0.9867| 0.9774  | 0.6018| 0.5366  | 0.6192 | -10.1446|
> | GA| 0.0278| 0.0235  | 0.0220| 0.0235  | 0.0000 | -104.7672 |
> | GA (MU + FQ)| 0.4924| 0.1131  | 0.2801| 0.0687  | 0.6019 | -5.2994   |
> | GA (MU + FQ + ES)  | 0.1476| 0.0507  | 0.1415     | 0.0379  | 0.5582 | -4.1383   |
> | NPO| 0.0285| 0.0243  | 0.0233| 0.0238  | 0.0479 | -0.4366   |
> | NPO (MU + FQ)| 0.0355| 0.0719  | 0.0309| 0.0252  | 0.5296 | -1.9297   |
> | NPO (MU + FQ + ES) | 0.0317| 0.0275  | 0.0256     | 0.0238  | 0.4713 | -1.0854   |

---

> > ### Author Response · Authors · 2025-11-25
> > **Response to Reviewer scYt [2/3]**
> >
> > **Table 2 (Part-II). Using other evaluation metrics beyond MU and FQ for CRU.**
> >
> > | Model| ES-exact|| ES-perturb || MU| FQ|
> > |-----|------|----|---|---|----|-----|
> > | **Phi-3.5-mini-instruct**   | ||||||
> > | Original| 0.9148| 0.9598  | 0.4593     | 0.4078  | 0.6648 | -7.2902   |
> > | GA| 0.0272| 0.0233  | 0.0215     | 0.0233  | 0.0000 | -104.7672 |
> > | GA (MU + FQ)| 0.3117| 0.1959  | 0.1335     | 0.1636  | 0.6245 | -4.8978   |
> > | GA (MU + FQ + ES)  | 0.7146| 0.5910  | 0.3077     | 0.2604  | 0.6111 | -6.5928   |
> > | NPO| 0.0272| 0.0233  | 0.0215     | 0.0233  | 0.2874 | -3.4365   |
> > | NPO (MU + FQ)| 0.0272| 0.0233  | 0.0215     | 0.0233  | 0.4977 | -0.9796   |
> > | NPO (MU + FQ + ES) | 0.0272| 0.0233  | 0.0215     | 0.0233  | 0.5218 | -2.6391   |
> >
> >
> >
> >
> >
> >
> > > **W2:** The finding that different layers have different effects on unlearning is interesting, though related ideas have been explored in [1–4].
> >
> >
> > **A2:** We sincerely thank the reviewer for acknowledging the interpretability-driven nature of the work, and also for bringing us those great works [1-4]. We agree that **CRU shares the conceptual connections with related ideas in editing and localization** studies [1–4], but our **contribution targets a different question under a different view**: how unlearning modifies internal representations and how those disruptions happen across depth. Whereas **prior work examines where knowledge is stored or how to rewrite/edit knowledge**, our framework introduces latent knowledge fragility as a new lens or unlearning-induced representation disruption. Unlike editing methods such as MEMIT or ROME [1–2], or feature-localization approaches applied to vision/diffusion models [3–4], **CRU does not require identifying feature circuits, semantic directions, or pre-defined target facts.** Instead, it performs a **training-free, post-hoc component replacement** between the unlearned and original models to reveal how different layers absorb or amplify unlearning updates, which is an analysis that, to our knowledge, has not been examined in prior literature.
> >
> >
> > While the numerical improvements in MU/FQ are moderate, we would like to clarify **two implications of the overall improvement to demonstrate its significance**. On the one hand, our CRU is mainly designed for improving the tradeoff of unlearning, where previously many unlearning methods can suffer from. Thus, the numerical results of MU and FQ may not be independently compared with each other's highest value, but consider both to measure the tradeoff balance. For example, although CRU does not achieve numerically higher MU with some baseline, the FQ is significantly better in numerical comparison, and vice versa. On the other hand, given that both MU and FQ compound scores aggregating multiple submetrics (e.g., factuality, coherence, truthfulness, etc.), **even small gains indicate significant improvements across all components**, e.g., ~0.03 in MU or 0.4 in FQ. Furthermore, **similar effect sizes are common in prior unlearning literature [1,2]** when measuring the retention–forgetting trade-off performance. And our CRU performs generally well across three benchmarks (TOFU, MUSE, WMDP) and across four distinct model families (Llama2, Llama3.2, Phi-3.5, Zephyr).
> >
> >
> > **We will incorporate a more explicit positioning of our method within the broader landscape of mechanistic interpretability and knowledge localization**, including detailed discussion of [1–4] and also RESTOR [5]. We appreciate this suggestion as it strengthens the conceptual framing of our work.
> >
> >
> > [1] Exploring Criteria of Loss Reweighting to Enhance LLM Unlearning. ICML 2025.
> >
> > [2] Alternate Preference Optimization for Unlearning Factual Knowledge in Large Language Models. ACL 2025.

---

> > > ### Author Response · Authors · 2025-11-25
> > > **Response to Reviewer scYt [3/3]**
> > >
> > > > **W3:** It would also be informative to evaluate on the RESTOR benchmark [5], which focuses on data-level unlearning and restorative ability. Many methods perform poorly there; I’m curious whether the proposed approach can actually improve those results, rather than merely offering more surgical edits.
> > >
> > > **A2:** We appreciate the reviewer’s suggestion and fully agree that **considering RESTOR can be an informative evaluation on understanding the data-level restorative ability**.  Following the reviewer’s suggestion, **we have conducted preliminary experiments on RESTOR and summarized the results in the following table.** Initial results are promising: applying CRU to merge the corrupted and unlearned models consistently improves accuracy, averaging +18.9 percentage points over GA, +7.1 over KL, and +2.65 over NPO. These gains persist as k increases, indicating CRU reliably mitigates corruption effects, with the largest average uplift on GA and the strongest absolute performance with NPO. These early results give us confidence that CRU adds genuine value even in the challenging RESTOR setting. We will include RESTOR results in the revised appendix and integrate a discussion in the main text explaining how CRU’s layer-wise replacement complements the goals of the RESTOR benchmark. We appreciate the reviewer’s suggestion again for the opportunity of verifying our method in other benchmarks.
> > >
> > > **Table 3. Performance comparison of Unlearning using the RESTOR benchmark.**
> > >
> > > | Clean |       | Corrupt | GA    | GA + CRU | KL    | KL + CRU | NPO   | NPO + CRU |
> > > |-------|-------|---------|-------|----------|-------|----------|-------|-----------|
> > > | 72.20 | k = 1 | 68.56   | 51.09 | **78.17**    | 65.94 | **78.60**    | 76.00 | **79.04**     |
> > > |       | k = 2 | 67.69   | 55.90 | **74.24**    | 69.43 | **75.55**    | 76.86 | **79.04**     |
> > > |       | k = 3 | 65.50   | 48.91 | **73.36**    | 69.00 | **73.36**    | 74.24 | **77.29**     |
> > > |       | k = 4 | 59.83   | 51.09 | **65.94**    | 66.38 | **73.36**    | 74.55 | **76.89**     |
> > > |       | k = 5 | 57.64   | 57.20 | **66.81**    | 66.38 | **71.62**    | 74.24 | **76.86**     |

---

### Official Review · Reviewer_Hd8d · 2025-11-01

**Soundness:** 3
**Presentation:** 3
**Contribution:** 2
**Rating:** 4
**Confidence:** 3

**Summary:**

This paper investigates a fundamental yet underexplored question in the field of machine unlearning for large language models (LLMs), how internal representations are structurally affected during unlearning. The authors introduce the concept of latent knowledge fragility, which captures how certain layers in LLMs are disproportionately damaged when removing specific knowledge. Through rigorous layer-wise analyses, they reveal that middle layers are especially fragile, leading to overall performance degradation.
To mitigate this problem, the paper proposes Component-wise Replacement Unlearning (CRU), a post-hoc, training-free method that selectively restores fragile components of the model from the original parameters. By identifying critical layers based on Model Utility (MU) and Forget Quality (FQ) scores, CRU reassembles a “patched” model that maintains the unlearning effect while recovering general capabilities.

**Strengths:**

1. The introduction of latent knowledge fragility provides a new theoretical lens to understand how unlearning interacts with internal representation geometry. Theoretical results (e.g., Proposition 1) give a principled basis for quantifying representational drift via CKA similarity.

2. The study systematically quantifies per-layer influence on model retention and forgetting, offering concrete insights into how unlearning propagates across the transformer hierarchy.

3. CRU requires no retraining or gradient computation, operating entirely post-hoc. Despite its simplicity, it achieves a superior trade-off between FQ and MU across various benchmarks (TOFU, MUSE, WMDP) and model scales (llama-3.2 1B & 3B, llama-2 7B, phi-3.5-mini, zephyr-7B).

**Weaknesses:**

1. Although evaluated empirically, restoring original parameters from the base model always entails some risk of reviving forgotten knowledge, especially if the targeted information is localized within the restored layers. While each layer inherently contains a mixture of heterogeneous knowledge, the proposed method restores certain layers as a whole, overlooking the fact that diverse types of information are entangled within a single layer. Although the method demonstrates improved performance over baselines on several unlearning benchmarks, it remains uncertain whether such advantages would generalize to more comprehensive or heterogeneous datasets.

2. A deeper comparison with the more straightforward and practical "structural freezing" approach is necessary. In Lines 314–317, the paper discusses the comparison between replacement and structural freezing. However, it is unclear which layers were frozen during the structural freezing experiments. If only the middle layers were frozen, further analysis would be necessary. The observation that middle layers exhibit larger representation differences was derived after unlearning; thus, freezing those same layers during unlearning may not be a valid comparison, as it is an independent analysis. A more comprehensive analysis (e.g., freezing up to varying depth levels, or freezing varying depth layers) would strengthen the validity of the comparison.

3. The formal analysis (Proposition 1) characterizes lower bounds of representational drift but does not model causal interactions between layers or nonlinear dependencies within the transformer stack. The paper also introduces several formal definitions (e.g., patched model, component-wise partitioner), which establish the theoretical foundation of the proposed method. However, these formulations mainly serve as notational formalities rather than substantial theoretical contributions.

**Questions:**

See weaknesses

---

> ### Author Response · Authors · 2025-11-25
> **Response to Reviewer Hd8d [1/3]**
>
> We sincerely thank the reviewer for the time and effort spent reviewing our paper and for acknowledging the new lens of exploring internal representation, systematic quantification, propagation insights, lightweight CRU, and comprehensive experiments. Below, we address the raised questions and concerns in detail:
>
> **W1:** Although evaluated empirically, restoring original parameters from the base model always entails some risk of reviving forgotten knowledge, especially if the targeted information is localized within the restored layers. While each layer inherently contains a mixture of heterogeneous knowledge, the proposed method restores certain layers as a whole, overlooking the fact that diverse types of information are entangled within a single layer. Although the method demonstrates improved performance over baselines on several unlearning benchmarks, it remains uncertain whether such advantages would generalize to more comprehensive or heterogeneous datasets.
>
> **A1:** We appreciate the reviewer’s thoughtful comments. **Regarding the risk of reviving forgotten knowledge**, we would like to clarify that **CRU is applied after the unlearning procedure** has already been conducted to forget the targeted information. Even when restoring a layer from the original model, the critical forward computations responsible for recalling or generating the forgotten content have already been disrupted by updates in other parts of the network. As a result, **restoring a “fragile” layer does not reconstruct the full forward path required for the forgotten output to re-emerge**. This is **widely supported by our experiments across all benchmarks following the literature** (e.g., TOFU [1], MUSE [2], and WMDP [3]): after applying CRU (in Table 1-3), FQ remains comparable to (and also slightly better than) those unlearning baseline methods, **indicating that no meaningful reactivation** of the forgotten knowledge occurs in practice. In other words, the restored layers stabilize general capabilities but do not reconnect the representational pathways required to regenerate the removed content.
> **Regarding generalization across unlearning datasets**, we agree that different datasets may exhibit different knowledge distributions, but **our evaluation already spans the most widely used and heterogeneous benchmarks in the unlearning literature [4,5,6]**, e.g., TOFU (fictional entities), MUSE (real-world news/books), and WMDP (large-scale domain-specific knowledge). These collectively cover a broad range of heterogeneous knowledge, aligning with the standard evaluation protocol used by recent unlearning methods (e.g., NPO, WTNPO, FLAT, RMU). CRU shows robust improvements across all of these settings, suggesting that its benefits are **not tied to a particular dataset or knowledge type**. While even broader evaluations are always valuable, we believe these three diverse benchmarks represent the current scope of unlearning research, and our results demonstrate robust generality within this landscape. We will clarify these points in the revision.
>
> [1] TOFU: A Task of Fictitious Unlearning for LLMs. COLM 2024.
>
> [2] MUSE: Machine Unlearning Six-Way Evaluation for Language Models. ICLR 2025.
>
> [3] The WMDP Benchmark: Measuring and Reducing Malicious Use With Unlearning. ICML 2024.
>
> [4] Negative preference optimization: From catastrophic collapse to effective unlearning. COLM 2024.
>
> [5] Exploring Criteria of Loss Reweighting to Enhance LLM Unlearning. ICML 2025.
>
> [6] Alternate Preference Optimization for Unlearning Factual Knowledge in Large Language Models. ACL 2025.

---

> > ### Author Response · Authors · 2025-11-25
> > **Response to Reviewer Hd8d [2/3]**
> >
> > **W2:** A deeper comparison with the more straightforward and practical "structural freezing" approach is necessary. In Lines 314–317, the paper discusses the comparison between replacement and structural freezing. However, it is unclear which layers were frozen during the structural freezing experiments. If only the middle layers were frozen, further analysis would be necessary. The observation that middle layers exhibit larger representation differences was derived after unlearning; thus, freezing those same layers during unlearning may not be a valid comparison, as it is an independent analysis. A more comprehensive analysis (e.g., freezing up to varying depth levels, or freezing varying depth layers) would strengthen the validity of the comparison.
> >
> > **A2:** We thank the reviewer for the thoughtful suggestion. We would like to explain that **our motivation for comparing** component-wise replacement with structural freezing is to **demonstrate the rationality of replacement choice** of those layers identified by patching, so we keep the same layers. Although both operate at the layer level, they fundamentally intervene at different stages of unlearning. **CRU modifies the model after unlearning has taken place**, repairing layers whose representations were overly disrupted. In contrast, **structural freezing constrains the optimization process itself**, preventing certain layers from updating. This restriction alters the unlearning dynamics, often limiting the model’s ability to achieve unlearning targets. Moreover, freezing assumes a priori knowledge of which layers should be protected, whereas **CRU relies entirely on post-hoc validation and makes no structural assumptions**. For this reason, structural freezing is not as good as CRU performs in improving the unlearning tradeoff. While we understand the reviewer’s intention for conducting a valid comparison without considering the same layer selected by CRU’s guidance, it would be out of the scope of original comparison.
> >
> > Empirically, to address the reviewer’s concern, we further conducted **layer-wise structural freezing experiments** to ensure a valid comparison without emphasizing the same layer-use. In the following table, we show the comparison of CRU with structural freezing across three different models, the results generally verify that **CRU can achieve the favorable unlearning tradeoff with higher MU and FQ**. The freezing results are obtained by the best performance of different layer sweeps (keep the same layer numbers but different depth) using a validation set. Across all configurations (like different depth levels), structural freezing did not offer a better trade-off or even fail on the basis of GA, **indicating the structural freezing may still disrupt the original optimization dynamics that constrains the representation expressiveness during unlearning**. These comprehensive freezing experiments confirm that the two approaches serve different purposes and that **CRU’s post-hoc replacement is more lightweight and efficient way** as it doesn’t require additional training. We appreciate the reviewer’s suggestion and will add extended freezing comparison to the appendix.
> >
> > **Table 1. Experiments on structural freezing of model layers.**
> > | Model| ES-exact  || ES-perturb |  | MU  | FQ  |
> > |----|----|----|----|----|----|---|
> > || retain| unlearn | retain| unlearn |||
> > | Llama-3.2-3B-Instruct| | ||| ||
> > | Original | 0.9013 | 0.9291| 0.4241| 0.4111| 0.6579 | -5.7157   |
> > | GA | 0.0332 | 0.0282| 0.0265| 0.0281  | 0.0000 | -104.7672 |
> > | GA (Ours)| 0.7251 | 0.2117| 0.3677| 0.1215  |**0.6691** | **-3.2700**   |
> > | GA (Freeze)| 0.0332 | 0.0282  | 0.0265| 0.0281  | 0.0000 | -104.7672 |
> > | NPO | 0.0336 | 0.0287  | 0.0271| 0.0281  | 0.0347 | -7.0539   |
> > | NPO (Ours)| 0.0999  | 0.0719  | 0.1058| 0.0846  | **0.5117** | **-1.5462**   |
> > | NPO (Freeze) | 0.0364  | 0.0292  | 0.0295| 0.0281  | 0.4784 | -1.5854  |
> > |  Llama-2-7b-chat-hf | | ||| ||
> > | Original | 0.9867| 0.9774| 0.6018| 0.5366  | 0.6192 | -10.1446  |
> > | GA | 0.0278| 0.0235| 0.0220| 0.0235  | 0.0000 | -104.7672 |
> > | GA (Ours)| 0.4924 | 0.1131| 0.2801  | 0.0687  | **0.6019** | **-5.2994**  |
> > | GA (Freeze)| 0.0278 | 0.0235| 0.0220  | 0.0235  | 0.0000 | -104.7672 |
> > | NPO | 0.0285 | 0.0243  | 0.0233| 0.0238  | 0.0479 | -0.4366   |
> > | NPO (Ours)| 0.0355  | 0.0719  | 0.0309  | 0.0252  | **0.5296** |**-1.9297** |
> > | NPO (Freeze) | 0.0314  | 0.0275  | 0.0276 | 0.0261 | 0.3913 | -3.1070 |
> > | Phi-3.5-mini-instruct ||||| ||
> > | Original | 0.9148 | 0.9598  | 0.4593| 0.4078  | 0.6648 | -7.2902   |
> > | GA | 0.0272  | 0.0233  | 0.0215| 0.0233  | 0.0000 | -104.7672 |
> > | GA (Ours) | 0.3117 | 0.1959| 0.1335| 0.1636  | **0.6245** | -4.8978   |
> > | GA (Freeze)| 0.0272 | 0.0233| 0.0215| 0.0233  | 0.0398 | **-0.3638**|
> > | NPO   | 0.0272 | 0.0233  | 0.0215| 0.0233  | 0.2874 | -3.4365   |
> > | NPO (Ours)   | 0.0272| 0.0233  | 0.0215| 0.0233  | **0.4977** | **-0.9796**   |
> > | NPO (Freeze) | 0.0272 | 0.0233  | 0.0215| 0.0233  | 0.3983 | -1.3084|

---

> > > ### Author Response · Authors · 2025-11-25
> > > **Response to Reviewer Hd8d [3/3]**
> > >
> > > **W3:** The formal analysis (Proposition 1) characterizes lower bounds of representational drift but does not model causal interactions between layers or nonlinear dependencies within the transformer stack. The paper also introduces several formal definitions (e.g., patched model, component-wise partitioner), which establish the theoretical foundation of the proposed method. However, these formulations mainly serve as notational formalities rather than substantial theoretical contributions.
> > >
> > > **A3:** We appreciate the reviewer’s perspective and agree that Proposition 1 does not attempt to model the full causal or nonlinear structure of transformers, as **doing so remains an open and extremely challenging problem** in mechanistic interpretability (as indicated in prior works [1,2]). Our goal in Proposition 1 is intentionally more focused: **to provide a quantitative link between representation drift and output-level deviations** (e.g., the MU/FQ changes), establishing a lower-bound relation that grounds the notion of latent knowledge fragility. The accompanying formal definitions, like the patched model and component-wise partitioner, are not intended as theoretical innovations on their own, but they play an important role in **formalizing CRU’s modular operations in a general, architecture-agnostic manner.** This allows us to precisely define how component-wise restoration is constructed and enables the method to extend cleanly to layers, MLPs, or other structural units.
> > >
> > > At the empirical level, **Proposition 1’s predictions align closely with our observations (as shown in the top part of Figure 4):** components showing larger CKA drift generally align with larger MU degradation when patched, and our concept-intention experiments also demonstrate that semantic abstractions (e.g., hallucination tendency, corrigibility) are disproportionately disrupted in the same regions. This **comprehensive alignment across multiple models, multiple datasets, and multiple patching evaluations** reinforces that the theoretical link, while lower-bound and not a causal model, is meaningful and explanatory for the scope of this work. We agree that a full causal or nonlinear analysis of layer interactions would be valuable future work, and we will revise the text to clarify the intended scope of Proposition 1 and how it complements our empirical findings.
> > >
> > > [1] A Mathematical Framework for Transformer Circuits. Anthropic 2021.
> > >
> > > [2] Emergent Symbolic Mechanisms Support Abstract Reasoning in Large Language Models. ICLR 2025.

---

### Official Review · Reviewer_G64q · 2025-11-01

**Soundness:** 3
**Presentation:** 2
**Contribution:** 3
**Rating:** 4
**Confidence:** 4

**Summary:**

This paper investigates how different layers of LLMs are affected during unlearning. The authors introduce the concept of latent knowledge fragility, which measures the sensitivity of internal representations under unlearning updates. Using layer-wise model patching, they analyze how shallow, middle, and deep transformer layers behave when unlearned. They observe that middle layers are particularly fragile and play a central role in balancing knowledge removal and model utility.

Based on this insight, the authors propose Component-wise Replacement Unlearning (CRU) — a lightweight, post-hoc framework that selectively restores fragile layers from the original model (without additional fine-tuning). They claim that CRU achieves a better trade-off between forgetting and retention compared with gradient-based methods, across benchmarks like TOFU, MUSE, and WMDP.

**Strengths:**

1. The paper focuses on analyzing unlearning effects from the viewpoint of internal representations, which is less explored in existing LLM unlearning literature.
2. CRU avoids extra training and introduces a modular, interpretable post-hoc procedure that could be useful for debugging or analysis.
3. Results are reported on multiple models and datasets, showing stable improvement in utility-forgetting trade-off.

**Weaknesses:**

1. The selection process of fragile layers (via top-k scores) is heuristic. It’s unclear why summing ranks of FQ and MU is an optimal or principled criterion. No ablation is presented to show sensitivity to k, nor justification for why layer replacement (discrete switch) is better than, say, soft interpolation or fine-tuning (though mentioned briefly, not analyzed rigorously).
2. The work claims novelty in “component-wise patching,” yet similar ideas exist in interchange interventions, representation surgery, and parameter-efficient editing. The novelty should be further clarified.
3. Reported improvements (Table 1–3) are sometimes numerically small and may not be statistically significant. This raises the concern of the proposed method.
4. The observation that middle layers are “fragile” may correlate with unlearning sensitivity but doesn’t prove causality, e.g., deeper layers might also appear fragile under different architectures or datasets. The U-shaped pattern (Figure 3) could simply reflect gradient distribution rather than intrinsic semantic hierarchy.

**Questions:**

1. How exactly is “latent knowledge fragility” quantitatively defined? Is it directly proportional to the observed drop in model utility, or is there a separate latent-space measure?
2. Why sum the rank of MU and FQ to get the patching score? Would a weighted or Pareto-based method yield different results?
3. How do the authors ensure that observed middle-layer fragility is not a byproduct of model scaling or optimizer dynamics rather than inherent semantic abstraction?
4. Have the authors tested CRU on models larger than 7B (e.g., 13B or 70B)? Does the same “middle-layer fragility” pattern persist across scales?
5. Why are methods like MEMIT[1], or KnowledgeEditor[2] not included as baselines, given their similarity to modular patching?
6. CRU is said to be “lightweight,” yet it involves multiple forward validations per layer. What is the computational overhead compared to single-pass fine-tuning methods?
7. Some Tables and Figures are dense; figure captions do not clearly indicate what metrics or settings are used (Figure 4).

[1] Mass-Editing Memory in a Transformer
[2] Editing Factual Knowledge in Language Models

---

> ### Author Response · Authors · 2025-11-25
> **Response to Reviewer G64q [1/6]**
>
> We sincerely thank the reviewer for the time and effort spent reviewing our paper and for acknowledging the significance of exploring internal representation, training-free CRU, and stable improvement on extensive experiments. Below, we address the raised questions and concerns in detail:
>
>
> > **W1.** The selection process of fragile layers (via top-k scores) is heuristic. It’s unclear why summing ranks of FQ and MU is an optimal or principled criterion. No ablation is presented to show sensitivity to k, nor justification for why layer replacement (discrete switch) is better than, say, soft interpolation or fine-tuning (though mentioned briefly, not analyzed rigorously).
>
>
> **A1:** We appreciate the reviewer’s comments regarding the top-k selection design and its properties. We would like to clarify the behind **rationality of such consideration** and the **corresponding verifications**. First, regarding the top-k selection, we would explain that **our design is an effectiveness-practicality balanced consideration**, which is principally guided by the multi-objective selection under monotonicity constraints while the objective scales differ. As we aim to improve the tradeoff of unlearning towards better utility preservation and achieve effective forgetting, the sum of ranks capture the component-wise property well and using ranks instead of raw metrics makes the procedure robust to heterogeneous scales across MU/FQ, which is also empirically grounded in the observed semantic hierarchy in transformers (as the structurally aligned evidence in Figure 4). While we should acknowledge that the top-k individually selected layers **may not always represent the globally optimal combination**, but **the space of all k-layer combinations grows combinatorially (e.g., consider selecting 7 from 32 layers 7b model, it has (32,7)=3365859 types)**, introducing significant complexity for exhaustive search. So, top-k selection represents a reasonable and practical design for enhancing the tradeoff by our component-wise replacement.
>
> Regarding the **corresponding verifications** of replacement, we thank the reviewer’s question, and **would further improve the visibility and detailed discussion** for **sensitivity to the choice of $k$ (the left of Figure 6)**, comparison with **soft interpolation (the right of Figure 6)**, and comparison with **fine-tuning (the right bottom of Figure 4)**. Specifically, we have provided ablation on different choices of $k$, the left of Figure 6 demonstrates that performance is robust over a broad range (e.g., 4-9), indicating a stable optimizing space instead of several sensitive points to achieve a better trade-off than the baselines (as the following table shown); We further compare the layer-wise replacement of CRU with the soft intervention of unlearned model with the original model in the right of Figure 6, the results verified ours consistently dominates soft interpretation across the entire MU–FQ frontier, as the latter didn’t consider the layer-wise heterogeneous. If the reviewer intends to compare the soft interpolation of different layers, we would note that it further introduces extra computation complexity for handling the proper interpolation parameters.
>
> **Table 1. Numerical results from the left of Figure 6.**
> |   | unlearn | retain  | 0| 1| 2| 3| 4| 5| 6| 7| 8| 9| 10| 11| 12| 13| 14| 15| 16|
> |----|-----|----|-----|-----|----|-----|----|---|-----|-----|-----|-----|----|------|---|-----|----|-----|----|
> | MU | 0.2203  | 0.4386| **0.5923**| **0.5864**| **0.5791**| **0.5598**| **0.5504**| **0.5345**| **0.5146**| **0.5033**| **0.4821**| **0.4604**| 0.4214| 0.3902| 0.3669| 0.3324| 0.3110| 0.2879| 0.2203  |
> | FQ| -2.3448 | -2.2030 | -8.7883 | -7.5302 | -5.9294 | -2.6391 | **-2.0646** | **-2.0646** | **-1.9297** | **-1.9297** | **-0.8779** | **-0.7804**| **-0.7804**| **-1.1950**| **-1.1950**| **-2.0646**| -2.2030 | -2.3448 | -2.3448 |
>
> As for the comparison to fine-tuning, we kindly refer to the right bottom of Figure 4, where **We compare the layer-wise replacement with the structural freezing for fine-tuning.** The former is a post-hoc operation: we first complete unlearning, then selectively restore specific layers with original parameters based on validation results. **These restored layers were never exposed to unlearning updates**. While the latter happens during unlearning, where certain layers are prevented from updating (e.g., frozen during unlearning). However, **this often disrupts learning dynamics (e.g., affects updates on other layers)**, leading to unstable gradients and degraded convergence of the unlearning, especially when the frozen layers are critical for routing representations. As a result, although both preserve original parameters in selected layers, **replacement avoids interfering with the optimization process for unlearning while freezing obtains a different unlearned model**, which leads to consistently better trade-offs of CRU as demonstrated in the Figure.

---

> > ### Author Response · Authors · 2025-11-25
> > **Response to Reviewer G64q [2/6]**
> >
> > > **W2.** The work claims novelty in “component-wise patching,” yet similar ideas exist in interchange interventions, representation surgery, and parameter-efficient editing. The novelty should be further clarified.
> >
> > **A2:** We thank the reviewer for pointing out the conceptual connections to interchange interventions, representation surgery, and parameter-efficient editing. While CRU is inspired by the broader family of modular and interpretability-oriented techniques, our contribution is not a direct reuse. but **a targeted and novel contribution** to link them to the unlearning scenario **under a new perspective to explore the unlearning effect** via post-hoc patching. We would like to clarify the novelty along each dimension below:
> >
> > - **Interchange interventions** are primarily diagnostic methods designed to probe where specific information is encoded by swapping activations to measure causal effects. In contrast, CRU introduces the **post-hoc component-wise substitution by considering the unlearned and original models**, which to our knowledge, is the first time to understand the unlearning effects via the component recombination. Specifically, we want to note **two major differences:** 1) we keep the unlearning behavior unaffected, which helps us to reveal the unlearning dynamics as a whole to different parts of the model's internal representation; 2) not local intervention with structural priors or extra optimization, making it general to different components and models. **Those two points are also applicable to the latter comparisons.**
> > - **Representation surgery** methods attempt to manually alter a specific known concept by inserting or modifying feature subspaces. These approaches require a predefined semantic direction or a model-specific feature localization. CRU does not rely on any concept annotation or pre-identified circuit. Instead, **our notion of latent knowledge fragility provides a general framework to quantify unlearning-induced disruption in internal representations**, without assuming knowledge of where particular concepts live. This produces a fundamentally different use case and a broader form of component manipulation that operates in different levels, not specifically-designed subspaces.
> > - **Parameter-efficient editing** aims to rewrite or insert knowledge with minimal parameter change. They optimize for enforcing a new fact, often by modifying very specific layers or neurons. CRU is distinct in both objective and mechanism: it does not edit or insert pre-defined information, performs no optimization, and **introduces training-free, model-wide restoration** based on post-hoc validation. Moreover, editing focuses on local factual corrections, whereas CRU directly **targets global retention–forgetting dynamics in unlearning**.
> >
> > Building on these perspectives, our work introduces the **latent knowledge fragility framework**, an unlearning-specific lens for analyzing the unlearning effect propagation across the model internals. CRU operationalizes this framework to provide a **new, modular, post-hoc solution** that selectively restores fragile components. To our knowledge, no prior method systematically explores the component-level restoration with MU/FQ-guided selection to optimize the unlearning trade-off. In summary, while CRU stands on the shoulders of prior interpretability and modularity work, its **conceptual focus (unlearning-induced fragility), objective (balancing forgetting and retention), and mechanism (training-free post-hoc restoration)** are unique. We believe this constitutes a novel and meaningful contribution that bridges interpretability views with practical unlearning.

---

> > > ### Author Response · Authors · 2025-11-25
> > > **Response to Reviewer G64q [3/6]**
> > >
> > > > **W3.** Reported improvements (Table 1–3) are sometimes numerically small and may not be statistically significant. This raises the concern of the proposed method.
> > >
> > > **A3:** We appreciate the reviewer’s concern. We would like to clarify **two implications of the overall improvement to demonstrate its significance**. On the one hand, our CRU is mainly designed for improving the tradeoff of unlearning, where previously many unlearning methods can suffer from. Thus, we would like to point out that the numerical results of MU and FQ may not be independently compared with each other's highest value, but consider both to measure the tradeoff balance. For example, although CRU does not achieve numerically higher MU with some baseline, the FQ is significantly better in numerical comparison, and vice versa. On the other hand, given that both MU and FQ compound scores aggregating multiple submetrics (e.g., factuality, coherence, truthfulness, etc.), **even small gains indicate significant improvements across all components**, e.g., ~0.03 in MU or 0.4 in FQ. Furthermore, **similar effect sizes are common in prior unlearning literature [1,2]** when measuring the retention–forgetting trade-off performance. For example, TOFU-based evaluations in WTNPO [1] also report MU improvements of a similar numerical range, as meaningful given the compound score evaluated across different subaspects. Within this context, the improvements achieved by CRU are well aligned with the scale and expectations of existing unlearning studies.
> > >
> > > [1] Exploring Criteria of Loss Reweighting to Enhance LLM Unlearning. ICML 2025.
> > > [2] Alternate Preference Optimization for Unlearning Factual Knowledge in Large Language Models. ACL 2025.
> > >
> > > > **W4.** The observation that middle layers are “fragile” may correlate with unlearning sensitivity but doesn’t prove causality, e.g., deeper layers might also appear fragile under different architectures or datasets. The U-shaped pattern (Figure 3) could simply reflect gradient distribution rather than intrinsic semantic hierarchy.
> > >
> > > **A4:** We appreciate the reviewer’s thoughtful observation. **We agree that the exact location of fragile layers can vary** with architecture/dataset, and **we do not intend to overgeneralize** middle-layer fragility as a universal or causal rule. **Our goal is instead to provide a lens and a tool**, via latent knowledge fragility and component-wise patching, **that allows practitioners to empirically characterize where unlearning exerts the strongest representational disruption for any given model or dataset for unlearning. In this sense, our argument is not the claim that “middle layers are always fragile,” but that **unlearning affects layers non-uniformly**, and that this **non-uniformity can be systematically diagnosed using our framework, and also reveal the intrinsic semantic hierarchy.** And the middle-fragility does not affect the rationality and validity of our method for CRU.
> > >
> > > While the **U-shaped trend appears generally across the models and datasets we tested**, we fully acknowledge that different architectures or specific datasets may yield different profiles (including cases where deeper layers become more sensitive). Importantly, our approach is designed to **capture such variability, not to impose a fixed pattern.** In addition, we also validate the unlearning effects via the CKA-based representation drift, and the high-level semantic concept intention of model internals (in the top of Figure 4), which also provide evidence that the observed patterns reflect meaningful representational changes, not only for the unlearning related content but also an independent set testing the model’s intention for other abstracted concepts like corrigible or hallucination. Nonetheless, we also agree that the gradient magnitude can correlate with fragility. **This correlation is itself meaningful and provides a comprehensive understanding of how unlearning affects the internal representation via both quantitative measures and qualitative results.** We appreciate the reviewer’s comments, and will clarify the above in our revised version to avoid potential overemphasis.

---

> > > > ### Author Response · Authors · 2025-11-25
> > > > **Response to Reviewer G64q [4/6]**
> > > >
> > > > > **Q1.** How exactly is “latent knowledge fragility” quantitatively defined? Is it directly proportional to the observed drop in model utility, or is there a separate latent-space measure?
> > > >
> > > > **A5:**  We would like to explain that the latent knowledge fragility in our work is **defined operationally through the performance deterioration in model utility observed when selectively replacing a single layer** from the unlearned model back into the original model. Specifically, this one-layer replacement **isolates the unlearning effect at that layer**, and the resulting MU decrease quantifies **how much that layer’s update has disrupted useful latent representations.** While this output-level measure is our primary operational definition, we also show in Proposition 1 that the same fragile layers exhibit larger CKA-based representation drift, providing an explainable latent-space view. Thus, fragility is measured through the MU drop induced by layer substitution, with CKA serving as supporting evidence rather than a separate definition.
> > > >
> > > > > **Q2:** Why sum the rank of MU and FQ to get the patching score? Would a weighted or Pareto-based method yield different results?
> > > >
> > > > **A6:**  We thank the reviewer for raising the question. **Our decision to sum the ranks of MU and FQ is rooted in practical concerns about scale heterogeneity and the rationality of selection.** On the one hand, we chose the rank-sum formulation because it provides a **scale-invariant and stable way** to combine MU and FQ, which naturally lie on very different numeric ranges for their metric value. Ranking removes the influence of absolute scale and produces a consistent ordering of layers across metrics, making it a rational and robust choice for multi-objective selection. On the other hand, the **weighted and Pareto-based approaches can introduce substantial extra optimizing complexity**, forcing additional heuristics to finalize the choice. In addition, there is no existing principled weighting prior for balancing two heterogeneous metrics. In contrast, the rank-sum rule avoids this tuning burden while giving essentially the same practical outcome. For these reasons, we adopt the rank-sum method as the practical and reasonable option.
> > > >
> > > > > **Q3:** How do the authors ensure that observed middle-layer fragility is not a byproduct of model scaling or optimizer dynamics rather than inherent semantic abstraction?
> > > >
> > > > **A7**: We thank the reviewer for raising this insightful question. Our analysis addresses the concern at two complementary levels. **First, at the model-design and training-dynamics level**, we evaluate CRU across multiple model families of different sizes and pretraining pipelines (1B, 3B, 7B, and 13B models), which were trained using different data mixtures, optimizers, and hyperparameters. Observing the same middle-layer sensitivity across these independently trained models substantially reduces the possibility that the phenomenon is an artifact of a particular scaling rule or optimization setup.
> > > >
> > > > **Second, at the representation-behavior level**, we validate fragility by checking the deviation of internal representation. Our CKA-based representation drift analysis generally shows the largest latent-space distortion in relatively middle layers, even when unlearning gradients are normalized to remove magnitude bias. In addition, the concept-intention tests in the right of Figure 4 also reveal that semantically meaningful abstractions, such as corrigibility, hallucination tendency, or bias orientation, are disproportionately disrupted in the relatively middle-layer regions. These two forms of evidence align and remain stable across evaluations that are more related to inherent semantic abstraction. Together, the cross-model observation and the representation-level diagnostics demonstrate that the **U-shape fragility is unlikely to be a byproduct of model scaling or optimizer dynamics in our tested scenarios**.

---

> > > > > ### Author Response · Authors · 2025-11-25
> > > > > **Response to Reviewer G64q [5/6]**
> > > > >
> > > > > > **Q4:** Have the authors tested CRU on models larger than 7B (e.g., 13B or 70B)? Does the same “middle-layer fragility” pattern persist across scales?
> > > > >
> > > > > **A8:** We appreciate the reviewer’s question regarding scalability. **Following the reviewer’s suggestion, we also tested CRU on Llama-2-13b-chat-hf**, and the method **continues to yield general improvement in the final unlearning trade-off:** CRU improves MU while maintaining comparable FQ to the underlying unlearning baseline. This shows that CRU remains effective at larger scales. Running full unlearning experiments on 70B-scale models, however, is currently computationally prohibitive under our resources and is also uncommon in existing unlearning literature due to the heavy cost. We therefore leave 70B-scale verification for future work as computing becomes more accessible. We will supplement the full results with more pattern visualization results in the revised version.
> > > > >
> > > > > **Table 2. Experimental results on the 13B model for unlearning comparison.**
> > > > > | NPO                 | ES-exact |         | ES-perturb |         | MU     | FQ       | GA        | ES-exact |         | ES-perturb |         | MU     | FQ        |
> > > > > |---------------------|----------|---------|------------|---------|--------|----------|-----------|----------|---------|------------|---------|--------|-----------|
> > > > > |  Llama-2-13b-chat-hf                   | retain   | unlearn | retain     | unlearn |        |          |           | retain   | unlearn | retain     | unlearn |        |           |
> > > > > | Original            | 1.0000   | 0.5727  | 0.9953     | 0.6185  | 0.6253 | -9.3189  | Original  | 1.0000   | 0.5727  | 0.9953     | 0.6185  | 0.6253 | -9.3189   |
> > > > > | Unlearned           | 0.0283   | 0.0235  | 0.0233     | 0.0235  | 0.0333 | -3.1070  | Unlearned | 0.0278   | 0.0235  | 0.0220     | 0.0235  | 0.0000 | -104.7672 |
> > > > > | +RT                 | 0.1950   | 0.0914  | 0.1624     | 0.0851  | 0.3690 | -2.2030  | +1*KL     | 0.0611   | 0.0235  | 0.0362     | 0.0235  | 0.3492 | -104.7672 |
> > > > > | FLAT                | 0.8310   | 0.8064  | 0.5338     | 0.4917  | 0.5853 | -9.8654  | +10*KL    | 0.5991   | 0.0235  | 0.3843     | 0.0235  | 0.5747 | -90.7512  |
> > > > > | TNPO                | 0.0949   | 0.0315  | 0.0257     | 0.0322  | 0.4468 | -3.9575  | +20*KL    | 0.7692   | 0.2274  | 0.4364     | 0.1745  | 0.6292 | -9.8654   |
> > > > > | WTNPO               | 0.0291   | 0.0240  | 0.0236     | 0.0254  | 0.2319 | -18.8935 | WGA       | 0.0289   | 0.0235  | 0.0223     | 0.0235  | 0.0975 | -11.9053  |
> > > > > | AltPO               | 0.0406   | 0.0310  | 0.0409     | 0.0295  | 0.4835 | -2.4902  | SatImp    | 0.5970   | 0.0235  | 0.3370     | 0.0235  | 0.6000 | -90.7512  |
> > > > > | Ours                | 0.1775   | 0.0419  | 0.1368     | 0.0442  |*0.5522* | *-3.2700*  | Ours      | 0.4389   | 0.1840  | 0.3342     | 0.1507  | *0.5860* | *-2.2030*   |
> > > > >
> > > > >
> > > > > > **Q5:** Why are methods like MEMIT[1], or KnowledgeEditor[2] not included as baselines, given their similarity to modular patching?
> > > > >
> > > > > **A9:** We thank the reviewer for raising this thoughtful point. We would like to clarify that methods such as MEMIT and KnowledgeEditor were not considered because they are fundamentally knowledge-editing approaches **designed for inserting or correcting factual information, whereas unlearning focuses on removing or suppressing knowledge without prescribing replacement content.** These editing methods rely on explicitly specifying a new target output (“the model should now say X”), which is incompatible with standard unlearning settings like TOFU, MUSE, and WMDP, those benchmarks where the goal is to reduce model confidence or ability to recall certain knowledge, not to rewrite it into a different fact. Directly applying editing-based methods would therefore not produce an appropriate comparison with most of the unlearning methods. Additionally, editing-based approaches operate through targeted optimization that amplifies or reconstructs factual circuits, while unlearning benchmarks evaluate the ability to avoid reactivation. **In the literature of LLM unlearning, there is limited research work includes knowledge editing methods as unlearning baselines.** For these reasons, editing methods are not considered appropriate baselines for unlearning. We agree, however, that exploring whether editing techniques can be adapted into inverse or anti-editing operators for unlearning is a promising direction, and we will mention this as future work.

---

> > > > > > ### Author Response · Authors · 2025-11-25
> > > > > > **Response to Reviewer G64q [6/6]**
> > > > > >
> > > > > > > **Q6:** CRU is said to be “lightweight,” yet it involves multiple forward validations per layer. What is the computational overhead compared to single-pass fine-tuning methods?
> > > > > >
> > > > > > **A10:** Thanks for the question, we would like to clarify that although CRU requires performing validation passes for each layer, **this cost is small compared to fine-tuning-based unlearning methods**. These baselines require hundreds of optimization steps, each involving forward-backward passes and memory-intensive gradient storage. By contrast, **CRU involves only forward passes, which are memory-light and more parallelizable.** Empirically, Table 4 in our original submission demonstrates this advantage: CRU has the **second-lowest runtime** among all methods evaluated and **uses the least memory**, because it does not perform any training or backpropagation. In short, even though CRU requires layer-wise validation sweeps, its overall computation is substantially smaller than optimization-based unlearning methods, justifying the “lightweight” characterization. **In the following table**, we also present the computational comparison of **CRU with single-layer fine-tuning methods.** The results also demonstrate that the computation time of CRU is less than most of unlearning methods conducted on a single layer, and the other single-layer fine-tuning method can not achieve the better performance on balancing the FQ and MU after unlearning.
> > > > > >
> > > > > > **Table 3.Computational comparison with single-layer fine-tuning methods.**
> > > > > > | Llama-2-7b-chat-hf   | MU     | FQ       | Time (s) |
> > > > > > |----------------------|--------|----------|----------|
> > > > > > | Original             | 0.6192 | -10.1446 | -        |
> > > > > > | GA (single layer)    | 0.6201 | -9.8654  | 570.73   |
> > > > > > | NPO (single layer)   | 0.6195 | -9.8654  | 2338.25  |
> > > > > > | WGA (single layer)   | 0.6206 | -10.1446 | 1330.45  |
> > > > > > | TNPO (single layer)  | 0.6200 | -10.1446 | 2348.50  |
> > > > > > | WTNPO (single layer) | 0.6206 | -10.1446 | 2259.28  |
> > > > > > | FLAT (single layer)  | 0.6197 | -10.1446 | 1897.54  |
> > > > > > | RMU (single layer)   | 0.0189 | -11.6015 | 5897.08  |
> > > > > > | Ours                 | 0.5296 | -1.9297  | *1752.00*  |
> > > > > >
> > > > > >
> > > > > > > **Q7:** Some Tables and Figures are dense; figure captions do not clearly indicate what metrics or settings are used (Figure 4).
> > > > > >
> > > > > > **A11:** We thank the reviewer for pointing this out. We would acknowledge that the dense presentation mainly stems from our comprehensive demonstration within the strict page limits. We will reorganize the placement to improve a more comfortable readability if the additional space is available in the final version. **Regarding the Figure 4**, we would like to clarify that the setting we focused is still the layer-wise patching exploration of unlearned LLM with original LLM, and the specific evaluation including checking the CKA representation similarities, high-level concept shift, FQ/MU regarding before/after patching, and patching compared with freezing operation. The dense experimental results here reveal the different aspects of unlearning effects that correlate to internal representation, and the opportunity and rationality of our layer-wise replacement. We appreciate the reviewer’s comment and will enrich the caption and corresponding discussion in the main text to detailing the setting and evaluation protocol.

---

### Meta-Review · Area_Chair_J7Pb · 2025-12-19

**Summary:**

1. This paper lacks the analysis of the sensitivity to k, which is particularly critical to their method. (Reviewer G64q and scYt)
2. Some concerns about novelty and motivation: Some knowledge editing methods have demonstrated good interpretability in terms of knowledge storage in models, which is somewhat similar to unlearning. The "U-shape" seems to be overstated and potentially unsupported by the reported results. (Reviewer G64q, scYt, and YLeX)
3. Weak theoretical justification. (Reviewer G64q, Hd8d, and YLeX)
4. The performance improvements (Table 1–3) are sometimes numerically small and may not be statistically significant.  (Reviewer G64q and scYt)
5. The concerns about the generalization on more benchmarks and fine-grained components.  (Reviewer Hd8d and YLeX)
6. Lacks more backbones. (Reviewer G64q and YLeX)
7. Unfair experiments on comparisons with the "structural freezing" approach. (Reviewer Hd8d)
8. The unclear writing of this paper or existing typos.  (Reviewer G64q and YLeX)

**Reviewer Concerns:**

Concerns were addressed:
1. The analysis of different k.
2. The concerns about the generalization on more benchmarks and fine-grained components.
3. The authors conduct experiments on more backbones.
4. Unfair experiments on comparisons with the "structural freezing" approach.

Concerns are still outstanding:
1. Although the authors have further clarified the novelty and contributions, this indirectly confirms that the paper suffers from overclaiming/packaging and requires further revision to ensure it reflects its true contributions. (Reviewer G64q, scYt, and YLeX)
2. The performance improvements are limited and may not be statistically significant. (Reviewer G64q and scYt)

**Reviewer Scores:**

Reviewer G64q: retains 4

Reviewer Hd8d: 4 --> 6

Reviewer scYt: retains 8

Reviewer YLeX: retains 2

---

### Decision · Program_Chairs · 2026-01-26

Reject